# Rank-1 Matrix Completion with Gradient Descent and Small Random Initialization

**Daesung Kim**
Samsung Electronics
dskim95phd@gmail.com

**Hye Won Chung**
School of Electrical Engineering
KAIST
hwchung@kaist.ac.kr

## Abstract

The nonconvex formulation of the matrix completion problem has received significant attention in recent years due to its affordable complexity compared to the convex formulation. Gradient Descent (GD) is a simple yet efficient baseline algorithm for solving nonconvex optimization problems. The success of GD has been witnessed in many different problems in both theory and practice when it is combined with random initialization. However, previous works on matrix completion require either careful initialization or regularizers to prove the convergence of GD. In this paper, we study the rank-1 symmetric matrix completion and prove that GD converges to the ground truth when small random initialization is used. We show that in a logarithmic number of iterations, the trajectory enters the region where local convergence occurs. We provide an upper bound on the initialization size that is sufficient to guarantee the convergence, and show that a larger initialization can be used as more samples are available. We observe that the implicit regularization effect of GD plays a critical role in the analysis, and for the entire trajectory, it prevents each entry from becoming much larger than the others.

## 1 Introduction

Recovering a low-rank matrix from a set of linear measurements is at the heart of many statistical learning problems. Depending on the structure of the matrix and the linear measurements, it reduces to various problems such as phase retrieval [1], blind deconvolution [2], and matrix sensing [3]. Matrix completion [4] is also one such type of problem where each measurement provides an entry of the matrix, and the goal is to recover the low-rank matrix from a partial, usually very sparse, observation of the entries. One of the most notable applications of matrix completion is collaborative filtering [5], which aims to predict user preferences for items based on a highly incomplete observation of user-item ratings. There are also a number of different applications, such as principal component analysis [6] and image reconstruction [7], just to name a few.

Extensive amount of work has been dedicated to provide an efficient recovery algorithm for matrix completion with theoretical guarantees [8]. The convex relaxation based nuclear norm minimization [4, 9] was the first algorithm proven to recover the matrix with near optimal sample complexity. Despite its theoretical success, the convex algorithm was found hard to be used in practical scenarios due to its unaffordable computational complexity and memory size. Therefore, the nonconvex formulation of matrix completion with quadratic loss has received significant attention in recent years. Many different algorithms have been proposed for the nonconvex problem, and their convergence toward the ground truth has been analyzed. Examples include optimization on Grassmann manifolds [10], alternating minimization [11], projected gradient descent [12], gradient descent with regularizer [13], and (vanilla) gradient descent [14, 15].

37th Conference on Neural Information Processing Systems (NeurIPS 2023).

Gradient descent (GD) has served as a baseline algorithm for solving nonconvex optimization problems. However, the convergence of GD to global minimizers is not guaranteed, and it can take exponential time to escape saddle points [16]. Nevertheless, GD with random initialization has been shown to successfully recover the global minimum in many different problems such as phase retrieval [1], matrix sensing [17], matrix factorization [18], and neural network training [19]. Previous work on matrix completion [14, 15] proved the convergence of GD under the spectral initialization, which locates the initial point in the local region of the minima. However, the role of random initialization in solving matrix completion with GD is not fully understood yet, although its success is observed in practice. Therefore, we aim to answer the following question:

*Can GD with random initialization solve the nonconvex matrix completion problem?*

We answer this question affirmatively and show that GD with small random initialization successfully converges to the ground truth for rank-1 symmetric matrix completion. In the analysis, we use vanilla GD, which does not incorporate any modifications, such as regularization or truncation, into the GD algorithm. We also characterize the entire trajectory that GD follows by showing that the trajectory is well approximated by the fully observed case. The small initialization plays a critical role in analyzing the trajectory of the early stages, where the randomly initialized vector is nearly orthogonal to the first eigenvector of the ground truth matrix. We provide a bound on the required initialization size for the algorithm to converge, and our bound suggests that one can use a larger initialization to improve the convergence speed as more samples are provided. However, in any case, GD with a small random initialization takes only logarithmic amount of time (with respect to the matrix dimension) to reach the point where local convergence can begin. To the best of our knowledge, this is the first result on matrix completion that proves the convergence of vanilla GD without a carefully designed initialization.

Although our result is restricted to the rank-1 case, we believe that this work provides an important evidence for understanding the more general rank-$r$ case. At the end of this paper, we will discuss some technical difficulties that the rank-$r$ case naturally has, and provide some empirical results related to them. However, studying the rank-1 matrix completion problem is not only motivated by theoretical interest, but the problem itself also appears in some practical problems such as crowdsourcing [20, 21].

**Related Works** This work is motivated by the recent success of small initialization in matrix factorization and matrix sensing. It was first conjectured in [22] that sufficiently small step sizes and initialization lead GD to converge to the minimum nuclear norm solution of a full-dimensional matrix sensing problem. The conjecture was proved in [17] for the fully overparameterized matrix sensing under the standard restricted isometry property (RIP). A recent study by [23] provided more general results by showing that the early iterations of GD with small initialization have spectral bias. Many other works such as [24, 25, 26] have also studied how GD or gradient flow with small initialization implicitly forces the recovered matrix to be low-rank. However, the recovery guarantee for matrix completion has not been provided by any work.

For the matrix sensing where RIP holds, the loss function has global benign geometry in that it does not contain any spurious local minima or non-strict saddle points [27]. In the case of matrix completion, a similar result was obtained but with a regularizer that penalizes the matrices with large rows [28]. Controlling the norm of each row (absolute value of each entry in the case of rank-1) is the biggest hurdle in the analysis of matrix completion. In the local convergence analysis of [14], it was proved that GD implicitly regularizes the largest $\ell_2$-norm of the rows of error matrices, showing that explicit regularization is unnecessary. In this paper, we also prove that such an implicit regularization is induced by GD when it starts from a point of small size. We show that the trajectory is close to the fully observed case in both $\ell_2$ and $\ell_\infty$ norms. Thus, the trajectory is confined to the region where it has benign geometry, and GD can converge without an explicit regularizer.

**Notations** We denote vectors with lowercase bold letters and matrices with uppercase bold letters. The components or entries of them are written without bold. We use $\|\cdot\|_2$ and $\|\cdot\|_\infty$ to denote $\ell_2$ and $\ell_\infty$-norm of vectors, respectively, and $\|\cdot\|_F$ is used for Frobenius norm of matrices. For any norm $\|\cdot\|$ and two vectors $\boldsymbol{x}, \boldsymbol{y}$, we let $\|\boldsymbol{x} \pm \boldsymbol{y}\| = \min\{\|\boldsymbol{x} + \boldsymbol{y}\|, \|\boldsymbol{x} - \boldsymbol{y}\|\}$. Asymptotic dependencies with respect to the matrix dimension are denoted with the standard big $O$ notations, or with the symbols, $\lesssim, \asymp$ and $\gtrsim$.

## 2 Problem Formulation

The matrix completion problem aims to reconstruct a low-rank matrix from partially observed entries. In this paper, we focus on the case where the ground truth matrix, denoted by $\boldsymbol{M}^\star \in \mathbb{R}^{n \times n}$, is a rank-1 positive semidefinite matrix. Thus, the ground truth matrix is decomposed as $\boldsymbol{M}^\star = \lambda^\star \boldsymbol{u}^\star \boldsymbol{u}^{\star\top}$ with $\lambda^\star > 0$ and a unit vector $\boldsymbol{u}^\star$. We define $\boldsymbol{x}^\star = \sqrt{\lambda^\star} \boldsymbol{u}^\star$ so that $\boldsymbol{M}^\star = \boldsymbol{x}^\star \boldsymbol{x}^{\star\top}$. To follow the standard incoherence assumption, we let $\|\boldsymbol{u}^\star\|_\infty = \sqrt{\frac{\mu}{n}}$ and allow $\mu$ to be as large as $\mathrm{poly}(\log n)$. We consider a random sampling model that is also symmetric as $\boldsymbol{M}^\star$. Each entry in the diagonal and the upper (or lower) triangular part of $\boldsymbol{M}^\star$ is independently revealed with probability $0 < p \le 1$. We consider the noisy case where Gaussian noise is added to each observation. Formally, we get as an observation the matrix $\boldsymbol{M}^\circ$ whose $(i,j)$th entry is $\frac{1}{p}\delta_{ij}(M^\star_{ij} + E_{ij})$, where $[\delta_{ij}]_{1 \le i \le j \le n}$ are independent Bernoulli random variables with expectation $p$ and $[E_{ij}]_{1 \le i \le j \le n}$ are independent Gaussian random variables with the distribution $\mathcal{N}(0, \sigma^2)$. They are both symmetric in the sense that $\delta_{ij} = \delta_{ji}$ and $E_{ij} = E_{ji}$ for all $1 \le i \le j \le n$. We use $\boldsymbol{E}$ to denote the symmetric matrix whose entries are $E_{ij}$. We denote the set of observed entries as $\Omega := \{(i,j) \mid \delta_{ij} = 1\}$, and define an operator $\mathcal{P}_\Omega$ on matrices that sets the entries not contained in $\Omega$ to zero. (e.g. $\boldsymbol{M}^\circ = \frac{1}{p}\mathcal{P}_\Omega(\boldsymbol{M}^\star + \boldsymbol{E})$)

To recover the matrix $\boldsymbol{M}^\star$, we find $\boldsymbol{x} \in \mathbb{R}^n$ that minimizes the nonconvex loss function $f(\boldsymbol{x})$, which is the sum of the squared differences on the observed entries. It is explicitly written as $f(\boldsymbol{x}) := \frac{1}{4p}\sum_{(i,j)\in\Omega}(x_i x_j - x^\star_i x^\star_j - E_{ij})^2$. We apply vanilla GD to solve the optimization problem starting from a small randomly initialized vector $\boldsymbol{x}^{(0)}$. Each entry of $\boldsymbol{x}^{(0)}$ is sampled independently from the Gaussian distribution $\mathcal{N}\left(0, \frac{1}{n}\beta_0^2\right)$, so that the squared norm of $\boldsymbol{x}^{(0)}$ is expected to be $\beta_0^2$. The update rule of GD is written as

$$\boldsymbol{x}^{(t+1)} = \boldsymbol{x}^{(t)} - \eta\nabla f\left(\boldsymbol{x}^{(t)}\right) = \boldsymbol{x}^{(t)} - \frac{\eta}{p}\mathcal{P}_\Omega\left(\boldsymbol{x}^{(t)}\boldsymbol{x}^{(t)\top}\right)\boldsymbol{x}^{(t)} + \eta\boldsymbol{M}^\circ\boldsymbol{x}^{(t)}, \tag{1}$$

where $\eta > 0$ is the step size.

We define $F$ as the loss function $f$ when all entries of $\boldsymbol{M}^\star$ are observed without noise, i.e., $F(\boldsymbol{x}) := \frac{1}{4}\left\|\boldsymbol{x}\boldsymbol{x}^\top - \boldsymbol{M}^\star\right\|_\mathrm{F}^2$. We also define $\widetilde{\boldsymbol{x}}^{(t)}$ as the trajectory of GD when it is applied to $F$ with the same initial point $\boldsymbol{x}^{(0)}$, i.e., $\widetilde{\boldsymbol{x}}^{(t)}$ is the trajectory of the fully observed case. Specifically, it evolves with

$$\widetilde{\boldsymbol{x}}^{(t+1)} = \widetilde{\boldsymbol{x}}^{(t)} - \eta\nabla F(\widetilde{\boldsymbol{x}}^{(t)}) = \widetilde{\boldsymbol{x}}^{(t)} - \eta\left\|\widetilde{\boldsymbol{x}}^{(t)}\right\|_2^2\widetilde{\boldsymbol{x}}^{(t)} + \eta\boldsymbol{M}^\star\widetilde{\boldsymbol{x}}^{(t)} \tag{2}$$

from the same starting point $\widetilde{\boldsymbol{x}}^{(0)} = \boldsymbol{x}^{(0)}$.

Lastly, we introduce the so-called *leave-one-out* sequences. These were the main ingredient in controlling the $\ell_\infty$-norm of trajectory in [14]. We use them for a similar purpose. For each $l \in [n]$, we define an operator $\mathcal{P}_\Omega^{(l)}$ such that $\mathcal{P}_\Omega^{(l)}(\boldsymbol{X})$ is equal to $\boldsymbol{X}$ on the $l$th row and column, and equal to $\frac{1}{p}\mathcal{P}_\Omega(\boldsymbol{X})$ otherwise. The $l$th leave-one-out sequence, $\boldsymbol{x}^{(t,l)}$, evolves with

$$\boldsymbol{x}^{(t+1,l)} = \boldsymbol{x}^{(t,l)} - \eta\mathcal{P}_\Omega^{(l)}\left(\boldsymbol{x}^{(t,l)}\boldsymbol{x}^{(t,l)\top}\right)\boldsymbol{x}^{(t,l)} + \eta\boldsymbol{M}^{(l)}\boldsymbol{x}^{(t,l)}, \tag{3}$$

for $\boldsymbol{x}^{(0,l)} = \boldsymbol{x}^{(0)}$, where $\boldsymbol{M}^{(l)} = \mathcal{P}_\Omega^{(l)}(\boldsymbol{M}^\star) + \boldsymbol{E}^{(l)}$, and $\boldsymbol{E}^{(l)}$ is obtained by zeroing out the $l$th row and column of $\frac{1}{p}\mathcal{P}_\Omega(\boldsymbol{E})$.

## 3 Main Results

In this section, we present our main results. The first main result concerns the global convergence of GD with small random initialization.

**Theorem 3.1.** *Let us consider a rank-1 matrix completion problem that recovers the matrix $\boldsymbol{M}^\star = \boldsymbol{x}^\star\boldsymbol{x}^{\star\top} \in \mathbb{R}^{n \times n}$ such that $\|\boldsymbol{x}^\star\|_2 = \sqrt{\lambda^\star}$ and $\|\boldsymbol{x}^\star\|_\infty = \sqrt{\frac{\mu}{n}}\|\boldsymbol{x}^\star\|_2$, where $\mu = O(\mathrm{poly}(\log n))$. Let the initial point $\boldsymbol{x}^{(0)} \in \mathbb{R}^n$ be sampled from the Gaussian distribution $\mathcal{N}(\boldsymbol{0}, \frac{1}{n}\beta_0^2\boldsymbol{I})$ and $\boldsymbol{x}^{(t)}$ be updated with* (1). *Suppose that a small step size with $\eta\lambda^\star < 0.1$ is used and the sample complexity satisfies $n^2p \gtrsim \mu^5 n\log^{22} n$. Then, there exists $T^\star = (1 + o(1))\frac{1}{\eta\lambda^\star}\log\frac{\sqrt{\lambda^\star}n}{\beta_0}$ such that*

$$\left\| \boldsymbol{x}^{(t)} \pm \boldsymbol{x}^\star \right\|_2 \lesssim \frac{1}{\sqrt{\log n}} \|\boldsymbol{x}^\star\|_2, \qquad (4) \qquad\qquad \max_{1 \le l \le n} \left\| \boldsymbol{x}^{(t)} - \boldsymbol{x}^{(t,l)} \right\|_2 \lesssim \frac{1}{\sqrt{\log n}} \|\boldsymbol{x}^\star\|_\infty, \quad (6)$$

$$\left\| \boldsymbol{x}^{(t)} \pm \boldsymbol{x}^\star \right\|_\infty \lesssim \frac{1}{\sqrt{\log n}} \|\boldsymbol{x}^\star\|_\infty, \qquad (5) \qquad\qquad \max_{1 \le l \le n} \left| (\boldsymbol{x}^{(t,l)} - \boldsymbol{x}^\star)_l \right| \lesssim \frac{1}{\sqrt{\log n}} \|\boldsymbol{x}^\star\|_\infty \quad (7)$$

*hold at $t = T^\star$ with probability at least $1 - o(1/\sqrt{\log n})$, if a sufficiently small initialization with*

$$\sqrt{\lambda^\star} n^{-10} \lesssim \beta_0 \lesssim \sqrt{\lambda^\star} \sqrt[4]{\frac{np}{\mu^5 \log^{26} n}} \frac{1}{\sqrt[4]{n}} \tag{8}$$

*is used and the noise satisfies $\sigma \lesssim \frac{\lambda^\star \mu}{n} \sqrt{\log n}$.*

Theorem 3.1 proves that, starting from a small random initialization, the trajectory of GD eventually enters the local region of the global minimizers $\pm \boldsymbol{x}^\star$ in terms of both $\ell_2$ and $\ell_\infty$ norms. Combined with the result of [14], GD starts to converge linearly to either $\boldsymbol{x}^\star$ or $-\boldsymbol{x}^\star$ after $t = T^\star$, as stated in the corollary below.

**Corollary 3.2.** *Suppose that the conditions in Theorem 3.1 are satisfied, and let $\rho$ be a constant such that $1 - \frac{\eta}{10} \le \rho < 1$. Then, with probability at least $1 - o(1/\sqrt{\log n})$, we have*

$$\left\| \boldsymbol{x}^{(t)} \pm \boldsymbol{x}^\star \right\|_2 \lesssim \left( \frac{1}{\sqrt{\log n}} \rho^{t - T^\star} + \frac{\sigma}{\lambda^\star} \sqrt{\frac{n}{p}} \right) \|\boldsymbol{x}^\star\|_2, \tag{9}$$

$$\left\| \boldsymbol{x}^{(t)} \pm \boldsymbol{x}^\star \right\|_\infty \lesssim \left( \frac{1}{\sqrt{\log n}} \rho^{t - T^\star} + \frac{\sigma}{\lambda^\star} \sqrt{\frac{n}{p}} \right) \|\boldsymbol{x}^\star\|_\infty, \tag{10}$$

*for all $T^\star \le t \le T = O(n^5)$.*

The desired global convergence result is provided by Corollary 3.2. Several remarks about Theorem 3.1 and Corollary 3.2 are in order.

**Matrix Recovery** Suppose $\boldsymbol{x}^{(t)}$ converges to a global minimum $\boldsymbol{y}^\star$ of the function $f$, which is different from $\pm \boldsymbol{x}^\star$. In such a case, despite achieving global convergence, the reconstructed matrix $\boldsymbol{y}^\star \boldsymbol{y}^{\star\top}$ deviates from the ground truth matrix $\boldsymbol{M}^\star$. However, Theorem 3.1 establishes that $\boldsymbol{x}^{(t)}$ converges exclusively to the correct global minima $\pm \boldsymbol{x}^\star$, so that the matrix $\boldsymbol{M}^\star$ is recovered with high probability.

**Leave-one-out Sequence** To apply the local convergence result of [14], in addition to (4) and (5), the existence of leave-one-out sequences $\{\boldsymbol{x}^{(t,l)}\}_{l \in [n]}$ satisfying (6) and (7) is required. Leave-one-out sequences also play a critical role and appear naturally in the proof of Theorem 3.1.

**Sample Complexity** The required sample complexity for Theorem 3.1 to hold is optimal up to a logarithmic factor compared to the statistical lower bound of $\Omega(n \log n)$. We have not done our best to optimize the log factors, and about half of them can be reduced with more delicate analysis. We will discuss this briefly in Section 6.

**Convergence Time** Considering that $\beta_0^{-1}$ is at most polynomial in $n$ (due to the lower bound of (8)), only $O(\log n)$ iterations are required for GD to enter the local region. It takes $O(\log(\frac{1}{\epsilon}))$ more iterations to achieve $\epsilon$-accuracy in the local region, so the total iteration complexity is given by $O(\log n) + O(\log(\frac{1}{\epsilon}))$.

**Initialization Size** Although small initialization provides a good geometry to GD, a larger initialization is preferred because the convergence time, $T^\star$, is inversely proportional to $\beta_0$. When the sample complexity is optimal, i.e., $n^2 p \asymp n \operatorname{poly}(\log n)$, an upper bound on the initialization size given by Theorem 3.1 is $n^{-\frac{1}{4}}$, ignoring the log factors. However, as more samples are provided, we are allowed to use a larger initialization to reduce the convergence time. When the sample complexity satisfies $n^2 p \asymp n^{1+a}$, the bound is $n^{-\frac{1}{4}(1-a)}$ ignoring the log factors. The bound becomes nearly constant as $a$ approaches 1, namely the fully observed case, and this is consistent with the previous result that small initialization is unnecessary for the fully observed case [18]. We also note that the lower bound of (8) is necessary in the proof of Theorem 3.1, since we derive probabilistic bounds for all iterations, and the lower bound limits the maximum number of iterations. However, we can further reduce the lower bound $n^{-10}$ to $n^{-c}$ for any constant $c > 10$ by tuning some constant factors during the proof.

**Noise Size** From the incoherence assumption, the maximum absolute value of entries of $M^\star$ is bounded by $\frac{\lambda^\star \mu}{n}$. The condition $\sigma \lesssim \frac{\lambda^\star \mu}{n}\sqrt{\log n}$ in Theorem 3.1 allows the standard deviation of the Gaussian noise to be much larger than the maximum entry. It also implies $\frac{\sigma}{\lambda^\star}\sqrt{\frac{n}{p}} \lesssim \mu\sqrt{\frac{\log n}{np}}$, so that the upper bounds in Corollary 3.2 are dominated by the first terms at $t = T^\star$ and they eventually converge to the second terms as $t$ increases.

**Estimation Error** The current estimation bounds (4) to (7) are all proportional to $\frac{1}{\sqrt{\log n}}$ times the norms of $x^\star$. However, if we do not allow the initialization size to grow with the sample complexity, we are able to obtain tighter bounds; if we use the fixed initialization size $n^{-\frac{1}{4}}$ regardless of the sample complexity, in Theorem 3.1, the factor $\frac{1}{\sqrt{\log n}}$ is improved to $\frac{1}{\sqrt{np}} + \frac{\sigma}{\lambda^\star}\sqrt{\frac{n}{p}}$, and the upper bound on noise size is also improved to $\sigma \lesssim \frac{\lambda^\star \mu}{n}\sqrt{np}$ (not being precise on the factors of $\mu$ and $\log n$ here). Then, the estimation error in Corollary 3.2 is improved to $\frac{1}{\sqrt{np}}\rho^t + \frac{\sigma}{\lambda^\star}\sqrt{\frac{n}{p}}$ to match the result of [14] which uses spectral initialization. Thus, we have a tradeoff between estimation error and initialization size.

The next main result concerns the trajectory of GD before it enters the local region. The theorem states that for all $t \leq T^\star$, $x^{(t)}$ stays close to the fully observed case $\widetilde{x}^{(t)}$ in both $\ell_2$ and $\ell_\infty$-norm.

**Theorem 3.3.** *Suppose that the conditions of Theorem 3.1 hold, and $T^\star$ is defined as in Theorem 3.1. Then, for all $t \leq T^\star$, we have*

$$\left\|x^{(t)} - \widetilde{x}^{(t)}\right\|_2 \lesssim \frac{1}{\sqrt{\log n}}\left\|\widetilde{x}^{(t)}\right\|_2, \qquad (11) \qquad \left\|x^{(t)} - \widetilde{x}^{(t)}\right\|_\infty \lesssim \frac{1}{\sqrt{\log n}}\left\|\widetilde{x}^{(t)}\right\|_\infty \qquad (12)$$

*with probability at least $1 - o(1/\sqrt{\log n})$.*

**Trajectory of GD** The sequence $\widetilde{x}^{(t)}$ is a linear combination of $x^{(0)}$ and $u^\star$ (see (C.1) in the appendix), and it is easy to analyze how $\widetilde{x}^{(t)}$ evolves. By showing that $x^{(t)}$ stays close to $\widetilde{x}^{(t)}$ for all iterations, we not only show the convergence of GD with small initialization as in Theorem 3.1, but also characterize the exact trajectory that GD follows by Theorem 3.3.

**Implicit Regularization** One can prove that $\widetilde{x}^{(t)}$ is incoherent up to some log factors over all iterations, and from (11) and (12), the incoherence of $x^{(t)}$ is bounded by that of $\widetilde{x}^{(t)}$. Thus, Theorem 3.3 shows that the incoherence of $x^{(t)}$ is *implicitly* controlled by GD without any regularizer. This is an improvement over the previous result on the global convergence of GD for matrix completion [28], where an explicit regularizer was used to control the $\ell_\infty$-norm of $x^{(t)}$, although no small initialization was used in that work.

## 4 Fully Observed Case and Proof Sketch

Before we explain the proof of Theorems 3.1 and 3.3, we describe the trajectory of the fully observed case. We characterize $\widetilde{x}^{(t)}$ with three variables: $\widetilde{\alpha}_t = \left|u^{\star\top}\widetilde{x}^{(t)}\right|$, $\widetilde{\beta}_t = \|\widetilde{x}^{(t)}\|_2$, and $\widetilde{\gamma}_t = \|\widetilde{x}_\perp^{(t)}\|_2$, where $\widetilde{x}_\perp^{(t)} = \widetilde{x}^{(t)} - u^\star u^{\star\top}\widetilde{x}^{(t)}$. According to (2), the three variables are updated with

$$\widetilde{\alpha}_{t+1} = (1 - \eta\widetilde{\beta}_t^2 + \eta\lambda^\star)\widetilde{\alpha}_t; \quad \widetilde{\gamma}_{t+1} = (1 - \eta\widetilde{\beta}_t^2)\widetilde{\gamma}_t;$$
$$\widetilde{\beta}_t^2 = \widetilde{\alpha}_t^2 + \widetilde{\gamma}_t^2.$$

At $t = 0$, due to random initialization, the initial vector is nearly orthogonal to $u^\star$, and we have $\widetilde{\alpha}_0 \approx \frac{1}{\sqrt{n}}\beta_0$ and $\widetilde{\gamma}_0 \approx \widetilde{\beta}_0 = \beta_0$. Also, due to the small initialization, the term $\eta\widetilde{\beta}_t^2$ is ignorable until $\widetilde{\beta}_t$ becomes sufficiently large, so $\widetilde{\alpha}_t$ grows exponentially at the rate of $1 + \eta\lambda^\star$, while $\widetilde{\gamma}_t$ remains still. Thus, in the early iterations where $(1+\eta\lambda^\star)^t$ is still much less than $\sqrt{n}$, $\widetilde{\beta}_t$ is kept close to its initial

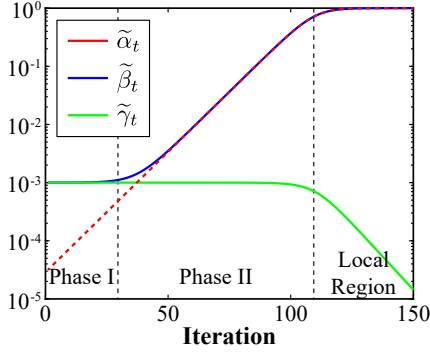

Figure 1: Evolution of the quantities $\widetilde{\alpha}_t$, $\widetilde{\beta}_t$, and $\widetilde{\gamma}_t$ simulated with $\widetilde{\alpha}_0 = \frac{1}{\sqrt{n}}\beta_0$, $\beta_0 = \frac{1}{n}$, $\lambda^\star = 1$, and $n = 1000$.

value $\beta_0$ while the trajectory becomes more parallel to $\boldsymbol{u}^\star$ as $\widetilde{\alpha}_t$ increases. When $(1+\eta\lambda^\star)^t$ becomes much larger than $\sqrt{n}$, the trajectory becomes almost parallel to $\boldsymbol{u}^\star$ in that $\widetilde{\beta}_t \approx \widetilde{\alpha}_t \gg \widetilde{\gamma}_t$. Until $\widetilde{\beta}_t$ (asymptotically) reaches $\frac{\sqrt{\lambda^\star}}{\sqrt{\log n}}$, we can consider $\widetilde{\alpha}_t$ as increasing at a rate of $(1+\eta\lambda^\star)$, and it takes about $\frac{1}{\log(1+\eta\lambda^\star)}\log\frac{\sqrt{\lambda^\star n}}{\beta_0}$ steps to reach this point. After that, we can no longer ignore the term $\eta\widetilde{\beta}_t^2$, and $\widetilde{\alpha}_t$ increases at a slower rate as $\widetilde{\beta}_t$ increases. We can show that $\widetilde{\beta}_t^2$ becomes sufficiently close to $\lambda^\star$ within $O(\log\log n)$ additional iterations, as stated in the following lemma.

**Lemma 4.1.** *Let $T_2'$ be the largest $t$ such that $\widetilde{\beta}_t^2 \leq \frac{\lambda^\star}{64\log n}$. At $t = T_2' + \frac{6\log\log n}{\log(1+\eta\lambda^\star)}$, we have $\widetilde{\beta}_t^2 \geq \lambda^\star\left(1 - \frac{1}{\log n}\right)$.*

Finally, local convergence to $\boldsymbol{u}^\star$ occurs in that $\widetilde{\alpha}_t$ approaches $\lambda^\star$ and $\widetilde{\gamma}_t$ decreases exponentially with the rate $(1-\eta\lambda^\star)$. The actual behavior of quantities $\widetilde{\alpha}_t, \widetilde{\beta}_t, \widetilde{\gamma}_t$ are plotted in Figure 1.

We define the iterates before $(1+\eta\lambda^\star)^t$ reaches $\frac{1}{\sqrt{np}}\sqrt{n}$, within some logarithmic factors, as Phase I, and the next iterates before $\widetilde{\beta}_t^2$ reaches $\lambda^\star\left(1 - \frac{1}{\log n}\right)$ as Phase II. Different techniques are used for each phase to prove that $\boldsymbol{x}^{(t)}$ stays close to $\widetilde{\boldsymbol{x}}^{(t)}$. At the end of Phase I, $\widetilde{\alpha}_t$ is increased to $\frac{1}{\sqrt{np}}\beta_0$ from its initial scale $\frac{1}{\sqrt{n}}\beta_0$, but it is still not dominant over $\beta_0$. Therefore, the magnitudes of both $\boldsymbol{x}^{(t)}$ and $\widetilde{\boldsymbol{x}}^{(t)}$ are kept close to $\beta_0$ throughout Phase I, and we take advantage of the small random initialization to show that the deviation of $\boldsymbol{x}^{(t)}$ from $\widetilde{\boldsymbol{x}}^{(t)}$ does not increase much, and is kept at $\sqrt{\frac{1}{np}}$ times the norms of $\boldsymbol{x}^{(t)}$. In Phase II, we show that $\boldsymbol{x}^{(t)} - \widetilde{\boldsymbol{x}}^{(t)}$ expands at a rate of at most $(1+\eta\lambda^\star)$. Since the norms of $\boldsymbol{x}^{(t)}$ also grows at a rate of $(1+\eta\lambda^\star)$ during most of Phase II, the norms of $\boldsymbol{x}^{(t)} - \widetilde{\boldsymbol{x}}^{(t)}$ remain negligible compared to those of $\boldsymbol{x}^{(t)}$. The next two sections give the main lemmas of Phase I and II, respectively, which are used to prove Theorems 3.1 and 3.3. For a visual representation of the results in the following two sections, please refer to Figure 2.

## 5 Phase I: Finding Direction

We provide detailed results and proof ideas for Phase I. Our main goal is to analyze the deviation of $\boldsymbol{x}^{(t)}$ from $\widetilde{\boldsymbol{x}}^{(t)}$. First, if we look at the update equations (1) and (2), the second term is proportional to the third power of $\left\|\boldsymbol{x}^{(t)}\right\|_2$, while the other terms depend linearly on $\left\|\boldsymbol{x}^{(t)}\right\|_2$. Thus, the second term is almost negligible due to the small initialization. Without the second terms, the difference between $\boldsymbol{x}^{(t)}$ and $\widetilde{\boldsymbol{x}}^{(t)}$ at $t = 1$ is $\eta(\boldsymbol{M}^\circ - \boldsymbol{M}^\star)\boldsymbol{x}^{(0)}$. From concentration inequalities, one can see that the $\ell_2$ and $\ell_\infty$ norms of $\eta(\boldsymbol{M}^\circ - \boldsymbol{M}^\star)\boldsymbol{x}^{(0)}$ are about $\frac{1}{\sqrt{np}}$ times smaller than those of $\widetilde{\boldsymbol{x}}^{(1)}$.

Due to the third terms of (1) and (2), the norms of $\boldsymbol{x}^{(t)} - \widetilde{\boldsymbol{x}}^{(t)}$ can grow exponentially at a rate of $(1+\eta\lambda^\star)$ in the worst case where $\boldsymbol{x}^{(t)} - \widetilde{\boldsymbol{x}}^{(t)}$ is parallel to $\boldsymbol{u}^\star$. In such a case, the norms of $\boldsymbol{x}^{(t)} - \widetilde{\boldsymbol{x}}^{(t)}$ would be larger than those of $\widetilde{\boldsymbol{x}}^{(t)}$ at the end of Phase I, since those of $\widetilde{\boldsymbol{x}}^{(t)}$ remain still in Phase I. However, we overcome this problem by proving that the bounds grow at most *polynomially* with respect to $t$, and since $t$ is at most $O(\log n)$, the bounds remain $\frac{1}{\sqrt{np}}$ times smaller than the norms of $\widetilde{\boldsymbol{x}}^{(t)}$ up to logarithmic factors throughout Phase I.

**Lemma 5.1.** *Let $T_1$ be the largest $t$ such that $(1+\eta\lambda^\star)^t \leq \sqrt{\frac{\mu^4\log^{21}n}{np}}\sqrt{n}$. Under the conditions of Theorem 3.1, with probability at least $1 - o(1/\sqrt{\log n})$, for all $t \leq T_1$, we have*

$$\left\|\boldsymbol{x}^{(t)} - \widetilde{\boldsymbol{x}}^{(t)}\right\|_2 \lesssim \mu\sqrt{\frac{\log n}{np}}\beta_0 t, \quad (13) \qquad \left\|\boldsymbol{x}^{(t)} - \widetilde{\boldsymbol{x}}^{(t)}\right\|_\infty \lesssim \sqrt{\frac{\mu^3\log^2 n}{np}}\frac{\beta_0}{\sqrt{n}}t^2. \quad (14)$$

$T_1$ is defined to be the end of Phase I. Lemma 5.1 proves Theorem 3.3 for Phase I.

**Proof of (13)** We will first demonstrate how to obtain the $\ell_2$-norm bound of Lemma 5.1. Let us define a sequence $\widehat{\boldsymbol{x}}^{(t)}$ that is updated as

$$\widehat{\boldsymbol{x}}^{(t+1)} = \widehat{\boldsymbol{x}}^{(t)} - \eta\left\|\widetilde{\boldsymbol{x}}^{(t)}\right\|_2^2\widehat{\boldsymbol{x}}^{(t)} + \eta\boldsymbol{M}^\circ\widehat{\boldsymbol{x}}^{(t)}; \quad \widehat{\boldsymbol{x}}^{(0)} = \boldsymbol{x}^{(0)}. \quad (15)$$

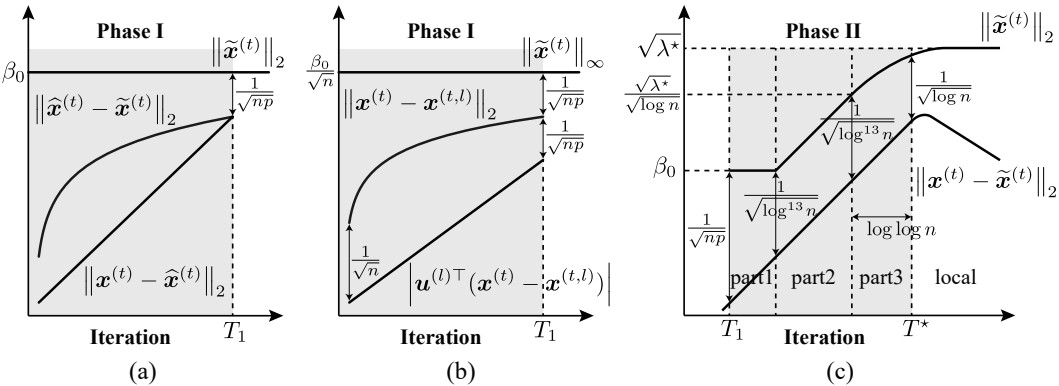

Figure 2: An illustrative description of trajectory of various quantities compared to the norms of $\boldsymbol{x}^{(t)}$ on logarithmic scales. Arrows between lines represent the ratio between them. The quantities depicted are not precise, and only the key factors are shown for simplicity. (a) In Phase I, $\|\widehat{\boldsymbol{x}}^{(t)} - \widetilde{\boldsymbol{x}}^{(t)}\|_2$ increases linearly and $\|\boldsymbol{x}^{(t)} - \widehat{\boldsymbol{x}}^{(t)}\|_2$ increases exponentially with the rate of $(1 + \eta\lambda^\star)$. They have the same scale at the end of Phase I. (b) In Phase I, even if the $\boldsymbol{u}^\star$ component of $\boldsymbol{x}^{(t)} - \boldsymbol{x}^{(t,l)}$ grows exponentially, it remains almost orthogonal to $\boldsymbol{u}^\star$ throughout the phase. (c) Phase II is divided into three parts according to the growth speed of $\boldsymbol{x}^{(t)}$ and $\boldsymbol{x}^{(t)} - \widetilde{\boldsymbol{x}}^{(t)}$. The ratio between them at the start and the end of each part is described, and it is at most $\frac{1}{\sqrt{\log n}}$ in Phase II.

Note that the norm of $\widetilde{\boldsymbol{x}}^{(t)}$ is used in the second term of (15). The update equation of $\widehat{\boldsymbol{x}}^{(t)}$ differs from $\widetilde{\boldsymbol{x}}^{(t)}$ in the third term and from $\boldsymbol{x}^{(t)}$ in the second term. We use $\widehat{\boldsymbol{x}}^{(t)}$ as a proxy for bounding $\left\|\boldsymbol{x}^{(t)} - \widetilde{\boldsymbol{x}}^{(t)}\right\|_2$. We first show that $\left\|\widehat{\boldsymbol{x}}^{(t)} - \widetilde{\boldsymbol{x}}^{(t)}\right\|_2$ grows at most linearly with respect to $t$.

**Lemma 5.2.** *With probability at least $1 - o(1/\sqrt{\log n})$, for all $t \le T_1$, we have*

$$\left\|\widehat{\boldsymbol{x}}^{(t)} - \widetilde{\boldsymbol{x}}^{(t)}\right\|_2 \lesssim \mu\sqrt{\frac{\log n}{np}}\beta_0 t. \tag{16}$$

The proof of this lemma is based on the fact that $\widehat{\boldsymbol{x}}^{(t)}$ is a product of $\boldsymbol{x}^{(0)}$ and a matrix polynomial of $\boldsymbol{I}$ and $\boldsymbol{M}^\circ$, while $\widetilde{\boldsymbol{x}}^{(t)}$ is a product between $\boldsymbol{x}^{(0)}$ and a matrix polynomial of $\boldsymbol{I}$ and $\boldsymbol{M}^\star$. We prove the lemma by comparing the two matrix polynomials. We remark that Lemma 5.2 holds regardless of the small initialization, but it relies on the randomness of $\boldsymbol{x}^{(0)}$.

Since $\boldsymbol{x}^{(t)}$ and $\widehat{\boldsymbol{x}}^{(t)}$ differ only in the second term, their initial difference is proportional to $\beta_0^3$. More precisely, it is $\frac{1}{\sqrt{np}}\beta_0^3$. We show that the difference grows exponentially at a rate of $(1 + \eta\lambda^\star)$.

**Lemma 5.3.** *If (14) holds for all $t \le T_1$, we have*

$$\left\|\boldsymbol{x}^{(t)} - \widehat{\boldsymbol{x}}^{(t)}\right\|_2 \lesssim \frac{1}{\lambda^\star}\sqrt{\frac{\mu^3 \log^3 n}{np}}(1 + \eta\lambda^\star)^t \beta_0^3 \tag{17}$$

*for all $t \le T_1$ with probability at least $1 - o(1/\sqrt{\log n})$.*

The upper bound in (17) becomes smaller than that of (16) if $(1 + \eta\lambda^\star)^t \beta_0^2 \le \lambda^\star\sqrt{\frac{1}{\mu \log^2 n}}$. One can check that this condition is satisfied from the definition of $T_1$ given in Lemma 5.1 and the bound on the initialization size (8). Thus, (13) is proved by (16) and (17).

**Proof of (14)**     We control the $l$th component of $\boldsymbol{x}^{(t)} - \widetilde{\boldsymbol{x}}^{(t)}$ using the $l$th leave-one-out sequence. Leave-one-out sequences have two important properties. First, because they are defined without only one row/column, they are extremely close to $\boldsymbol{x}^{(t)}$, and at $t = 1$, $\left\|\boldsymbol{x}^{(t)} - \boldsymbol{x}^{(t,l)}\right\|_2$ is about $\frac{1}{\sqrt{np}}\frac{\beta_0}{\sqrt{n}}$. Second, the $l$th component of the $l$th leave-one-out sequence evolves similarly to that of $\widetilde{\boldsymbol{x}}^{(t)}$ and is easy to analyze. With these two properties, we bound the $l$th component of $\boldsymbol{x}^{(t)} - \widetilde{\boldsymbol{x}}^{(t)}$ as

$$\left|\left(\boldsymbol{x}^{(t)} - \widetilde{\boldsymbol{x}}^{(t)}\right)_l\right| \le \left\|\boldsymbol{x}^{(t)} - \boldsymbol{x}^{(t,l)}\right\|_2 + \left|\left(\boldsymbol{x}^{(t,l)} - \widetilde{\boldsymbol{x}}^{(t)}\right)_l\right|. \tag{18}$$

We claim that both $\left\|\boldsymbol{x}^{(t)} - \boldsymbol{x}^{(t,l)}\right\|_2$ and $\left|\left(\boldsymbol{x}^{(t,l)} - \widetilde{\boldsymbol{x}}^{(t)}\right)_l\right|$ increase at most polynomially with respect to $t$ from the initial scale $\frac{1}{\sqrt{np}}\frac{\beta_0}{\sqrt{n}}$.

**Lemma 5.4.** *With probability at least $1 - o(1/\sqrt{\log n})$, for all $t \leq T_1$, we have*

$$\left\|\boldsymbol{x}^{(t)} - \boldsymbol{x}^{(t,l)}\right\|_2 \lesssim \mu\sqrt{\frac{\log^2 n}{np}}\frac{\beta_0}{\sqrt{n}}t, \quad (19) \qquad \left|\left(\boldsymbol{x}^{(t,l)} - \widetilde{\boldsymbol{x}}^{(t)}\right)_l\right| \lesssim \sqrt{\frac{\mu^3 \log^2 n}{np}}\frac{\beta_0}{\sqrt{n}}t^2. \quad (20)$$

As explained for $\boldsymbol{x}^{(t)} - \widetilde{\boldsymbol{x}}^{(t)}$, due to the third terms of (1) and (3), $\boldsymbol{x}^{(t)} - \boldsymbol{x}^{(t,l)}$ can also grow exponentially at the rate of $(1 + \eta\lambda^\star)$ in the worst case where $\boldsymbol{x}^{(t)} - \boldsymbol{x}^{(t,l)}$ is parallel to $\boldsymbol{u}^\star$. This contradicts our result (19) that $\left\|\boldsymbol{x}^{(t)} - \boldsymbol{x}^{(t,l)}\right\|_2$ grows only linearly. We show that $\boldsymbol{x}^{(t)} - \boldsymbol{x}^{(t,l)}$ remains nearly orthogonal to $\boldsymbol{u}^\star$ in Phase I, and thus the worst case does not occur.

**Lemma 5.5.** *For all $l \in [n]$ and $t \leq T_1$, we have*

$$\left|\boldsymbol{u}^{(l)\top}(\boldsymbol{x}^{(t)} - \boldsymbol{x}^{(t,l)})\right| \lesssim \sqrt{\frac{\mu^3 \log^2 n}{np}}(1 + \eta\lambda^\star)^t\frac{\beta_0}{n}$$

*with probability at least $1 - o(1/\sqrt{\log n})$, where $\boldsymbol{u}^{(l)}$ is the first eigenvector of $\boldsymbol{M}^{(l)}$.*

Note that $\boldsymbol{u}^{(l)}$ is almost parallel to $\boldsymbol{u}^\star$ (see Lemma A.5 in the appendix). The $\boldsymbol{u}^{(l)}$ component of $\boldsymbol{x}^{(t)} - \boldsymbol{x}^{(t,l)}$ is initialized to the order of $\frac{1}{\sqrt{np}}\frac{\beta_0}{n}$, which is $\frac{1}{\sqrt{n}}$ times smaller than $\left\|\boldsymbol{x}^{(t)} - \boldsymbol{x}^{(t,l)}\right\|_2$. Although it is increased exponentially, from the definition of $T_1$, the $\boldsymbol{u}^{(l)}$ component remains much smaller than $\left\|\boldsymbol{x}^{(t)} - \boldsymbol{x}^{(t,l)}\right\|_2$ in Phase I.

One can see that $\left|\left(\boldsymbol{x}^{(t,l)} - \widetilde{\boldsymbol{x}}^{(t)}\right)_l\right|$ increases by $\left\|\boldsymbol{x}^{(t)} - \widetilde{\boldsymbol{x}}^{(t)}\right\|_2\|\boldsymbol{u}^\star\|_\infty$ at each step, and summing the bound (13) up to $t$ gives (20). Finally, (14) is obtained by putting (19) and (20) into (18).

## 6 Phase II: Expansion

In the next phase, we show that the bounds obtained in Phase I are increased at a rate of $(1 + \eta\lambda^\star)$.

**Lemma 6.1.** *Let $T_2$ be the largest $t$ such that $\widetilde{\beta}_t^2 \leq \lambda^\star\left(1 - \frac{1}{\log n}\right)$. Then, for all $T_1 < t \leq T_2$, we have*

$$\left\|\boldsymbol{x}^{(t)} - \widetilde{\boldsymbol{x}}^{(t)}\right\|_2 \lesssim \mu\sqrt{\frac{\log^3 n}{np}}\beta_0(1 + \eta\lambda^\star)^{t-T_1}, \tag{21}$$

$$\left\|\boldsymbol{x}^{(t)} - \boldsymbol{x}^{(t,l)}\right\|_2 \lesssim \mu\sqrt{\frac{\log^5 n}{np}}\frac{\beta_0}{\sqrt{n}}(1 + \eta\lambda^\star)^{t-T_1}, \tag{22}$$

$$\left\|\boldsymbol{x}^{(t)} - \widetilde{\boldsymbol{x}}^{(t)}\right\|_\infty \lesssim \sqrt{\frac{\mu^3 \log^8 n}{np}}\frac{\beta_0}{\sqrt{n}}(1 + \eta\lambda^\star)^{t-T_1}, \tag{23}$$

$$\left|\left(\boldsymbol{x}^{(t,l)} - \widetilde{\boldsymbol{x}}^{(t)}\right)_l\right| \lesssim \sqrt{\frac{\mu^3 \log^8 n}{np}}\frac{\beta_0}{\sqrt{n}}(1 + \eta\lambda^\star)^{t-T_1}, \tag{24}$$

*with probability at least $1 - o(1/\sqrt{\log n})$.*

$T_2$ is defined as the end of Phase II. We will explain how Lemma 6.1 leads to Theorem 3.3 in Phase II. Let us first focus on (21) and (11). We can divide Phase II into three parts according to the behavior of $\left\|\widetilde{\boldsymbol{x}}^{(t)}\right\|_2$. First, $\left\|\widetilde{\boldsymbol{x}}^{(t)}\right\|_2$ is kept close to $\beta_0$ until $(1 + \eta\lambda^\star)^t$ becomes $\sqrt{n}$, or $(1 + \eta\lambda^\star)^{t-T_1}$ becomes $\sqrt{np}$. In this part, although the bounds increase exponentially with the rate of $(1 + \eta\lambda^\star)$, the factor $\frac{1}{\sqrt{np}}$, which was already present in (13) of Phase I, compensates for this increase. At the end of the first part, $\left\|\boldsymbol{x}^{(t)} - \widetilde{\boldsymbol{x}}^{(t)}\right\|_2$ is smaller than $\left\|\widetilde{\boldsymbol{x}}^{(t)}\right\|_2$ by some log factors. Next, $\left\|\widetilde{\boldsymbol{x}}^{(t)}\right\|_2$ grows at the rate of $(1 + \eta\lambda^\star)$ until it reaches $\frac{\sqrt{\lambda^\star}}{8\sqrt{\log n}}$. Since both $\left\|\boldsymbol{x}^{(t)} - \widetilde{\boldsymbol{x}}^{(t)}\right\|_2$ and $\left\|\widetilde{\boldsymbol{x}}^{(t)}\right\|_2$ increase with $(1 + \eta\lambda^\star)$,

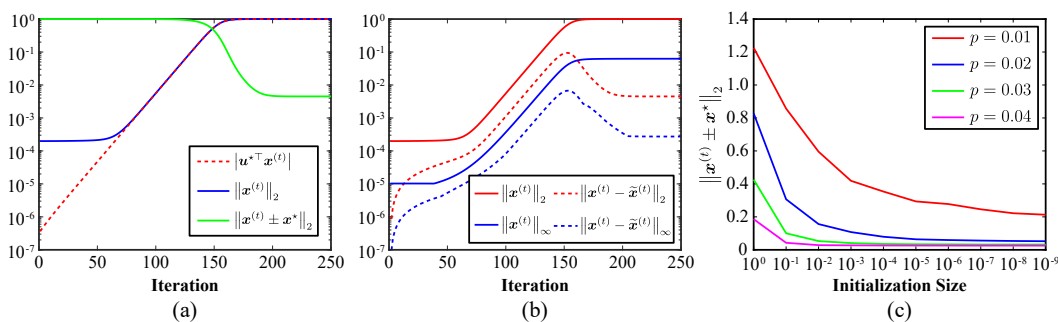

Figure 3: (a) Evolution of the quantities $|\boldsymbol{u}^{\star\top}\boldsymbol{x}^{(t)}|$ and $\|\boldsymbol{x}^{(t)}\|_2$, which behave similarly to $\widetilde{\alpha}_t$ and $\widetilde{\beta}_t$, respectively, and $\|\boldsymbol{x}^{(t)} \pm \boldsymbol{x}^\star\|_2$, which shows local convergence. (b) Comparison between the norms of $\boldsymbol{x}^{(t)}$ and $\boldsymbol{x}^{(t)} - \widetilde{\boldsymbol{x}}^{(t)}$. (c) Convergence of GD with respect to the initialization size and sampling probability. $\|\boldsymbol{x}^{(t)} \pm \boldsymbol{x}^\star\|_2$ was measured at $t = \frac{1}{\log(1+\eta\lambda^\star)} \log \frac{\sqrt{\lambda^\star n}}{\beta_0} + 100$ and averaged over 1000 trials.

the ratio between them is maintained in the second part. Finally, in the remaining iterations, $\left\|\widetilde{\boldsymbol{x}}^{(t)}\right\|_2$ increases with $(1 - \eta\widetilde{\beta}_t^2 + \eta\lambda^\star)$ at each step, and the increment becomes smaller as it converges to $\sqrt{\lambda^\star}$. Thus, as in the first part, $\left\|\boldsymbol{x}^{(t)} - \widetilde{\boldsymbol{x}}^{(t)}\right\|_2$ increases faster than $\left\|\widetilde{\boldsymbol{x}}^{(t)}\right\|_2$. However, from Lemma 4.1, the length of this part is $O(\log \log n)$, and the ratio between $\left\|\boldsymbol{x}^{(t)} - \widetilde{\boldsymbol{x}}^{(t)}\right\|_2$ and $\left\|\widetilde{\boldsymbol{x}}^{(t)}\right\|_2$ increases only by $\log^6 n$. We prove that the log factors already present at the end of the second part compensate this, and finally (11) holds for all $t$ in Phase II. A more delicate analysis may prove that $\left\|\boldsymbol{x}^{(t)} - \widetilde{\boldsymbol{x}}^{(t)}\right\|_2$ grows at the same rate as $\left\|\widetilde{\boldsymbol{x}}^{(t)}\right\|_2$ in the third part, and this will reduce the required sample complexity by at most $\log^{12} n$. A similar argument can be used to prove that the bounds for $\left\|\boldsymbol{x}^{(t)} - \widetilde{\boldsymbol{x}}^{(t)}\right\|_\infty$, $\left\|\boldsymbol{x}^{(t)} - \boldsymbol{x}^{(t,l)}\right\|_2$, and $\left|(\boldsymbol{x}^{(t,l)} - \widetilde{\boldsymbol{x}}^{(t)})_l\right|$ are smaller than $\left\|\widetilde{\boldsymbol{x}}^{(t)}\right\|_\infty$ by some log factors throughout Phase II.

At the end of Phase II, $\widetilde{\boldsymbol{x}}^{(t)}$ is very close to $\pm\boldsymbol{x}^\star$ in both $\ell_2$ and $\ell_\infty$ norms (see Corollary C.3 in the appendix), so one can replace $\widetilde{\boldsymbol{x}}^{(t)}$ of Lemma 6.1 with $\pm\boldsymbol{x}^\star$ to prove (4) to (7) of Theorem 3.1. Hence, we can let $T^\star = T_2$, and as explained in Section 4, $T_2$ is approximately given by $\frac{1}{\log(1+\eta\lambda^\star)} \log \frac{\sqrt{\lambda^\star n}}{\beta_0} + O(\log \log n)$.

## 7 Simulation

In this section, we present some simulation results that support our theoretical findings.

**Trajectory of GD** With the dimension $n = 5000$, we constructed the ground truth vector $\boldsymbol{u}^\star$ by sampling it from the Gaussian distribution $\mathcal{N}(\boldsymbol{0}, \frac{1}{n}\boldsymbol{I})$ and normalizing it to have unit norm. We let $\lambda^\star = 1$ so that the matrix $\boldsymbol{M}^\star$ is given by $\boldsymbol{u}^\star\boldsymbol{u}^{\star\top}$, and we randomly sampled the matrix symmetrically with a sampling rate of $p = 0.1$ and Gaussian noise of $\sigma = \frac{0.1}{n}$. The initialization size was set to $\beta_0 = \frac{1}{n}$ and a step size of 0.1 was used for GD. Figure 3 (a) and (b) represent one trial of the experiment, but similar graphs were obtained in each repetition of the experiment. The evolution of some important quantities such as $\left\|\boldsymbol{x}^{(t)}\right\|_2$ and $\left|\boldsymbol{u}^{\star\top}\boldsymbol{x}^{(t)}\right|$ is shown in Figure 3(a). As in the fully observed case, the signal component $\left|\boldsymbol{u}^{\star\top}\boldsymbol{x}^{(t)}\right|$ increases at the the rate of $(1 + \eta\lambda^\star)$ until it approaches $\sqrt{\lambda^\star}$, and a local convergence to $\boldsymbol{x}^\star$ occurs, where $\left\|\boldsymbol{x}^{(t)} - \boldsymbol{x}^\star\right\|_2$ decreases exponentially and saturates at the level determined by the noise size $\sigma$. In Figure 3(b), we describe the deviation of $\boldsymbol{x}^{(t)}$ from $\widetilde{\boldsymbol{x}}^{(t)}$ in both $\ell_2$ and $\ell_\infty$ norms. The solid lines represent the norms of $\boldsymbol{x}^{(t)}$ and the dotted lines represent those of $\boldsymbol{x}^{(t)} - \widetilde{\boldsymbol{x}}^{(t)}$. We can see that there is a gap between the solid and the dotted lines during the whole iterations. Thus, $\boldsymbol{x}^{(t)}$ stays close to the trajectory of the fully observed case, as we proved in Theorem 3.3.

**Small Initialization** In the next experiment, we investigated the importance of a small initialization for the convergence of GD. We used the same conditions as in the previous experiment except

$n = 500$. We measured $\left\| \boldsymbol{x}^{(t)} \pm \boldsymbol{x}^\star \right\|_2$ at $t = \frac{1}{\log(1+\eta\lambda^\star)} \log \frac{\sqrt{\lambda^\star n}}{\beta_0} + 100$ and averaged it over 1000 trials. We repeated the experiment while changing the initialization size from $10^0$ to $10^{-9}$ and the sampling probability from $0.01$ to $0.04$. The result is summarized in Figure 3(c). For all sampling probabilities, the small initialization improves the convergence of GD. Also, the performance starts to saturate at much larger initialization sizes as the sampling probability increases, and this is consistent with our finding (8) that a larger initialization is possible as more samples are available.

## 8  Discussion

In this paper, we showed that for rank-1 symmetric matrix completion with $\ell_2$ loss, GD can converge to the ground truth starting from a small random initialization. Ignoring log factors, the bound on the initialization size is $n^{-\frac{1}{4}}$ when the optimal $n \operatorname{poly}(\log n)$ samples are provided , and the bound becomes larger as more samples are provided. The result is interesting because the loss function does not have global benign geometry if no regularizer is applied. Our result does not use any explicit regularizer and relies only on the implicit regularizing effect of GD.

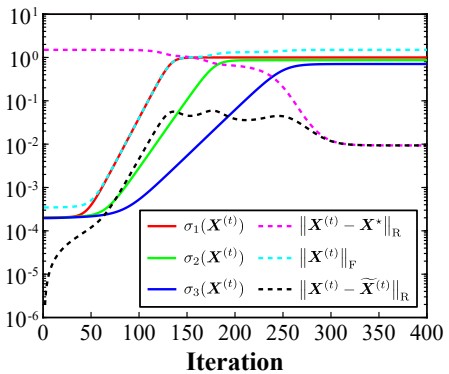

Figure 4: Trajectory of GD obtained for a rank-3 matrix with non-zero eigenvalues $1, 0.75, 0.5$. The same parameters were used as in Figure 3.

The most important future work is an extension to the rank-$r$ case. Suppose that $\boldsymbol{M}^\star$ is a rank-$r$ matrix and its eigendecomposition is given by $\boldsymbol{U}^\star \boldsymbol{\Sigma}^\star \boldsymbol{U}^{\star\top} = \boldsymbol{X}^\star \boldsymbol{X}^{\star\top}$, where $\boldsymbol{\Sigma}^\star = \operatorname{diag}(\lambda_1^\star, \cdots, \lambda_r^\star)$ and $\boldsymbol{X}^\star = \boldsymbol{U}^\star \boldsymbol{\Sigma}^{\star\frac{1}{2}}$. Then, the trajectory of GD becomes an $n \times r$ matrix $\boldsymbol{X}^{(t)}$, which is updated as

$$\boldsymbol{X}^{(t+1)} = \boldsymbol{X}^{(t)} - \frac{\eta}{p} \mathcal{P}_\Omega \left( \boldsymbol{X}^{(t)} \boldsymbol{X}^{(t)\top} \right) \boldsymbol{X}^{(t)} + \eta \boldsymbol{M}^\circ \boldsymbol{X}^{(t)}.$$

Each entry of $\boldsymbol{X}^{(0)}$ is sampled independently from the Gaussian distribution $\mathcal{N}\left(0, \frac{1}{n}\beta_0^2\right)$ as in the rank-1 case.

An instance of $\boldsymbol{X}^{(t)}$ is shown in Figure 4. The same conditions as in Figure 3 are used, except that the ground truth matrix is a rank-3 matrix with non-zero eigenvalues $1, 0.75, 0.5$. The singular values of $\boldsymbol{X}^{(t)}$ behave similarly to $\left\| \boldsymbol{x}^{(t)} \right\|_2$ in the rank-1 case. In the early iterations, where the orthogonal components dominate, the singular values stay close to their initial scale $\beta_0$. After that, each singular value $\sigma_i(\boldsymbol{X}^{(t)})$ increases at a rate of $(1 + \eta\lambda_i^\star)$ and saturates at $\sqrt{\lambda_i^\star}$. We use $\left\| \boldsymbol{X} - \boldsymbol{Y} \right\|_{\mathrm{R}}$ to denote the Frobenius norm between $\boldsymbol{X}$ and $\boldsymbol{Y}$ under best rotational alignment. $\left\| \boldsymbol{X}^{(t)} - \boldsymbol{X}^\star \right\|_{\mathrm{R}}$ decreases exponentially and saturates at the level determined by the noise size $\sigma$, after all singular values have saturated, as local convergence begins.

To extend the results of the rank-1 case, we need to show that $\left\| \boldsymbol{X}^{(t)} - \widetilde{\boldsymbol{X}}^{(t)} \right\|_{\mathrm{R}}$ remains much smaller than $\left\| \boldsymbol{X}^{(t)} \right\|_{\mathrm{F}}$ throughout the iterations, where $\widetilde{\boldsymbol{X}}^{(t)}$ is the trajectory of the fully observed case. Before $\sigma_1(\boldsymbol{X}^{(t)})$ saturates around $\sqrt{\lambda_1^\star}$, it behaves similarly to $\left\| \boldsymbol{x}^{(t)} - \widetilde{\boldsymbol{x}}^{(t)} \right\|_2$ of the rank-1 case, i.e., it expands at a rate of $(1 + \eta\lambda_1^\star)$ along with $\left\| \boldsymbol{X}^{(t)} \right\|_{\mathrm{F}}$ after the early iterations. However, because each singular value grows at a different rate, a different phenomenon is observed for the rank-$r$ case. During the iterations before $\sigma_{i+1}(\boldsymbol{X}^{(t)})$ saturates after $\sigma_i(\boldsymbol{X}^{(t)})$ does, both $\left\| \boldsymbol{X}^{(t)} - \widetilde{\boldsymbol{X}}^{(t)} \right\|_{\mathrm{R}}$ and $\left\| \boldsymbol{X}^{(t)} \right\|_{\mathrm{F}}$ do not increase much. Our current theory can only show that $\left\| \boldsymbol{X}^{(t)} - \widetilde{\boldsymbol{X}}^{(t)} \right\|_{\mathrm{R}}$ increases at a rate less than $(1 + \eta\lambda_{i+1}^\star)$, and in order for $\left\| \boldsymbol{X}^{(t)} - \widetilde{\boldsymbol{X}}^{(t)} \right\|_{\mathrm{R}}$ to remain much smaller than $\left\| \boldsymbol{X}^{(t)} \right\|_{\mathrm{F}}$, additional sample complexity is required to compensate for the exponential increases. Therefore, we expect that the convergence of GD for the case of rank-$r$ can be proved with the techniques developed in this paper if $n^{1+\Theta(\kappa-1)} \operatorname{poly}(\kappa, r, \log n)$ samples are provided, where $\kappa = \frac{\lambda_1^\star}{\lambda_r^\star}$ is the condition number. Nevertheless, whether GD can converge with the optimal $n \operatorname{poly}(\kappa, r, \log n)$ samples for the rank-$r$ matrix completion problem remains an open problem.

## Acknowledgments and Disclosure of Funding

This research was supported by the National Research Foundation of Korea under grant 2021R1C1C11008539.

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

Detailed proofs for the results explained in the main text are provided in this appendix. We say that an event happens *with high probability* if it happens with probability at least $1 - \frac{1}{n^C}$ for a constant $C > 0$ and $C$ can be made arbitrary large by controlling constant factors. A union of $\text{poly}(n)$ number of events that happens with high probability still happens with high probability. For a matrix $\boldsymbol{A}$, we denote the spectral norm by $\|\boldsymbol{A}\|$ and the maximum absolute value of entries by $\|\boldsymbol{A}\|_\infty$. Also, the largest $\ell_2$-norm of rows of $\boldsymbol{A}$ is denoted as $\|\boldsymbol{A}\|_{2,\infty}$.

## A   Spectral Analysis

We introduce some spectral bounds related to random sampling and Gaussian noise.

**Lemma A.1.** *If $n^2 p \gtrsim n \log n$, we have*

$$\left\| \frac{1}{p}\mathcal{P}_\Omega(\boldsymbol{M}^\star) - \boldsymbol{M}^\star \right\| \lesssim \lambda^\star \mu \sqrt{\frac{\log n}{np}}$$

*with high probability.*

**Lemma A.2.** *If $n^2 p \gtrsim \mu n \log n$, for all $l \in [n]$, we have*

$$\left\| \frac{1}{p}\mathcal{P}_\Omega(\boldsymbol{M}^\star) - \mathcal{P}_\Omega^{(l)}(\boldsymbol{M}^\star) \right\| \lesssim \lambda^\star \sqrt{\frac{\mu}{np}}$$

*with high probability.*

**Lemma A.3.** *If $n^2 p \gtrsim n \log^2 n$, we have*

$$\left\| \frac{1}{p}\mathcal{P}_\Omega(\boldsymbol{E}) \right\| \lesssim \sigma \sqrt{\frac{n}{p}}$$

*with high probability.*

Note that Lemma A.3 also implies that $\left\| \boldsymbol{E}^{(l)} \right\| \lesssim \sigma \sqrt{\frac{n}{p}}$ for all $l \in [n]$ with high probability. Combined with the condition $\sigma \lesssim \frac{\lambda^\star \mu}{n}\sqrt{\log n}$, we have

$$\left\| \frac{1}{p}\mathcal{P}_\Omega(\boldsymbol{E}) \right\| \lesssim \lambda^\star \mu \sqrt{\frac{\log n}{np}}, \quad \left\| \boldsymbol{E}^{(l)} \right\| \lesssim \lambda^\star \mu \sqrt{\frac{\log n}{np}}$$

for all $l \in [n]$. Proofs for Lemmas A.1 and A.2 are provided in Appendix G. Check Lemma 11 of [12] for the proof of Lemma A.3.

Next, we state bounds on the eigenvalues of $\boldsymbol{M}^\circ$ and $\boldsymbol{M}^{(l)}$. The first eigenvalues of $\boldsymbol{M}^\circ$ and $\boldsymbol{M}^{(l)}$ are denoted as $\lambda^\circ$ and $\lambda^{(l)}$, respectively. The following lemma is derived from Lemmas A.1 and A.2 with Weyl's Theorem.

**Lemma A.4.** *If $n^2 p \gtrsim \mu n \log n$, we have*

$$|\lambda^\circ - \lambda^\star| \lesssim \lambda^\star \mu \sqrt{\frac{\log n}{np}}, \tag{A.1}$$

$$\left| \lambda^{(l)} - \lambda^\star \right| \lesssim \lambda^\star \mu \sqrt{\frac{\log n}{np}} \tag{A.2}$$

*for all $l \in [n]$ with high probability.*

Lastly, Lemmas A.1 and A.2 with Davis-Kahan Theorem give the following lemma.

**Lemma A.5.** *If $n^2 p \gtrsim \mu n \log n$, we have*

$$\left\| \boldsymbol{u}^{(l)} - \boldsymbol{u}^\star \right\|_2 \lesssim \mu \sqrt{\frac{\log n}{np}}$$

*for all $l \in [n]$ with high probability.*

# B  Initialization

In this section, we introduce some properties that the initialization vector $x^{(0)}$ satisfies. Recall that each entry of $x^{(0)}$ is sampled from $\mathcal{N}(0, \frac{1}{n}\beta_0^2)$ independently. We use $H$ to denote the perturbation $M^\circ - M^\star$.

**Lemma B.1.** *The initialization vector $x^{(0)}$ satisfies*

$$\frac{1}{2}\beta_0 \le \left\|x^{(0)}\right\|_2 \le \frac{3}{2}\beta_0 \tag{B.1}$$

*with probability at least $1 - e^{-n/32}$, and*

$$\left\|x^{(0)}\right\|_\infty \le 2\sqrt{\log n}\frac{\beta_0}{\sqrt{n}}, \tag{B.2}$$

$$\left|u^{\star\top}H^s x^{(0)}\right| \le 2\sqrt{\log n}\frac{\beta_0}{\sqrt{n}}\|H\|^s, \quad \forall s \le 30\log n, \tag{B.3}$$

*with probability at least $1 - \frac{1}{n} - \frac{30\log n}{n^2}$. It also satisfies*

$$\frac{1}{\sqrt{\log n}}\frac{\beta_0}{\sqrt{n}} \le \left|u^{\star\top}x^{(0)}\right| \tag{B.4}$$

*with probability at least $1 - \frac{1}{2\sqrt{\log n}}$.*

*Proof.* To bound $\left\|x^{(0)}\right\|_2$, we use the following basic concentration inequality that holds for i.i.d. standard normal variables $\{X_i\}_{i\in[n]}$.

$$\mathbb{P}\left[\left|\frac{1}{n}\sum_{i=1}^n X_i^2 - 1\right| \ge t\right] \le 2e^{-nt^2/8}$$

If we put $t = \frac{1}{2}$, with probability at least $1 - e^{-n/32}$, we have

$$\frac{1}{2}\beta_0^2 \le \left\|x^{(0)}\right\|_2^2 \le \frac{3}{2}\beta_0^2,$$

and this implies (B.1).

For a centered Gaussian random variable with standard deviation $\sigma$, we have

$$\mathbb{P}[|X| \ge t] \le e^{-\frac{t^2}{2\sigma^2}}.$$

Hence, an entry of $x^{(0)}$ is less than $2\sqrt{\log n}\frac{\beta_0}{\sqrt{n}}$ with probability at least $1 - \frac{1}{n^2}$, and all entries of $x^{(0)}$ are less than $2\sqrt{\log n}\frac{\beta_0}{\sqrt{n}}$ with probability at least $1 - \frac{1}{n}$. $u^{\star\top}H^s x^{(0)}$ follows a centered Gaussian distribution with standard deviation $\|H^s u^\star\|_2 \le \|H\|^s\|u^\star\|_2$ for all $s$, and (B.3) holds with probability at least $1 - \frac{10\log n}{n^2}$.

For a random variable $X$ that is sampled from $\mathcal{N}(0, \sigma^2)$, we have

$$\mathbb{P}[|X| \le t] \le \frac{t}{\sqrt{2\pi\sigma^2}}.$$

Hence, we have

$$\frac{1}{\sqrt{\log n}}\frac{\beta_0}{\sqrt{n}} \le \left|u^{\star\top}x^{(0)}\right|$$

with probability at least $1 - \frac{1}{2\sqrt{\log n}}$. $\qquad\square$

Lemma B.1 implies that the $u^\star$ component of $x^{(0)}$ is in the range

$$\frac{1}{\sqrt{\log n}}\frac{\beta_0}{\sqrt{n}} \le \left|u^{\star\top}x^{(0)}\right| \le 2\sqrt{\log n}\frac{\beta_0}{\sqrt{n}}. \tag{B.5}$$

**Lemma B.2.** *We have*

$$\left\| a\boldsymbol{x}^{(0)} + b\boldsymbol{u}^\star \right\|_\infty \geq (|a|\beta_0 + |b|)\frac{1}{\sqrt{n}} \tag{B.6}$$

*for all $a, b$, with probability at least $1 - \exp\left(-\frac{n}{2\mu}\right)$.*

*Proof.* The probability that an entry of $\boldsymbol{x}^{(0)}$ is less than $\frac{\beta_0}{\sqrt{n}}$ is bounded by $\frac{1}{\sqrt{2\pi}}$. Without loss of generality, let us assume that all entries of $\boldsymbol{u}^\star$ are not negative and $a, b \geq 0$. There are at least $\frac{n}{\mu}$ entries of $\boldsymbol{u}^\star$ that are larger than $\frac{1}{\sqrt{n}}$. For such entries, the probability that all entries of $\boldsymbol{x}^{(0)}$ is less than $\frac{\beta_0}{\sqrt{n}}$ is bounded by $\left(\frac{1}{\sqrt{2\pi}}\right)^{\frac{n}{\mu}} \leq \exp\left(-\frac{n}{2\mu}\right)$. Hence, for at least one position, both entries of $\boldsymbol{x}^{(0)}$ and $\boldsymbol{u}^\star$ are larger than $\frac{\beta_0}{\sqrt{n}}$ and $\frac{1}{\sqrt{n}}$, respectively, with probability at least $1 - \exp\left(-\frac{n}{2\mu}\right)$. $\qquad\square$

In the following sections, we assume that we are given an initialization vector $\boldsymbol{x}^{(0)}$ that satisfies (B.1) to (B.6).

## C   Fully Observed Case

We provide some lemmas related to $\widetilde{\boldsymbol{x}}^{(t)}$ in this section. We first note that $\widetilde{\boldsymbol{x}}^{(t)}$ is explicitly written as

$$\widetilde{\boldsymbol{x}}^{(t)} = \prod_{s=1}^{t}(1 - \eta\widetilde{\beta}_s^2)\boldsymbol{x}^{(0)} + \prod_{s=1}^{t}(1 - \eta\widetilde{\beta}_s^2 + \eta\lambda^\star)(\boldsymbol{u}^{\star\top}\boldsymbol{x}^{(0)})\boldsymbol{u}^\star := A^{(t)}\boldsymbol{x}^{(0)} + B^{(t)}\boldsymbol{u}^\star. \tag{C.1}$$

Let us define $T_2'$ as the last $t$ such that $\widetilde{\beta}_t^2 \leq \frac{\lambda^\star}{64\log n}$. We claim that $T_2' \leq \frac{64\log n}{\eta\lambda^\star}$ and prove this later. Then, for all $t \leq T_2'$, we have

$$\frac{1}{4}(1 + \eta\lambda^\star)^t \leq \prod_{s=1}^{t}(1 - \eta\widetilde{\beta}_s^2 + \eta\lambda^\star) \leq (1 + \eta\lambda^\star)^t$$

$$\frac{1}{4} \leq \prod_{s=1}^{t}(1 - \eta\widetilde{\beta}_s^2) \leq 1 \tag{C.2}$$

because

$$\prod_{s=1}^{T_2'}\left(\frac{1 + \eta\lambda^\star - \eta\widetilde{\beta}_s^2}{1 + \eta\lambda^\star}\right) \geq \prod_{s=1}^{T_2'}(1 - \eta\widetilde{\beta}_s^2) \geq \left(1 - \frac{\eta\lambda^\star}{64\log n}\right)^{\frac{64\log n}{\eta\lambda^\star}} \geq \frac{1}{4}$$

if $\frac{\eta\lambda^\star}{64\log n} \leq \frac{1}{2}$. Note that the upper bounds in (C.2) hold even if $t > T_2$.

From (C.2), we have the approximation $\widetilde{\boldsymbol{x}}^{(t)} \approx \boldsymbol{x}^{(0)} + (1 + \eta\lambda^\star)^t(\boldsymbol{u}^{\star\top}\boldsymbol{x}^{(0)})\boldsymbol{u}^\star$ for all $t \leq T_2'$ and the $\ell_2$-norm of $\widetilde{\boldsymbol{x}}^{(t)}$ is also approximately given by $\left(1 + \frac{(1+\eta\lambda^\star)^t}{\sqrt{n}}\right)\beta_0$. The $\ell_\infty$-norm is about $\frac{1}{\sqrt{n}}$ times smaller than the $\ell_2$-norm. We make this observation rigorous with the following lemma.

**Lemma C.1.** *For all $t \leq T_2'$, we have*

$$\frac{1}{8}\frac{1}{\sqrt{\log n}}\left(1 + \frac{(1 + \eta\lambda^\star)^t}{\sqrt{n}}\right)\beta_0 \leq \left\|\widetilde{\boldsymbol{x}}^{(t)}\right\|_2 \leq 2\sqrt{\log n}\left(1 + \frac{(1 + \eta\lambda^\star)^t}{\sqrt{n}}\right)\beta_0,$$

$$\frac{1}{4}\frac{1}{\sqrt{\log n}}\left(1 + \frac{(1 + \eta\lambda^\star)^t}{\sqrt{n}}\right)\frac{\beta_0}{\sqrt{n}} \leq \left\|\widetilde{\boldsymbol{x}}^{(t)}\right\|_\infty \leq 2\sqrt{\log n}\left(1 + (1 + \eta\lambda^\star)^t\sqrt{\frac{\mu}{n}}\right)\frac{\beta_0}{\sqrt{n}}.$$

*Proof.* For brevity, les us drop the superscript $(t)$ and write $\widetilde{\boldsymbol{x}} = A\boldsymbol{x}^{(0)} + B\boldsymbol{u}^\star$. For the upper bounds, we may use the triangle inequality

$$\left\|A\boldsymbol{x}^{(0)} + B\boldsymbol{u}^\star\right\| \leq A\left\|\boldsymbol{x}^{(0)}\right\| + |B|\|\boldsymbol{u}^\star\|.$$

If we use (B.5) and (C.2), we get the upper bounds for $A$ and $B$. We have $\left\|\boldsymbol{x}^{(0)}\right\|_2 \leq 2\beta_0$ by (B.1), and the $\ell_\infty$-norm of $\boldsymbol{x}^{(0)}$ is controlled through (B.2). These finish the proof for the upper bounds.

From the definition of $B$, we have $B(\boldsymbol{u}^{\star\top}\boldsymbol{x}^{(0)}) \geq 0$, and

$$
\begin{aligned}
\left\|A\boldsymbol{x}^{(0)} + B\boldsymbol{u}^\star\right\|_2^2 &= A^2\left\|\boldsymbol{x}^{(0)}\right\|_2^2 + B^2 + 2AB(\boldsymbol{u}^{\star\top}\boldsymbol{x}^{(0)}) \\
&\geq A^2\left\|\boldsymbol{x}^{(0)}\right\|_2^2 + B^2 \\
&\geq \frac{1}{4}\left(A\left\|\boldsymbol{x}^{(0)}\right\|_2 + |B|\right)^2.
\end{aligned}
$$

(B.1) and (B.4) together with the lower bound in (C.2) give the desired lower bound for $\|\widetilde{\boldsymbol{x}}\|_2$. The lower bound for $\ell_\infty$-norm is directly implied from Lemma B.2 together with (B.4) and (C.2). $\qquad\square$

With Lemma C.1, we have

$$
\frac{1}{8}\frac{1}{\sqrt{\log n}}\frac{(1+\eta\lambda^\star)^{T_2'}}{\sqrt{n}}\beta_0 \leq \frac{1}{8}\frac{1}{\sqrt{\log n}}\left(1 + \frac{(1+\eta\lambda^\star)^{T_2'}}{\sqrt{n}}\right)\beta_0 \leq \left\|\widetilde{\boldsymbol{x}}^{(T_2')}\right\|_2 \leq \frac{\sqrt{\lambda^\star}}{8\sqrt{\log n}},
$$

and thus

$$
T_2' \leq \frac{1}{\log(1+\eta\lambda^\star)}\log\frac{\sqrt{\lambda^\star n}}{\beta_0} \leq \frac{11\log n}{\log(1+\eta\lambda^\star)} \leq \frac{64\log n}{\eta\lambda^\star}.
$$

For $t \leq T_1$ where $(1+\eta\lambda^\star)^t$ is not big, the bounds in Lemma C.1 are simplified to

$$
\frac{1}{\sqrt{\log n}}\beta_0 \lesssim \left\|\widetilde{\boldsymbol{x}}^{(t)}\right\|_2 \lesssim \sqrt{\log n}\,\beta_0, \tag{C.3}
$$

$$
\frac{1}{\sqrt{\log n}}\frac{\beta_0}{\sqrt{n}} \lesssim \left\|\widetilde{\boldsymbol{x}}^{(t)}\right\|_\infty \lesssim \sqrt{\log n}\frac{\beta_0}{\sqrt{n}}. \tag{C.4}
$$

After $\widetilde{\boldsymbol{x}}^{(t)}$ becomes almost parallel to $\boldsymbol{u}^\star$ and before $T_2'$, we could approximate $\widetilde{\beta}_t$ as increasing with the rate $(1+\eta\lambda^\star)$. However, after $T_2'$, this approximation is invalid, and $\widetilde{\beta}_t$ grows at a slower rate as it increases and it eventually converges to $\sqrt{\lambda^\star}$. How much iterations will be required for it to reach $\sqrt{\lambda^\star}\sqrt{1-\frac{1}{\log n}}$ after $T_2'$? With Lemma C.2, we will prove that $O(\log\log n)$ iterations are required after $T_2'$.

**Lemma C.2.** *At* $t = T_2' + \frac{6\log\log n}{\log(1+\eta\lambda^\star)}$, *we have* $\widetilde{\beta}_t^2 \geq \lambda^\star\left(1 - \frac{1}{\log n}\right)$.

*Proof.* From the decomposition (C.1), we have

$$
\left|\left\|\widetilde{\boldsymbol{x}}^{(t)}\right\|_2 - \left|B^{(t)}\right|\right| \leq \left\|\boldsymbol{x}^{(0)}\right\|_2
$$

$$
\left|\left|\boldsymbol{u}^{\star\top}\boldsymbol{x}^{(0)}\right| - \left|B^{(t)}\right|\right| \leq \left|A^{(t)}\right|\left|\boldsymbol{u}^{\star\top}\boldsymbol{x}^{(0)}\right| \leq \left\|\boldsymbol{x}^{(0)}\right\|_2,
$$

and thus

$$
\left|\widetilde{\alpha}_t - \widetilde{\beta}_t\right| \leq 2\left\|\boldsymbol{x}^{(0)}\right\|_2 \leq \frac{\sqrt{\lambda^\star}}{3\log^2 n}. \tag{C.5}
$$

holds for all $t$. Because $\widetilde{\beta}_t \gtrsim \frac{1}{\log n}$ for all $t \geq T_2'$, (C.5) implies that $\widetilde{\beta}_t$ is well approximated by $\widetilde{\alpha}_t$. Hence, we will focus on $\widetilde{\alpha}_t$, which is an increasing sequence that evolves with

$$
\widetilde{\alpha}_{t+1} = (1 - \eta\widetilde{\beta}_t^2 + \eta\lambda^\star)\widetilde{\alpha}_t.
$$

For all $i \geq 1$, let $N_i$ be the last $t$ such that $\lambda^\star - \widetilde{\alpha}_t^2 \geq \frac{\lambda^\star}{e^i}$. Then, we have

$$
\lambda^\star - \widetilde{\alpha}_{N_i}^2 \geq \frac{\lambda^\star}{e^i} > \lambda^\star - \widetilde{\alpha}_{N_i+1}^2. \tag{C.6}
$$

Let $i \geq 2$. For all $N_{i-1} < t \leq N_i$,

$$\frac{\widetilde{\alpha}_{t+1}}{\widetilde{\alpha}_t} = 1 - \eta\widetilde{\beta}_t^2 + \eta\lambda^\star = 1 + \eta(\lambda^\star - \widetilde{\alpha}_t^2) + \eta(\widetilde{\alpha}_t^2 - \widetilde{\beta}_t^2) \geq 1 + \frac{\eta\lambda^\star}{e^i} - \frac{\eta\lambda^\star}{\log^2 n} \geq 1 + 0.99\frac{\eta\lambda^\star}{e^i}.$$

We used (C.5), (C.6), and the fact that $\widetilde{\alpha}_t, \widetilde{\beta}_t \leq \sqrt{\lambda^\star}$ for all $t$. This implies

$$\left(1 + 0.99\frac{\eta\lambda^\star}{e^i}\right)^{N_i - N_{i-1} - 1} x_{N_{i-1}+1} \leq x_{N_i}.$$

From the lower and upper bounds provided by (C.6), we have

$$\sqrt{\lambda^\star}\sqrt{1 - \frac{1}{e^{i-1}}}\left(1 + 0.99\frac{\eta\lambda^\star}{e^i}\right)^{N_i - N_{i-1} - 1} \leq \sqrt{\lambda^\star}\sqrt{1 - \frac{1}{e^i}},$$

$$\left(1 + 0.99\frac{\eta\lambda^\star}{e^i}\right)^{2(N_i - N_{i-1} - 1)} \leq \frac{e^i - 1}{e^i}\frac{e^{i-1}}{e^{i-1} - 1} = \frac{e^{i-1} - \frac{1}{e}}{e^{i-1} - 1} \leq 1 + \frac{1}{e^{i-1}}.$$

Taking log on both sides and using the inequality $\frac{1}{2}x < \log(1 + x) < x$ that holds for $0 < x < 1$, we get

$$N_i - N_{i-1} \leq 1 + \frac{1}{2}\frac{\log\left(1 + \frac{1}{e^{i-1}}\right)}{\log\left(1 + 0.99\frac{\eta\lambda^\star}{e^i}\right)} \leq 1 + \frac{e}{0.99\eta\lambda^\star}.$$

For $t \leq N_1$, we have

$$\frac{\widetilde{\alpha}_{t+1}}{\widetilde{\alpha}_t} \geq 1 + 0.99\frac{\eta\lambda^\star}{e},$$

and thus

$$\sqrt{\lambda^\star}\sqrt{1 - \frac{1}{e}} \geq \alpha_{N_1} \geq \left(1 + 0.99\frac{\eta\lambda^\star}{e}\right)^{N_1}\widetilde{\alpha}_{T_2'} = \left(1 + 0.99\frac{\eta\lambda^\star}{e}\right)^{N_1}\sqrt{\frac{\lambda^\star}{21\log n}}.$$

Taking log on both sides we get

$$N_1 \leq 3\frac{\log\log n}{\eta\lambda^\star}.$$

Hence, we have

$$N_{\log\log n + 1} + 1 \leq \left(1 + \frac{e}{0.99\eta\lambda^\star}\right)\log\log n + N_1 + 1$$

$$\leq \left(1 + \frac{e}{0.99\eta\lambda^\star}\right)\log\log n + 3\frac{\log\log n}{\eta\lambda^\star} + 1$$

$$\leq \frac{6\log\log n}{\log(1 + \eta\lambda^\star)},$$

but at $t = N_{\log\log n + 1} + 1$, it holds that

$$\widetilde{\alpha}_t^2 > \lambda^\star\left(1 - \frac{1}{e\log n}\right),$$

and we have

$$\widetilde{\beta}_t^2 > \lambda^\star\left(1 - \frac{1}{\log n}\right)$$

as desired. Note that $\widetilde{\beta}_t$ is also an increasing sequence as $\widetilde{\alpha}_t$. $\qquad\square$

It is implied from Lemma C.2 that $T_2 \leq \frac{1}{\log(1 + \eta\lambda^\star)}\log\frac{\sqrt{\lambda^\star n}}{\beta_0} + \frac{6\log\log n}{\log(1 + \eta\lambda^\star)} = (1 + o(1))\frac{1}{\eta\lambda^\star}\log\frac{\sqrt{\lambda^\star n}}{\beta_0}$. The following corollary shows that $\widetilde{x}^{(t)}$ is sufficiently close to $x^\star$ at $t = T_2$.

**Corollary C.3.** *At* $t = T_2$*, we have*

$$\min\left\{\left\|\widetilde{x}^{(t)} - x^\star\right\|_2, \left\|\widetilde{x}^{(t)} + x^\star\right\|_2\right\} \lesssim \frac{1}{\sqrt{\log n}}\|x^\star\|_2, \tag{C.7}$$

$$\min\left\{\left\|\widetilde{x}^{(t)} - x^\star\right\|_\infty, \left\|\widetilde{x}^{(t)} + x^\star\right\|_\infty\right\} \lesssim \frac{1}{\sqrt{\log n}}\|x^\star\|_\infty. \tag{C.8}$$

*Proof.* When $B^{(t)} > 0$, from the decomposition

$$\boldsymbol{x}^{(t)} - \boldsymbol{x}^\star = A^{(t)}\boldsymbol{x}^{(0)} + (B^{(t)} - \widetilde{\beta}_t)\boldsymbol{u}^\star + (\widetilde{\beta}_t - \sqrt{\lambda^\star})\boldsymbol{u}^\star,$$

we have

$$\left\|\boldsymbol{x}^{(t)} - \boldsymbol{x}^\star\right\|_2 \le 2\left\|\boldsymbol{x}^{(0)}\right\|_2 + \frac{\sqrt{\lambda^\star}}{\sqrt{\log n}} \le \frac{2\sqrt{\lambda^\star}}{\sqrt{\log n}} = \frac{2}{\sqrt{\log n}}\|\boldsymbol{x}^\star\|_2.$$

For the cases $B^{(t)} < 0$ and $\ell_\infty$-norm, we may use similar technique. $\qquad\square$

# D  Phase I

## D.1  Proof of Lemma 5.2

In this subsection, we provide a proof to the following lemma, which is a formal statement of Lemma 5.2.

**Lemma D.1.** *With high probability, there exists a universal constant $c_0 > 0$ such that*

$$\left\|\widehat{\boldsymbol{x}}^{(t)} - \widetilde{\boldsymbol{x}}^{(t)}\right\|_2 \le c_0\mu\sqrt{\frac{\log n}{np}}\beta_0 t \tag{D.1}$$

*for all $t \le T_1$, if $n^2 p \gtrsim \mu^4 n \log^{21} n$ and the initialization point $\boldsymbol{x}^{(0)}$ satisfies (B.1) to (B.6).*

*Proof.* Let us rewrite the update equations (2) and (15) as

$$\widetilde{\boldsymbol{x}}^{(t+1)} = \left(\boldsymbol{I} - \eta\widetilde{\beta}_t^2 + \eta\boldsymbol{M}^\star\right)\widetilde{\boldsymbol{x}}^{(t)},$$

$$\widehat{\boldsymbol{x}}^{(t+1)} = \left(\boldsymbol{I} - \eta\widetilde{\beta}_t^2 + \eta\boldsymbol{M}^\circ\right)\widehat{\boldsymbol{x}}^{(t)},$$

where $\widetilde{\beta}_t = \left\|\widetilde{\boldsymbol{x}}^{(t)}\right\|_2^2$. Then, $\widehat{\boldsymbol{x}}^{(t)} - \widetilde{\boldsymbol{x}}^{(t)}$ is a product between $\boldsymbol{x}^{(0)}$ and $P^{(t)}(\boldsymbol{I}, \boldsymbol{M}^\star, \boldsymbol{H})$, which is a matrix polynomial of $\boldsymbol{I}, \boldsymbol{M}^\star, \boldsymbol{H}$, where $\boldsymbol{H} = \boldsymbol{M}^\circ - \boldsymbol{M}^\star$.

$$P^{(t)}(\boldsymbol{I}, \boldsymbol{M}^\star, \boldsymbol{H}) := \left(\prod_{s=1}^{t}\left((1 - \eta\widetilde{\beta}_s^2)\boldsymbol{I} + \eta\boldsymbol{M}^\star + \eta\boldsymbol{H}\right) - \prod_{s=1}^{t}\left((1 - \eta\widetilde{\beta}_s^2)\boldsymbol{I} + \eta\boldsymbol{M}^\star\right)\right) \tag{D.2}$$

$$\widehat{\boldsymbol{x}}^{(t)} - \widetilde{\boldsymbol{x}}^{(t)} = P^{(t)}(\boldsymbol{I}, \boldsymbol{M}^\star, \boldsymbol{H})\boldsymbol{x}^{(0)} \tag{D.3}$$

We classify the terms that appear after expanding the matrix polynomial $P^{(t)}(\boldsymbol{I}, \boldsymbol{M}^\star, \boldsymbol{H})$ into two types; 1) the terms that contain $\boldsymbol{H}$ but not $\boldsymbol{M}^\star$, 2) the terms that contain both $\boldsymbol{H}$ and $\boldsymbol{M}^\star$. We define $P_1^{(t)}(\boldsymbol{I}, \boldsymbol{H})$ to be a matrix polynomial of $\boldsymbol{I}$ and $\boldsymbol{H}$, which is equal to summation of the first type, and it is explicitly written as

$$P_1^{(t)}(\boldsymbol{I}, \boldsymbol{H}) = \prod_{s=1}^{t}\left((1 - \eta\widetilde{\beta}_s^2)\boldsymbol{I} + \eta\boldsymbol{H}\right) - \prod_{s=1}^{t}(1 - \eta\widetilde{\beta}_s^2)\boldsymbol{I}.$$

We correspondingly define $P_2^{(t)}(\boldsymbol{I}, \boldsymbol{M}^\star, \boldsymbol{H})$ to be summation of the second type, and it is equal to

$$P_2^{(t)}(\boldsymbol{I}, \boldsymbol{M}^\star, \boldsymbol{H}) = P^{(t)}(\boldsymbol{I}, \boldsymbol{M}^\star, \boldsymbol{H}) - P_1^{(t)}(\boldsymbol{I}, \boldsymbol{H}).$$

For $x, y \in \mathbb{R}$, we define $P_1^{(t)}(x, y)$ as the value that is obtained by substituting $x, y$ instead of $\boldsymbol{I}, \boldsymbol{H}$, respectively. For example, $P_1^{(t)}(1, 2) = \prod_{s=1}^{t}(1 - \eta\widetilde{\beta}_s^2 + 2\eta) - \prod_{s=1}^{t}(1 - \eta\widetilde{\beta}_s^2)$. For $x, y, z \in \mathbb{R}$, $P_2^{(t)}(x, y, z)$ is defined in a similar manner.

We bound the contribution of each type separately because the triangle inequality gives

$$\left\|\widehat{\boldsymbol{x}}^{(t)} - \widetilde{\boldsymbol{x}}^{(t)}\right\|_2 \le \left\|P_1^{(t)}(\boldsymbol{I}, \boldsymbol{H})\boldsymbol{x}^{(0)}\right\|_2 + \left\|P_2^{(t)}(\boldsymbol{I}, \boldsymbol{M}^\star, \boldsymbol{H})\boldsymbol{x}^{(0)}\right\|_2.$$

Every term in $P_1^{(t)}(\boldsymbol{I}, \boldsymbol{H})$ is $\boldsymbol{H}^s$ times a constant. We have $\left\|\boldsymbol{H}^s\boldsymbol{x}^{(0)}\right\|_2 \le \|\boldsymbol{H}\|^s\left\|\boldsymbol{x}^{(0)}\right\|_2$, and hence with triangle inequality

$$\left\|P_1^{(t)}(\boldsymbol{I}, \boldsymbol{H})\boldsymbol{x}^{(0)}\right\|_2 \le P_1^{(t)}(1, \|\boldsymbol{H}\|)\beta_0.$$

If $n^2p \gtrsim \mu^2 n \log^3 n$, we can further bound $P_1^{(t)}(1, \|\boldsymbol{H}\|)$ as

$$
\begin{aligned}
P_1^{(t)}(1, \|\boldsymbol{H}\|) &= \prod_{s=1}^{t}(1 - \eta\widetilde{\beta}_s^2 + \eta\|\boldsymbol{H}\|) - \prod_{s=1}^{t}(1 - \eta\widetilde{\beta}_s^2) \\
&= \prod_{s=1}^{t}(1 - \eta\widetilde{\beta}_s^2)\left(\prod_{s=1}^{t}\left(1 + \frac{\eta\|\boldsymbol{H}\|}{1 - \eta\widetilde{\beta}_s^2}\right) - 1\right) \\
&\leq \left(1 + \frac{\eta}{1 - \eta\lambda^{\star}}\|\boldsymbol{H}\|\right)^t - 1 \\
&\leq \left(\exp\left(\frac{\eta}{1 - \eta\lambda^{\star}}\|\boldsymbol{H}\|t\right) - 1\right) \\
&\leq \frac{2\eta}{1 - \eta\lambda^{\star}}\|\boldsymbol{H}\|t
\end{aligned}
$$

The third line uses the fact that $\widetilde{\beta}_t^2 \leq \lambda^{\star}$ for all $t \leq T_1$. The fourth and fifth lines are derived from an elementary inequality $1 + x \leq e^x \leq 1 + 2x$, which holds for small $x > 0$. Note that $\eta\|\boldsymbol{H}\|t \lesssim \eta\lambda^{\star}\mu\sqrt{\frac{\log n}{np}} \ll 1$ from Lemmas A.1 and A.3, and the fact that $T_1 \lesssim \log n$.

We can decompose every term of second type as a product of $\eta$, $\lambda^{\star}$, $\boldsymbol{u}^{\star}$, $\boldsymbol{H}^s\boldsymbol{u}^{\star}$, $\boldsymbol{u}^{\star\top}\boldsymbol{H}^s\boldsymbol{x}^{(0)}$, $\boldsymbol{u}^{\star\top}\boldsymbol{H}^s\boldsymbol{u}^{\star}$, and $\boldsymbol{u}^{\star\top}\boldsymbol{x}^{(0)}$. We describe this with some examples.

$$
\begin{aligned}
(\eta\boldsymbol{H})^{s_1}(\eta\boldsymbol{M}^{\star})(\eta\boldsymbol{H})^{s_2}\boldsymbol{x}^{(0)} &= \eta^{s_1+s_2+1}\lambda^{\star}(\boldsymbol{H}^{s_1}\boldsymbol{u}^{\star})(\boldsymbol{u}^{\star\top}\boldsymbol{H}^{s_2}\boldsymbol{x}^{(0)}) \\
(\eta\boldsymbol{H})^s(\eta\boldsymbol{M}^{\star})\boldsymbol{x}^{(0)} &= \eta^{s+1}(\boldsymbol{H}^s\boldsymbol{u}^{\star})(\boldsymbol{u}^{\star\top}\boldsymbol{x}^{(0)}) \\
(\eta\boldsymbol{M}^{\star})(\eta\boldsymbol{H})^s(\eta\boldsymbol{M}^{\star})\boldsymbol{x}^{(0)} &= \eta^{s+2}\lambda^{\star 2}\boldsymbol{u}^{\star}(\boldsymbol{u}^{\star\top}\boldsymbol{H}\boldsymbol{u}^{\star})(\boldsymbol{u}^{\star\top}\boldsymbol{x}^{(0)}) \\
(\eta\boldsymbol{M}^{\star})(\eta\boldsymbol{H})^s\boldsymbol{x}^{(0)} &= \eta^{s+1}\lambda^{\star}\boldsymbol{u}^{\star}(\boldsymbol{u}^{\star\top}\boldsymbol{H}^s\boldsymbol{x}^{(0)})
\end{aligned}
$$

The terms $\boldsymbol{H}^s\boldsymbol{u}^{\star}$ and $\boldsymbol{u}^{\star\top}\boldsymbol{H}^s\boldsymbol{u}^{\star}$ are bounded with

$$
\|\boldsymbol{H}^s\boldsymbol{u}^{\star}\|_2 \leq \|\boldsymbol{H}\|^s, \quad \left|\boldsymbol{u}^{\star\top}\boldsymbol{H}^s\boldsymbol{u}^{\star}\right| \leq \|\boldsymbol{H}\|^s, \tag{D.4}
$$

and the terms that contain $\boldsymbol{x}^{(0)}$ are bounded with (B.3). For every term of second type that includes $s_1$ times of $\eta\boldsymbol{M}^{\star}$ and $s_2$ times of $\eta\boldsymbol{H}$, the bounds (D.4) and (B.3) imply that $\ell_2$-norm of the term multiplied by $\boldsymbol{x}^{(0)}$ is at most

$$
(\eta\lambda^{\star})^{s_1}(\eta\|\boldsymbol{H}\|)^{s_2}2\sqrt{\frac{\log n}{n}}\beta_0.
$$

Hence, similar to the first type, we have

$$
\left\|P_2^{(t)}(\boldsymbol{I}, \boldsymbol{M}^{\star}, \boldsymbol{H})\boldsymbol{x}^{(0)}\right\|_2 \leq P_2^{(t)}(1, \lambda^{\star}, \|\boldsymbol{H}\|)2\sqrt{\frac{\log n}{n}}\beta_0.
$$

If $n^2p \gtrsim \mu^2 n \log^3 n$, we can further bound $P_2^{(t)}(1, \lambda^{\star}, \|\boldsymbol{H}\|)$ as

$$
\begin{aligned}
P_2^{(t)}(1, \lambda^{\star}, \|\boldsymbol{H}\|) &= \prod_{s=1}^{t}(1 - \eta\beta_s^2 + \eta\lambda^{\star} + \eta\|\boldsymbol{H}\|) - \prod_{s=1}^{t}\left(1 - \eta\beta_s^2 + \eta\lambda^{\star}\right) - P_1^{(t)}(1, \|\boldsymbol{H}\|) \\
&\leq \prod_{s=1}^{t}(1 - \eta\beta_s^2 + \eta\lambda^{\star} + \eta\|\boldsymbol{H}\|) - \prod_{s=1}^{t}\left(1 - \eta\beta_s^2 + \eta\lambda^{\star}\right) \\
&\leq \left(\prod_{s=1}^{t}(1 - \eta\beta_s^2 + \eta\lambda^{\star})\right)\left(\prod_{s=1}^{t}\left(1 + \frac{\eta}{1 - \eta\beta_s^2 + \eta\lambda^{\star}}\|\boldsymbol{H}\|\right) - 1\right) \\
&\leq (1 + \eta\lambda^{\star})^t\left((1 + \eta\|\boldsymbol{H}\|)^t - 1\right) \\
&\leq 2\eta\|\boldsymbol{H}\|t(1 + \eta\lambda^{\star})^t.
\end{aligned}
$$

Combining all, we have

$$\left\|\widehat{\boldsymbol{x}}^{(t)} - \widetilde{\boldsymbol{x}}^{(t)}\right\|_2 \le 4\eta\|\boldsymbol{H}\|t\left(1 + \sqrt{\frac{\log n}{n}}(1 + \eta\lambda^\star)^t\right)\beta_0$$

$$\le 4\eta\|\boldsymbol{H}\|t\left(1 + \sqrt{\frac{\log n}{n}}(1 + \eta\lambda^\star)^{T_1}\right)\beta_0$$

$$\le 4\eta\|\boldsymbol{H}\|t\left(1 + \sqrt{\frac{\mu^4\log^{22}n}{np}}\right)\beta_0$$

$$\le c_0\mu\sqrt{\frac{\log n}{np}}\beta_0 t$$

for all $t \le T_1$ for some constant $c_0 > 0$ if $n^2p \gtrsim \mu^4 n\log^{22}n$. $\qquad\square$

## D.2 Proof of Lemmas 5.3 to 5.5

We prove Lemmas 5.3 to 5.5 all together in an inductive manner.

**Lemma D.2.** *Suppose that the initialization point $\boldsymbol{x}^{(0)}$ satisfies (B.1) to (B.6). If $n^2p \gtrsim \mu^5 n\log^{22}n$, for all $t \le T_1$, we have*

$$\left\|\boldsymbol{x}^{(t)} - \widetilde{\boldsymbol{x}}^{(t)}\right\|_2 \le 2c_0\mu\sqrt{\frac{\log n}{np}}\beta_0 t, \tag{D.5}$$

$$\left\|\boldsymbol{x}^{(t)} - \widetilde{\boldsymbol{x}}^{(t)}\right\|_\infty \le (3c_0 + c_5)\sqrt{\frac{\mu^3\log^2 n}{np}}\frac{\beta_0}{\sqrt{n}}t^2, \tag{D.6}$$

$$\left\|\boldsymbol{x}^{(t)} - \widehat{\boldsymbol{x}}^{(t)}\right\|_2 \le c_2(3c_0 + c_5 + 1)\frac{1}{\lambda^\star}\sqrt{\frac{\mu^3\log^3 n}{np}}(1 + \eta\lambda^\star)^t\beta_0^3, \tag{D.7}$$

$$\left\|\boldsymbol{x}^{(t)} - \boldsymbol{x}^{(t,l)}\right\|_2 \le c_5\mu\sqrt{\frac{\log^2 n}{np}}\frac{\beta_0}{\sqrt{n}}t, \tag{D.8}$$

$$\left|\boldsymbol{u}^{(l)\top}(\boldsymbol{x}^{(t)} - \boldsymbol{x}^{(t,l)})\right| \le c_6\sqrt{\frac{\mu^3\log^2 n}{np}}(1 + \eta\lambda^\star)^t\frac{\beta_0}{n}, \tag{D.9}$$

$$\left|\left(\boldsymbol{x}^{(t,l)} - \widetilde{\boldsymbol{x}}^{(t)}\right)_l\right| \le 3c_0\sqrt{\frac{\mu^3\log^2 n}{np}}\frac{\beta_0}{\sqrt{n}}t^2, \tag{D.10}$$

*with high probability, where $c_2, c_5, c_6$ are positive constants.*

Before we start the proof, we introduce some notations. For $\boldsymbol{x} \in \mathbb{R}^n$, let us define

$$\|\boldsymbol{x}\|_{2,i} = \sqrt{\frac{1}{p}\sum_{j=1}^n \delta_{ij}x_j^2}, \quad \boldsymbol{I}_{\boldsymbol{x}} = \frac{1}{\|\boldsymbol{x}\|_2^2}\operatorname{diag}(\|\boldsymbol{x}\|_{2,1}^2, \cdots, \|\boldsymbol{x}\|_{2,n}^2).$$

$\|\boldsymbol{x}\|_{2,i}$ is the $\ell_2$-norm of $\boldsymbol{x}$ estimated with sampling of the $i$th row. With this notation, we can write the gradient of $f$ as

$$\nabla f(\boldsymbol{x}) = \|\boldsymbol{x}\|_2^2\boldsymbol{I}_{\boldsymbol{x}}\boldsymbol{x} - \boldsymbol{M}^\circ\boldsymbol{x}.$$

The function $g$ is defined as

$$g(\boldsymbol{x}) = \frac{1}{4p}\left\|\mathcal{P}_\Omega(\boldsymbol{x}\boldsymbol{x}^\top)\right\|_{\mathrm{F}}^2,$$

and its gradient satisfies

$$\nabla f(\boldsymbol{x}) = \nabla g(\boldsymbol{x}) - \boldsymbol{M}^\circ\boldsymbol{x}.$$

The Hessian of $g(\boldsymbol{x})$ is equal to

$$\nabla^2 g(\boldsymbol{x}) = \|\boldsymbol{x}\|_2^2 \boldsymbol{I_x} + \frac{2}{p}\mathcal{P}_\Omega(\boldsymbol{x}\boldsymbol{x}^\top).$$

The base case $t = 0$ for induction hypotheses (D.5) to (D.10) trivially hold because all three sequences $\boldsymbol{x}^{(t)}$, $\widehat{\boldsymbol{x}}^{(t)}$, $\widetilde{\boldsymbol{x}}^{(t)}$ start from the same point. Now, we assume that the hypotheses hold up to the $t$th iteration and show that they hold at the $(t+1)$st iteration. For brevity, we drop the superscript $(t)$ from $\boldsymbol{x}^{(t)}$, $\boldsymbol{x}^{(t,l)}$, $\widehat{\boldsymbol{x}}^{(t)}$, $\widetilde{\boldsymbol{x}}^{(t)}$ and denote them as $\boldsymbol{x}$, $\boldsymbol{x}^{(l)}$, $\widehat{\boldsymbol{x}}$, $\widetilde{\boldsymbol{x}}$, respectively. Also, recall that $T_1$ is defined to be the last $t$ such that $(1 + \eta\lambda^\star)^t \le \sqrt{\frac{\mu^4 \log^{21} n}{np}}\sqrt{n}$, and the magnitude of initialization satisfies $\beta_0^2 \lesssim \lambda^\star \sqrt{\frac{np}{\mu^5 \log^{26} n}}\frac{1}{\sqrt{n}}$ so that there exists a constant $c_1 > 0$ such that $(1 + \eta\lambda^\star)^t \beta_0^2 \le c_1 \frac{\lambda^\star}{\sqrt{\mu \log^5 n}}$.

**(D.7) at $(t+1)$**   We first decompose $\boldsymbol{x}^{(t+1)} - \widehat{\boldsymbol{x}}^{(t+1)}$ as

$$\boldsymbol{x}^{(t+1)} - \widehat{\boldsymbol{x}}^{(t+1)} = (\boldsymbol{I} + \eta\boldsymbol{M}^\circ)(\boldsymbol{x} - \widehat{\boldsymbol{x}}) - \eta\|\boldsymbol{x}\|_2^2 \boldsymbol{I_x}\boldsymbol{x} + \eta\|\widetilde{\boldsymbol{x}}\|_2^2 \widehat{\boldsymbol{x}}$$

$$= \left(\boldsymbol{I} - \eta\|\widetilde{\boldsymbol{x}}\|_2^2 \boldsymbol{I} + \eta\boldsymbol{M}^\circ\right)(\boldsymbol{x} - \widehat{\boldsymbol{x}}) - \eta\left(\|\boldsymbol{x}\|_2^2 \boldsymbol{I_x} - \|\widetilde{\boldsymbol{x}}\|_2^2 \boldsymbol{I}\right)\boldsymbol{x}. \qquad \text{(D.11)}$$

With the help of Lemma G.8, we bound the maximum entry of a diagonal matrix $\|\boldsymbol{x}\|_2^2 \boldsymbol{I_x} - \|\widetilde{\boldsymbol{x}}\|_2^2 \boldsymbol{I}$. We have

$$\max_{i\in[n]}\left|\|\boldsymbol{x}\|_{2,i}^2 - \|\widetilde{\boldsymbol{x}}\|_2^2\right| \lesssim n\|\boldsymbol{x} - \widetilde{\boldsymbol{x}}\|_\infty\|\boldsymbol{x} + \widetilde{\boldsymbol{x}}\|_\infty + \sqrt{\frac{\log n}{p}}\|\widetilde{\boldsymbol{x}}\|_2\|\widetilde{\boldsymbol{x}}\|_\infty + \frac{\log n}{p}\|\widetilde{\boldsymbol{x}}\|_\infty^2$$

$$\lesssim (3c_0 + c_5)\sqrt{\frac{\mu^3 \log^3 n}{np}}\beta_0^2 t^2 + \sqrt{\frac{\log^2 n}{np}}\beta_0^2 + \frac{\log^2 n}{np}\beta_0^2$$

$$\lesssim \sqrt{\frac{\mu^3 \log^3 n}{np}}\beta_0^2((3c_0 + c_5)t^2 + 1)$$

if $n^2 p \gtrsim n\log^2 n$. Hence, there exists a universal constant $c_2 > 0$ that is independent of $t$ such that

$$\left(\max_{i\in[n]}\left|\left\|\boldsymbol{x}^{(t)}\right\|_{2,i}^2 - \left\|\widetilde{\boldsymbol{x}}^{(t)}\right\|_2^2\right|\right)\left\|\boldsymbol{x}^{(t)}\right\|_2 \le \frac{1}{2}c_2\sqrt{\frac{\mu^3 \log^3 n}{np}}\beta_0^3((3c_0 + c_5)t^2 + 1)$$

for all $t \le T_1$. With the decomposition (D.11), we have

$$\left\|\boldsymbol{x}^{(t+1)} - \widehat{\boldsymbol{x}}^{(t+1)}\right\|_2 \le \left(1 - \eta\|\widetilde{\boldsymbol{x}}\|_2^2 + \eta\lambda^\circ\right)\|\boldsymbol{x} - \widehat{\boldsymbol{x}}\|_2 + \eta\left(\max_{i\in[n]}\left|\|\boldsymbol{x}\|_{2,i}^2 - \|\widetilde{\boldsymbol{x}}\|_2^2\right|\right)\|\boldsymbol{x}\|_2$$

$$\le (1 + \eta\lambda^\circ)\|\boldsymbol{x} - \widehat{\boldsymbol{x}}\|_2 + \frac{1}{2}c_2(\eta\lambda^\star)^3 \frac{1}{\lambda^\star}\sqrt{\frac{\mu^3 \log^3 n}{np}}\beta_0^3((3c_0 + c_5)t^2 + 1).$$

From (A.1), there exists a universal constant $c_3 > 0$ such that $\eta\lambda^\circ \le \eta\lambda^\star + \frac{c_3}{\log^2 n}$ if $n^2 p \gtrsim \mu^2 n\log^5 n$. Combining all, for all $s \le t$, we have

$$\left\|\boldsymbol{x}^{(s+1)} - \widehat{\boldsymbol{x}}^{(s+1)}\right\|_2 \le \left(1 + \eta\lambda^\star + \frac{c_3}{\log^2 n}\right)\left\|\boldsymbol{x}^{(s)} - \widehat{\boldsymbol{x}}^{(s)}\right\|_2$$

$$+ \frac{1}{2}c_2(\eta\lambda^\star)^3 \frac{1}{\lambda^\star}\sqrt{\frac{\mu^3 \log^3 n}{np}}\beta_0^3((3c_0 + c_5)s^2 + 1).$$

An analysis on the recursive equation

$$x_{s+1} = \left(1 + \eta\lambda^\star + \frac{c_3}{\log^2 n}\right)x_s + \frac{1}{2}c_2(\eta\lambda^\star)^3 \frac{1}{\lambda^\star}\sqrt{\frac{\mu^3 \log^3 n}{np}}\beta_0^3((3c_0 + c_5)s^2 + 1), \quad x_0 = 0,$$

proves that

$$\left\|\boldsymbol{x}^{(t+1)} - \widehat{\boldsymbol{x}}^{(t+1)}\right\|_2 \le c_2(3c_0 + c_5 + 1)\frac{1}{\lambda^\star}\sqrt{\frac{\mu^3 \log^3 n}{np}}(1 + \eta\lambda^\star)^{t+1}\beta_0^3.$$

**(D.8) at $(t+1)$**  We decompose $\boldsymbol{x}^{(t+1)} - \boldsymbol{x}^{(t+1,l)}$ as

$$\boldsymbol{x}^{(t+1)} - \boldsymbol{x}^{(t+1,l)} = (1 - \eta\|\widetilde{\boldsymbol{x}}\|_2^2)(\boldsymbol{x} - \boldsymbol{x}^{(l)}) - 2\eta \underbrace{\widetilde{\boldsymbol{x}}\widetilde{\boldsymbol{x}}^\top(\boldsymbol{x} - \boldsymbol{x}^{(l)})}_{\text{①}}$$

$$- \eta \underbrace{\int_0^1 \left( \nabla^2 g(\boldsymbol{x}(\tau)) - \left( \|\widetilde{\boldsymbol{x}}\|_2^2 \boldsymbol{I} + 2\widetilde{\boldsymbol{x}}\widetilde{\boldsymbol{x}}^\top \right) \right)(\boldsymbol{x} - \boldsymbol{x}^{(l)}) \mathrm{d}\tau}_{\text{②}}$$

$$- \eta \underbrace{\left( \frac{1}{p}\mathcal{P}_{\Omega_l}\left( \boldsymbol{x}^{(t,l)}\boldsymbol{x}^{(t,l)\top} \right) - \mathcal{P}_l\left( \boldsymbol{x}^{(t,l)}\boldsymbol{x}^{(t,l)\top} \right) \right)\boldsymbol{x}^{(t,l)}}_{\text{③}}$$

$$+ \eta \underbrace{\lambda^{(l)}\boldsymbol{u}^{(l)}\boldsymbol{u}^{(l)\top}(\boldsymbol{x}^{(t)} - \boldsymbol{x}^{(t,l)})}_{\text{④}} + \eta \underbrace{\left( \boldsymbol{M}^\circ - \lambda^{(l)}\boldsymbol{u}^{(l)}\boldsymbol{u}^{(l)\top} \right)(\boldsymbol{x}^{(t)} - \boldsymbol{x}^{(t,l)})}_{\text{⑤}}$$

$$+ \eta \underbrace{\left( \frac{1}{p}\mathcal{P}_\Omega(\boldsymbol{M}^\star) - \mathcal{P}_\Omega^{(l)}(\boldsymbol{M}^\star) \right)\boldsymbol{x}^{(t,l)}}_{\text{⑥}} + \eta \underbrace{\left( \frac{1}{p}\mathcal{P}_\Omega(\boldsymbol{E}) - \boldsymbol{E}^{(l)} \right)\boldsymbol{x}^{(t,l)}}_{\text{⑦}},$$

where $\boldsymbol{x}^{(l)}(\tau) = \boldsymbol{x}^{(l)} + \tau(\boldsymbol{x} - \boldsymbol{x}^{(l)})$. ① is easily bounded by

$$\|①\|_2 \le \|\widetilde{\boldsymbol{x}}\|_2^2 \|\boldsymbol{x} - \boldsymbol{x}^{(l)}\|_2 \lesssim \beta_0^2 \|\boldsymbol{x} - \boldsymbol{x}^{(l)}\|_2 \lesssim \lambda^\star \sqrt{\frac{1}{\mu^5 \log^{26} n}} \sqrt{\frac{np}{n}} \|\boldsymbol{x} - \boldsymbol{x}^{(l)}\|_2.$$

From Lemma G.10 and (C.4), for all $0 \le \tau \le 1$, we have

$$\left\| \nabla^2 g(\boldsymbol{x}^{(l)}(\tau)) - \left( \|\widetilde{\boldsymbol{x}}\|_2^2 \boldsymbol{I} + 2\widetilde{\boldsymbol{x}}\widetilde{\boldsymbol{x}}^\top \right) \right\| \lesssim n\|\boldsymbol{x}(\tau) - \widetilde{\boldsymbol{x}}\|_\infty(\|\boldsymbol{x}(\tau)\|_\infty + \|\widetilde{\boldsymbol{x}}\|_\infty) + \sqrt{\frac{\log^3 n}{np}}\beta_0^2. \tag{D.12}$$

From the definition of $\boldsymbol{x}(\tau)$, we have

$$\left\| \boldsymbol{x}^{(l)}(\tau) - \widetilde{\boldsymbol{x}} \right\|_\infty \le (1 - \tau)\left\| \boldsymbol{x}^{(l)} - \boldsymbol{x} \right\|_2 + \|\boldsymbol{x} - \widetilde{\boldsymbol{x}}\|_\infty \lesssim \sqrt{\frac{\mu^3 \log^6 n}{np}} \frac{\beta_0}{\sqrt{n}},$$

where the last inequality is from the induction hypotheses (D.6), (D.8), and the fact that $t \le T_1 \lesssim \log n$. Inserting this bound back to (D.12), we get

$$\left\| \nabla^2 g(\boldsymbol{x}^{(l)}(\tau)) - \left( \|\widetilde{\boldsymbol{x}}\|_2^2 \boldsymbol{I} + 2\widetilde{\boldsymbol{x}}\widetilde{\boldsymbol{x}}^\top \right) \right\| \lesssim \sqrt{\frac{\mu^3 \log^7 n}{np}}\beta_0^2, \tag{D.13}$$

which also implies

$$\|②\|_2 \lesssim \sqrt{\frac{\mu^3 \log^7 n}{np}}\beta_0^2 \|\boldsymbol{x} - \boldsymbol{x}^{(l)}\|_2 \lesssim \lambda^\star \sqrt{\frac{1}{\mu^2 n \log^{19} n}} \|\boldsymbol{x} - \boldsymbol{x}^{(l)}\|_2.$$

It is implied from Lemma G.11 that

$$\|③\|_2 \le \left\| \boldsymbol{x}^{(l)} \right\|_\infty^2 \sqrt{\frac{\log n}{p}} \left\| \boldsymbol{x}^{(l)} \right\|_2 \lesssim \sqrt{\frac{\log^4 n}{np}} \frac{\beta_0^3}{\sqrt{n}} \lesssim \lambda^\star \sqrt{\frac{1}{\mu^5 n \log^{22} n}} \frac{\beta_0}{\sqrt{n}}.$$

A bound on ④ follows from the induction hypothesis (D.9) and the spectral bound (A.2).

$$\|④\|_2 \le \lambda^{(l)} \left| \boldsymbol{u}^{(l)\top}(\boldsymbol{x}^{(t)} - \boldsymbol{x}^{(t,l)}) \right| \lesssim \lambda^\star \sqrt{\frac{\mu^3 \log^2 n}{np}} \frac{\beta_0}{n}(1 + \eta\lambda^\star)^t \lesssim \lambda^\star \frac{\sqrt{\mu^7 \log^{23} n}}{np} \frac{\beta_0}{\sqrt{n}}.$$

The second largest eigenvalue of $\boldsymbol{M}^{(l)}$ is at most $\left\| \boldsymbol{M}^{(l)} - \boldsymbol{M}^\star \right\|$ by Weyl's Theorem, and from Lemmas A.1 and A.2, we have

$$\left\| \boldsymbol{M}^{(l)} - \boldsymbol{M}^\star \right\| \lesssim \left\| \boldsymbol{M}^{(l)} - \boldsymbol{M}^\circ \right\| + \|\boldsymbol{M}^\circ - \boldsymbol{M}^\star\| \lesssim \lambda^\star \mu \sqrt{\frac{\log n}{np}}.$$

Hence, we get

$$\left\|\text{⑤}\right\|_2 \le \left(\left\|\boldsymbol{M}^\circ - \boldsymbol{M}^{(l)}\right\| + \left\|\boldsymbol{M}^{(l)} - \lambda^{(l)}\boldsymbol{u}^{(l)}\boldsymbol{u}^{(l)\top}\right\|\right)\left\|\boldsymbol{x} - \boldsymbol{x}^{(l)}\right\|_2$$

$$\lesssim \lambda^\star\mu\sqrt{\frac{\log n}{np}}\left\|\boldsymbol{x} - \boldsymbol{x}^{(l)}\right\|_2.$$

Lastly, we apply Lemmas G.11 and G.13 to get

$$\left\|\text{⑥}\right\|_2 \lesssim \lambda^\star\mu\sqrt{\frac{\log n}{np}}\frac{\beta_0}{\sqrt{n}}, \quad \left\|\text{⑦}\right\|_2 \lesssim \lambda^\star\mu\sqrt{\frac{\log^2 n}{np}}\frac{\beta_0}{\sqrt{n}},$$

There exists a universal constant $c_4 > 0$ such that

$$\eta\left(2\left\|\text{①}\right\|_2 + \left\|\text{②}\right\|_2 + \left\|\text{⑤}\right\|_2\right) \le \frac{c_4}{\log^2 n}\left\|\boldsymbol{x} - \boldsymbol{x}^{(l)}\right\|_2$$

if $n^2 p \gtrsim \mu^2 n \log^5 n$, and there exists a universal constant $c_5 > 0$ such that

$$\eta\left(\left\|\text{③}\right\|_2 + \left\|\text{④}\right\|_2 + \left\|\text{⑥}\right\|_2 + \left\|\text{⑦}\right\|_2\right) \le \frac{1}{2}c_5\mu\sqrt{\frac{\log^2 n}{np}}\frac{\beta_0}{\sqrt{n}}.$$

if $n^2 p \gtrsim \mu^5 n \log^{20} n$. Combining all, we have

$$\left\|\boldsymbol{x}^{(s+1)} - \boldsymbol{x}^{(s+1,l)}\right\|_2 \le \left(1 - \eta\left\|\widetilde{\boldsymbol{x}}^{(s)}\right\|_2^2 + \frac{c_4}{\log^2 n}\right)\left\|\boldsymbol{x}^{(s)} - \boldsymbol{x}^{(s,l)}\right\|_2 + \frac{1}{2}c_5\mu\sqrt{\frac{\log^2 n}{np}}\frac{\beta_0}{\sqrt{n}}$$

$$\le \left(1 + \frac{c_4}{\log^2 n}\right)\left\|\boldsymbol{x}^{(s)} - \boldsymbol{x}^{(s,l)}\right\|_2 + \frac{1}{2}c_5\mu\sqrt{\frac{\log^2 n}{np}}\frac{\beta_0}{\sqrt{n}}.$$

for all $s \le t$. An analysis on the recursive equation

$$x_{s+1} = \left(1 + \frac{c_4}{\log^2 n}\right)x_s + \frac{1}{2}c_5\mu\sqrt{\frac{\log^2 n}{np}}\frac{\beta_0}{\sqrt{n}}, \quad x_0 = 0$$

gives the desired bound

$$\left\|\boldsymbol{x}^{(t+1)} - \boldsymbol{x}^{(t+1,l)}\right\|_2 \le \frac{\log^2 n}{c_4}\left(\left(1 + \frac{c_4}{\log^2 n}\right)^{t+1} - 1\right)\frac{1}{2}c_5\mu\sqrt{\frac{\log^2 n}{np}}\frac{\beta_0}{\sqrt{n}}$$

$$\le c_5\mu\sqrt{\frac{\log^2 n}{np}}\frac{\beta_0}{\sqrt{n}}(t+1),$$

where we used basic inequalities $(1 + x)^a \le e^{ax}$ and $e^{ax} - 1 \le 2ax$ which hold if $x$ is small and $ax$ is small, respectively.

**(D.9) at $(t+1)$**  We decompose $x^{(t+1)} - x^{(t+1,l)}$ as

$$x^{(t+1)} - x^{(t+1,l)}$$

$$= (x - \eta \nabla f(x)) - \left(x^{(l)} - \eta \nabla f^{(l)}(x^{(l)})\right)$$

$$= (x - \eta \nabla g(x)) - \left(x^{(l)} - \eta \nabla g^{(l)}(x^{(l)})\right) + \eta \left(M^\circ x - M^{(l)} x^{(l)}\right)$$

$$= (x - \eta \nabla g(x)) - \left(x^{(l)} - \eta \nabla g(x^{(l)})\right) - \eta \left(\nabla g(x^{(l)}) - \nabla g^{(l)}(x^{(l)})\right)$$

$$\quad + \eta (M^\circ - M^{(l)}) x + \eta M^{(l)}(x - x^{(l)})$$

$$= \int_0^1 (I - \eta \nabla^2 g(x^{(l)}(\tau)))(x - x^{(l)}) \, d\tau - \eta \left(\frac{1}{p} \mathcal{P}_{\Omega_l}\left(x^{(l)} x^{(l)\top}\right) - \mathcal{P}_l\left(x^{(l)} x^{(l)\top}\right)\right) x^{(l)}$$

$$\quad + \eta M^{(l)}(x - x^{(l)}) + \eta \left(M^\circ - M^{(l)}\right)(x - x^{(l)}) + \eta \left(M^\circ - M^{(l)}\right) x^{(l)}$$

$$= (1 - \eta \|\widetilde{x}\|_2^2)(x - x^{(l)}) - 2\eta \widetilde{x} \widetilde{x}^\top (x - x^{(l)})$$

$$\quad - \eta \int_0^1 \left(\nabla^2 g(x^{(l)}(\tau)) - \left(\|\widetilde{x}\|_2^2 I + 2\widetilde{x}\widetilde{x}^\top\right)\right)(x - x^{(l)}) d\tau$$

$$\quad - \eta \left(\frac{1}{p} \mathcal{P}_{\Omega_l}\left(x^{(l)} x^{(l)\top}\right) - \mathcal{P}_l\left(x^{(l)} x^{(l)\top}\right)\right) x^{(l)}$$

$$\quad + \eta M^{(l)}(x - x^{(l)}) + \eta \left(M^\circ - M^{(l)}\right)(x - x^{(l)}) + \eta \left(M^\circ - M^{(l)}\right) x^{(l)},$$

where $x^{(l)}(\tau) = x^{(l)} + \tau(x - x^{(l)})$. Then, we take inner product with $u^{(l)}$ on both sides.

$$u^{(l)\top}\left(x^{(t+1)} - x^{(t+1,l)}\right) = \left(1 - \eta \|\widetilde{x}\|_2^2\right) u^{(l)\top}(x - x^{(l)}) - 2\eta \underbrace{u^{(l)\top} \widetilde{x} \widetilde{x}^\top \left(x - x^{(l)}\right)}_{\text{①}}$$

$$- \eta \underbrace{\int_0^1 u^{(l)\top}\left(\nabla^2 g(x(\tau)) - \left(\|\widetilde{x}\|_2^2 I + 2\widetilde{x}\widetilde{x}^\top\right)\right)(x - x^{(l)}) d\tau}_{\text{②}}$$

$$- \eta \underbrace{u^{(l)\top}\left(\frac{1}{p} \mathcal{P}_{\Omega_l}\left(x^{(l)} x^{(l)\top}\right) - \mathcal{P}_l\left(x^{(l)} x^{(l)\top}\right)\right) x^{(l)}}_{\text{③}}$$

$$+ \eta \lambda^{(l)} u^{(l)\top}(x - x^{(l)}) + \eta \underbrace{u^{(l)\top}\left(M^\circ - M^{(l)}\right)(x - x^{(l)})}_{\text{④}}$$

$$+ \eta \underbrace{u^{(l)\top}\left(M^\circ - M^{(l)}\right) x^{(l)}}_{\text{⑤}}$$

For the term ①, we have

$$\left|u^{(l)\top} \widetilde{x}\right| \le \left|u^{(l)\top} x^{(0)}\right| + (1 + \eta \lambda^\star)^t \left|u^{\star\top} x^{(0)}\right| \left|u^{(l)\top} u^\star\right| \lesssim \sqrt{\frac{\log n}{n}}(1 + \eta \lambda^\star)^t \beta_0$$

by Lemma A.5, and thus,

$$|①| \le \left|u^{(l)\top} \widetilde{x}\right| \|\widetilde{x}\|_2 \left\|x - x^{(l)}\right\|_2 \lesssim \sqrt{\frac{\log n}{n}}(1 + \eta \lambda^\star)^t \beta_0^2 \left\|x - x^{(l)}\right\|_2$$

$$\lesssim \mu \sqrt{\frac{\log^3 n}{np}} \frac{\beta_0^3}{n}(1 + \eta \lambda^\star)^t t$$

$$\lesssim \lambda^\star \sqrt{\frac{\mu}{np}} \frac{\beta_0}{n}.$$

The definition of Phase I was used to bound $(1 + \eta\lambda^\star)^t t$ in deriving the last line. We use (D.13) to get

$$|\textcircled{2}| \lesssim \int_0^1 \left\| \nabla^2 g(\boldsymbol{x}(\tau)) - \left( \|\widetilde{\boldsymbol{x}}\|_2^2 \boldsymbol{I} + 2\widetilde{\boldsymbol{x}}\widetilde{\boldsymbol{x}}^\top \right) \right\| \left\| \boldsymbol{x} - \boldsymbol{x}^{(l)} \right\|_2 \left\| \boldsymbol{u}^{(l)} \right\|_2 \mathrm{d}\tau$$

$$\lesssim \sqrt{\frac{\mu^3 \log^7 n}{np}} \beta_0^2 \cdot \mu \sqrt{\frac{\log^2 n}{np}} \frac{\beta_0}{\sqrt{n}} t \lesssim \lambda^\star \sqrt{\frac{\mu}{np \log^{15} n}} \frac{\beta_0}{n}.$$

We apply Lemma G.12 to $\textcircled{3}$ to yield

$$|\textcircled{3}| \lesssim \left\| \boldsymbol{x}^{(l)} \right\|_\infty^2 \sqrt{\frac{\log n}{p}} \left( \left\| \boldsymbol{u}^{(l)} \right\|_2 \left\| \boldsymbol{x}^{(l)} \right\|_\infty + \left\| \boldsymbol{x}^{(l)} \right\|_2 \left\| \boldsymbol{u}^{(l)} \right\|_\infty \right)$$

$$\lesssim \sqrt{\frac{\mu \log^4 n}{np}} \frac{\beta_0^3}{n} \lesssim \lambda^\star \sqrt{\frac{1}{\mu^3 n \log^{22} n}} \frac{\beta_0}{n}.$$

We divide $\textcircled{4}$ into two terms that are related to sampling and noise, respectively.

$$\textcircled{4} = \boldsymbol{u}^{(l)\top} \left( \frac{1}{p} \mathcal{P}_\Omega(\boldsymbol{M}^\star) - \mathcal{P}_\Omega^{(l)}(\boldsymbol{M}^\star) \right) (\boldsymbol{x} - \boldsymbol{x}^{(l)}) + \boldsymbol{u}^{(l)\top} \left( \frac{1}{p} \mathcal{P}_\Omega(\boldsymbol{E}) - \boldsymbol{E}^{(l)} \right) (\boldsymbol{x} - \boldsymbol{x}^{(l)})$$

Then, Cauchy-Schwartz inequality is applied to yield

$$|\textcircled{4}| \leq \left\| \left( \frac{1}{p} \mathcal{P}_\Omega(\boldsymbol{M}^\star) - \mathcal{P}_\Omega^{(l)}(\boldsymbol{M}^\star) \right) \boldsymbol{u}^{(l)} \right\|_2 \left\| \boldsymbol{x} - \boldsymbol{x}^{(l)} \right\|_2 + \left\| \left( \frac{1}{p} \mathcal{P}_\Omega(\boldsymbol{E}) - \boldsymbol{E}^{(l)} \right) \boldsymbol{u}^{(l)} \right\|_2 \left\| \boldsymbol{x} - \boldsymbol{x}^{(l)} \right\|_2.$$

Applying Lemmas G.11 and G.13 to the two terms, respectively, we get

$$|\textcircled{4}| \lesssim \lambda^\star \sqrt{\frac{\mu^3 \log^2 n}{np}} \frac{1}{\sqrt{n}} \left\| \boldsymbol{x} - \boldsymbol{x}^{(l)} \right\|_2 \lesssim \lambda^\star \frac{\sqrt{\mu^5 \log^5 n}}{np} \frac{\beta_0}{n}.$$

For the term $\textcircled{5}$, we decompose it into two terms as for $\textcircled{4}$.

$$|\textcircled{5}| \leq \left| \boldsymbol{u}^{(l)\top} \left( \frac{1}{p} \mathcal{P}_\Omega(\boldsymbol{M}^\star) - \mathcal{P}_\Omega^{(l)}(\boldsymbol{M}^\star) \right) \boldsymbol{x}^{(l)} \right| + \left| \boldsymbol{u}^{(l)\top} \left( \frac{1}{p} \mathcal{P}_\Omega(\boldsymbol{E}) - \boldsymbol{E}^{(l)} \right) \boldsymbol{x}^{(l)} \right|$$

Then, we apply Lemmas G.12 and G.14 to each term to obtain

$$|\textcircled{5}| \lesssim \lambda^\star \sqrt{\frac{\mu^3 \log^3 n}{np}} \frac{\beta_0}{n}.$$

Combining all, there exists a universal constant $c_6 > 0$ such that

$$\eta \left( 2|\textcircled{1}| + |\textcircled{2}| + |\textcircled{3}| + |\textcircled{4}| + |\textcircled{5}| \right) \leq \frac{\eta\lambda^\star}{2} c_6 \sqrt{\frac{\mu^3 \log^3 n}{np}} \frac{\beta_0}{n}$$

if $n^2 p \gtrsim \mu^2 n \log^3 n$, and there exists a universal constant $c_7 > 0$ such that

$$\eta\lambda^{(l)} \leq \eta\lambda^\star + \frac{c_7}{\log^2 n}$$

by (A.2) if $n^2 p \gtrsim \mu^2 n \log^5 n$. Finally, we have

$$\left| \boldsymbol{u}^{(l)\top} \left( \boldsymbol{x}^{(t+1)} - \boldsymbol{x}^{(t+1,l)} \right) \right| \leq \left( 1 - \eta\|\widetilde{\boldsymbol{x}}\|_2^2 + \eta\lambda^{(l)} \right) \left| \boldsymbol{u}^{(l)\top}(\boldsymbol{x} - \boldsymbol{x}^{(l)}) \right| + \frac{\eta\lambda^\star}{2} c_6 \sqrt{\frac{\mu^3 \log^3 n}{np}} \frac{\beta_0}{n}$$

$$\leq \left( 1 + \eta\lambda^\star + \frac{c_7}{\log^2 n} \right) \left| \boldsymbol{u}^{(l)\top}(\boldsymbol{x} - \boldsymbol{x}^{(l)}) \right| + \frac{\eta\lambda^\star}{2} c_6 \sqrt{\frac{\mu^3 \log^3 n}{np}} \frac{\beta_0}{n}.$$

An analysis on the recursive equation

$$x_{t+1} = \left( 1 + \eta\lambda^\star + \frac{c_7}{\log^2 n} \right) x_t + \frac{\eta\lambda^\star}{2} c_6 \sqrt{\frac{\mu^3 \log^3 n}{np}} \frac{\beta_0}{n}, \quad x_0 = 0$$

gives the bound

$$\left| \boldsymbol{u}^{(l)\top} \left( \boldsymbol{x}^{(t+1)} - \boldsymbol{x}^{(t+1,l)} \right) \right| \leq c_6 \sqrt{\frac{\mu^3 \log^3 n}{np}} (1 + \eta\lambda^\star)^{t+1} \frac{\beta_0}{n}.$$

**(D.10) at $(t+1)$**    We decompose $(\boldsymbol{x}^{(t+1,l)} - \widetilde{\boldsymbol{x}}^{(t+1)})_l$ as

$$(\boldsymbol{x}^{(t+1,l)} - \widetilde{\boldsymbol{x}}^{(t+1)})_l = \left(1 - \eta\|\widetilde{\boldsymbol{x}}\|_2^2\right)(\boldsymbol{x}^{(l)} - \widetilde{\boldsymbol{x}})_l + \eta\lambda^\star \boldsymbol{u}^{\star\top}(\boldsymbol{x}^{(l)} - \widetilde{\boldsymbol{x}})u_l^\star + \eta\left(\|\widetilde{\boldsymbol{x}}\|_2^2 - \left\|\boldsymbol{x}^{(l)}\right\|_2^2\right)x_l^{(l)},$$

and this implies

$$\left|(\boldsymbol{x}^{(t+1,l)} - \widetilde{\boldsymbol{x}}^{(t+1)})_l\right| \le \left(1 - \eta\|\widetilde{\boldsymbol{x}}\|_2^2\right)\left|(\boldsymbol{x}^{(l)} - \widetilde{\boldsymbol{x}})_l\right|$$
$$+ \eta\lambda^\star\left\|\boldsymbol{x}^{(l)} - \widetilde{\boldsymbol{x}}\right\|_2\|\boldsymbol{u}^\star\|_\infty + \eta\|\widetilde{\boldsymbol{x}}\|_2\left\|\boldsymbol{x}^{(l)} - \widetilde{\boldsymbol{x}}\right\|_2\|\widetilde{\boldsymbol{x}}\|_\infty.$$

From (D.5) and (D.8), we have

$$\left\|\boldsymbol{x}^{(l)} - \widetilde{\boldsymbol{x}}\right\|_2 \le \left\|\boldsymbol{x}^{(l)} - \boldsymbol{x}\right\|_2 + \|\boldsymbol{x} - \widetilde{\boldsymbol{x}}\|_2 \le 3c_0\mu\sqrt{\frac{\log n}{np}}\beta_0 t.$$

If $n$ is sufficiently large, we have

$$\eta\lambda^\star\|\boldsymbol{u}^\star\|_\infty + \eta\|\widetilde{\boldsymbol{x}}\|_2\|\widetilde{\boldsymbol{x}}\|_\infty \le \sqrt{\frac{\mu\log n}{n}}$$

for all $t \le T_1$ because

$$\eta\lambda^\star\|\boldsymbol{u}^\star\|_\infty + \eta\|\widetilde{\boldsymbol{x}}\|_2\|\widetilde{\boldsymbol{x}}\|_\infty \lesssim \eta\lambda^\star\sqrt{\frac{\mu}{n}} + \eta\sqrt{\frac{\log^2 n}{n}}\beta_0^2 \lesssim \sqrt{\frac{\mu}{n}}.$$

Hence, we have

$$\left|(\boldsymbol{x}^{(s+1,l)} - \widetilde{\boldsymbol{x}}^{(s+1)})_l\right| \le \left|(\boldsymbol{x}^{(s,l)} - \widetilde{\boldsymbol{x}}^{(s)})_l\right| + 3c_0\sqrt{\frac{\mu^3\log^2 n}{np}}\frac{\beta_0}{\sqrt{n}}s$$

for all $s \le t$. Finally, we have

$$\left|(\boldsymbol{x}^{(t+1,l)} - \widetilde{\boldsymbol{x}}^{(t+1)})_l\right| \le 3c_0\sqrt{\frac{\mu^3\log^2 n}{np}}\frac{\beta_0}{\sqrt{n}}\sum_{s=1}^t s$$
$$\le 3c_0\sqrt{\frac{\mu^3\log^2 n}{np}}\frac{\beta_0}{\sqrt{n}}(t+1)^2.$$

**(D.5) at $(t+1)$**    We can obtain this through the combination of (D.7) and Lemma 5.2.

$$\left\|\boldsymbol{x}^{(t+1)} - \widetilde{\boldsymbol{x}}^{(t+1)}\right\|_2 \le \left\|\boldsymbol{x}^{(t+1)} - \widehat{\boldsymbol{x}}^{(t+1)}\right\|_2 + \left\|\widehat{\boldsymbol{x}}^{(t+1)} - \widetilde{\boldsymbol{x}}^{(t+1)}\right\|_2$$
$$\le c_2(3c_0 + c_5 + 1)\frac{1}{\lambda^\star}\sqrt{\frac{\mu^3\log^3 n}{np}}(1 + \eta\lambda^\star)^{t+1}\beta_0^3 + c_0\mu\sqrt{\frac{\log n}{np}}\beta_0(t+1)$$
$$\le c_2(3c_0 + c_5 + 1)\frac{1}{\lambda^\star}\sqrt{\frac{\mu^3\log^3 n}{np}}(1 + \eta\lambda^\star)^{T_1}\beta_0^3 + c_0\mu\sqrt{\frac{\log n}{np}}\beta_0(t+1)$$
$$\le c_1c_2(3c_0 + c_5 + 1)\mu\sqrt{\frac{1}{np\log^2 n}}\beta_0 + c_0\mu\sqrt{\frac{\log n}{np}}\beta_0(t+1)$$
$$\le 2c_0\mu\sqrt{\frac{\log n}{np}}\beta_0(t+1)$$

**(D.6) at $(t+1)$**  The $l$th component of $\boldsymbol{x}^{(t+1)} - \widetilde{\boldsymbol{x}}^{(t+1)}$ is bounded by

$$
\begin{aligned}
\left|(\boldsymbol{x}^{(t+1)} - \widetilde{\boldsymbol{x}}^{(t+1)})_l\right| &\leq \left\|\boldsymbol{x}^{(t+1)} - \boldsymbol{x}^{(t+1,l)}\right\|_\infty + \left|(\boldsymbol{x}^{(t+1,l)} - \widetilde{\boldsymbol{x}}^{(t+1)})_l\right| \\
&\leq \left\|\boldsymbol{x}^{(t+1)} - \boldsymbol{x}^{(t+1,l)}\right\|_2 + \left|(\boldsymbol{x}^{(t+1,l)} - \widetilde{\boldsymbol{x}}^{(t+1)})_l\right| \\
&\leq c_5\mu\sqrt{\frac{\log^2 n}{np}}\frac{\beta_0}{\sqrt{n}}(t+1) + 3c_0\sqrt{\frac{\mu^3 \log^2 n}{np}}\frac{\beta_0}{\sqrt{n}}(t+1)^2 \\
&\leq (3c_0 + c_5)\sqrt{\frac{\mu^3 \log^2 n}{np}}\frac{\beta_0}{\sqrt{n}}(t+1)^2.
\end{aligned}
$$

**At $t = T_1$**  Because $T_1 \lesssim \log n$, it is implied from Lemma D.2 that at $t = T_1$, there exists a constant $c_7 > 0$ such that

$$
\begin{aligned}
\left\|\boldsymbol{x}^{(t)} - \widetilde{\boldsymbol{x}}^{(t)}\right\|_2 &\leq c_7\mu\sqrt{\frac{\log^3 n}{np}}\beta_0, \\
\left\|\boldsymbol{x}^{(t)} - \widetilde{\boldsymbol{x}}^{(t)}\right\|_\infty &\leq c_7\sqrt{\frac{\mu^3 \log^8 n}{np}}\frac{\beta_0}{\sqrt{n}}, \\
\left\|\boldsymbol{x}^{(t)} - \boldsymbol{x}^{(t,l)}\right\|_2 &\leq c_7\mu\sqrt{\frac{\log^4 n}{np}}\frac{\beta_0}{\sqrt{n}}, \\
\left|\left(\boldsymbol{x}^{(t,l)} - \widetilde{\boldsymbol{x}}^{(t)}\right)_l\right| &\leq c_7\sqrt{\frac{\mu^3 \log^8 n}{np}}\frac{\beta_0}{\sqrt{n}}.
\end{aligned}
\tag{D.14}
$$

These bounds serve as a base case for the induction of the next part.

# E  Phase II

This section is mostly devoted to the proof of Lemma E.1 which is a formal version of Lemma 6.1.

**Lemma E.1.** *Suppose that (D.14) holds at $t = T_1$ and the initialization point $\boldsymbol{x}^{(0)}$ satisfies (B.1) to (B.6). Then, for all $T_1 < t \leq T_2$, we have*

$$
\left\|\boldsymbol{x}^{(t)} - \widetilde{\boldsymbol{x}}^{(t)}\right\|_2 \leq 2c_7\mu\sqrt{\frac{\log^3 n}{np}}\beta_0(1 + \eta\lambda^\star)^{t-T_1}, \tag{E.1}
$$

$$
\left\|\boldsymbol{x}^{(t)} - \widetilde{\boldsymbol{x}}^{(t)}\right\|_\infty \leq c_{13}\sqrt{\frac{\mu^3 \log^8 n}{np}}\frac{\beta_0}{\sqrt{n}}(1 + \eta\lambda^\star)^{t-T_1}, \tag{E.2}
$$

$$
\left\|\boldsymbol{x}^{(t)} - \boldsymbol{x}^{(t,l)}\right\|_2 \leq 2c_7\mu\sqrt{\frac{\log^5 n}{np}}\frac{\beta_0}{\sqrt{n}}(1 + \eta\lambda^\star)^{t-T_1}, \tag{E.3}
$$

$$
\left|\left(\boldsymbol{x}^{(t,l)} - \widetilde{\boldsymbol{x}}^{(t)}\right)_l\right| \leq 3c_7\sqrt{\frac{\mu^3 \log^8 n}{np}}\frac{\beta_0}{\sqrt{n}}(1 + \eta\lambda^\star)^{t-T_1}, \tag{E.4}
$$

*with high probability, where $T_2$ is the largest $t$ such that $\widetilde{\beta}_t^2 \leq \lambda^\star\left(1 - \frac{1}{\log n}\right)$, and $c_{13} > 0$ is a constant.*

**Proof of Theorems 3.1 and 3.3**  We first explain how Theorems 3.1 and 3.3 are derived from Lemmas D.2 and E.1. We first focus on $\left\|\boldsymbol{x}^{(t)} - \widetilde{\boldsymbol{x}}^{(t)}\right\|_2$. For $t \leq T_1$, from Lemma D.2 and (C.3), we have

$$
\left\|\boldsymbol{x}^{(t)} - \widetilde{\boldsymbol{x}}^{(t)}\right\|_2 \lesssim \mu\sqrt{\frac{\log^3 n}{np}}\beta_0 \lesssim \frac{1}{\log n}\beta_0 \lesssim \frac{1}{\sqrt{\log n}}\left\|\widetilde{\boldsymbol{x}}^{(t)}\right\|_2
$$

provided that $n^2 p \gtrsim \mu^2 n \log^7 n$. For $T_1 < t \leq T_2$, from the definition of $T_1$ and Lemma E.1, we have

$$\left\| \boldsymbol{x}^{(t)} - \widetilde{\boldsymbol{x}}^{(t)} \right\|_2 \lesssim \sqrt{\frac{1}{\log^{18} n}} \frac{\beta_0}{\sqrt{n}} (1 + \eta \lambda^\star)^t.$$

From the lower bound of Lemma C.1, for all $t \leq T_2'$, we have

$$\left\| \boldsymbol{x}^{(t)} - \widetilde{\boldsymbol{x}}^{(t)} \right\|_2 \lesssim \sqrt{\frac{1}{\log^{18} n}} \left( 1 + \frac{(1 + \eta \lambda^\star)^t}{\sqrt{n}} \right) \beta_0 \lesssim \sqrt{\frac{1}{\log^{17} n}} \left\| \widetilde{\boldsymbol{x}}^{(t)} \right\|_2 \lesssim \frac{1}{\sqrt{\log n}} \left\| \widetilde{\boldsymbol{x}}^{(t)} \right\|_2.$$

Now, for $T_2' < t \leq T_2$, we have

$$\left\| \boldsymbol{x}^{(t)} - \widetilde{\boldsymbol{x}}^{(t)} \right\|_2 \lesssim \sqrt{\frac{1}{\log^{18} n}} \left( 1 + \frac{(1 + \eta \lambda^\star)^{T_2'}}{\sqrt{n}} \right) \beta_0 (1 + \eta \lambda^\star)^{t - T_2'}$$

$$\lesssim \sqrt{\frac{1}{\log^{17} n}} \left\| \widetilde{\boldsymbol{x}}^{(T_2')} \right\|_2 (1 + \eta \lambda^\star)^{t - T_2'}.$$

For any $T_2' < t \leq T_2$, it is Lemma C.2 implied from Lemma C.2 that $(1 + \eta \lambda^\star)^{t - T_2'} \leq \log^6 n$, and we have $\left\| \widetilde{\boldsymbol{x}}^{(T_2')} \right\|_2 \leq \|\widetilde{\boldsymbol{x}}\|_2$. Hence, we get

$$\left\| \boldsymbol{x}^{(t)} - \widetilde{\boldsymbol{x}}^{(t)} \right\|_2 \lesssim \sqrt{\frac{1}{\log^{17} n}} \sqrt{\log^{12} n} \left\| \widetilde{\boldsymbol{x}}^{(t)} \right\|_2 \lesssim \frac{1}{\sqrt{\log n}} \left\| \widetilde{\boldsymbol{x}}^{(t)} \right\|_2, \tag{E.5}$$

and the proof for (11) of Theorem 3.3 is completed. If we combine this with (C.7), we are able to prove (4) of Theorem 3.1.

We move on to $\left\| \boldsymbol{x}^{(t)} - \widetilde{\boldsymbol{x}}^{(t)} \right\|_\infty$. For $t \leq T_1$, from Lemma D.2 and (C.4), we have

$$\left\| \boldsymbol{x}^{(t)} - \widetilde{\boldsymbol{x}}^{(t)} \right\|_\infty \lesssim \sqrt{\frac{\mu^3 \log^8 n}{np}} \frac{\beta_0}{\sqrt{n}} \lesssim \frac{1}{\log n} \frac{\beta_0}{\sqrt{n}} \lesssim \frac{1}{\sqrt{\log n}} \left\| \widetilde{\boldsymbol{x}}^{(t)} \right\|_\infty$$

provided that $n^2 p \gtrsim \mu^3 n \log^{10} n$. For $T_1 < t \leq T_2'$, from the definition of $T_1$ and Lemma E.1, we have

$$\left\| \boldsymbol{x}^{(t)} - \widetilde{\boldsymbol{x}}^{(t)} \right\|_\infty \lesssim \sqrt{\frac{1}{\log^{13} n}} \frac{\beta_0}{\sqrt{n}} (1 + \eta \lambda^\star)^t.$$

From the lower bound of Lemma C.1, for all $t \leq T_2'$, we have

$$\left\| \boldsymbol{x}^{(t)} - \widetilde{\boldsymbol{x}}^{(t)} \right\|_\infty \lesssim \sqrt{\frac{1}{\log^{13} n}} \left( 1 + \frac{(1 + \eta \lambda^\star)^t}{\sqrt{n}} \right) \frac{\beta_0}{\sqrt{n}} \lesssim \sqrt{\frac{1}{\log^{12} n}} \left\| \widetilde{\boldsymbol{x}}^{(t)} \right\|_\infty \lesssim \frac{1}{\sqrt{\log n}} \left\| \widetilde{\boldsymbol{x}}^{(t)} \right\|_\infty.$$

Now, for $T_2' < t \leq T_2$, if we do the same as before, we get

$$\left\| \boldsymbol{x}^{(t)} - \widetilde{\boldsymbol{x}}^{(t)} \right\|_\infty \lesssim \sqrt{\frac{1}{\log^{13} n}} \sqrt{\log^{12} n} \left\| \widetilde{\boldsymbol{x}}^{(t)} \right\|_2 \frac{1}{\sqrt{n}} \lesssim \frac{1}{\sqrt{\log n}} \left\| \widetilde{\boldsymbol{x}}^{(t)} \right\|_\infty,$$

and the proof for (12) of Theorem 3.3 is completed. If we combine this with (C.8), we are able to prove (5) of Theorem 3.1.

Going through a similar way with (22) and (24), we can complete the proof of Theorems 3.1 and 3.3.

**Proof of Lemma E.1** Before we start the proof, we define a function $G$ as

$$G(\boldsymbol{x}) = \frac{1}{4} \left\| \boldsymbol{x} \boldsymbol{x}^\top \right\|_F^2.$$

The gradient of $G$ satisfies

$$\nabla F(\boldsymbol{x}) = \nabla G(\boldsymbol{x}) - \boldsymbol{M}^\star \boldsymbol{x}.$$

Now, we assume that the hypotheses hold up to the $t$th iteration and show that they hold at the $(t+1)$st iteration. For brevity, we drop the superscript $(t)$ from $\boldsymbol{x}^{(t)}, \boldsymbol{x}^{(t,l)}, \widetilde{\boldsymbol{x}}^{(t)}$ and denote them as $\boldsymbol{x}, \boldsymbol{x}^{(l)}, \widetilde{\boldsymbol{x}}$, respectively.

**(E.1) at $(t+1)$** We decompose $\boldsymbol{x}^{(t+1)} - \widetilde{\boldsymbol{x}}^{(t+1)}$ as

$\boldsymbol{x}^{(t+1)} - \widetilde{\boldsymbol{x}}^{(t+1)}$

$= (\boldsymbol{x} - \eta \nabla f(\boldsymbol{x})) - (\widetilde{\boldsymbol{x}} - \eta \nabla F(\widetilde{\boldsymbol{x}}))$

$= (\boldsymbol{x} - \eta \nabla g(\boldsymbol{x})) - (\widetilde{\boldsymbol{x}} - \eta \nabla G(\widetilde{\boldsymbol{x}})) + \eta \left( \boldsymbol{M}^\circ \boldsymbol{x} - \boldsymbol{M}^\star \widetilde{\boldsymbol{x}} \right)$

$= (\boldsymbol{x} - \eta \nabla g(\boldsymbol{x})) - (\widetilde{\boldsymbol{x}} - \eta \nabla g(\widetilde{\boldsymbol{x}})) - \eta \left( \nabla g(\widetilde{\boldsymbol{x}}) - \nabla G(\widetilde{\boldsymbol{x}}) \right) + \eta \boldsymbol{M}^\star (\boldsymbol{x} - \widetilde{\boldsymbol{x}}) + \eta (\boldsymbol{M}^\circ - \boldsymbol{M}^\star) \boldsymbol{x}$

$= \int_0^1 (\boldsymbol{I} - \eta \nabla^2 g(\boldsymbol{x}(\tau))(\boldsymbol{x} - \widetilde{\boldsymbol{x}}) \, \mathrm{d}\tau - \eta \|\widetilde{\boldsymbol{x}}\|_2^2 \left( \boldsymbol{I}_{\widetilde{\boldsymbol{x}}} - \boldsymbol{I} \right) \widetilde{\boldsymbol{x}} + \eta \boldsymbol{M}^\star (\boldsymbol{x} - \widetilde{\boldsymbol{x}}) + \eta (\boldsymbol{M}^\circ - \boldsymbol{M}^\star) \boldsymbol{x}$

$= \underbrace{\left( (1 - \eta \|\widetilde{\boldsymbol{x}}\|_2^2) \boldsymbol{I} - 2\eta \widetilde{\boldsymbol{x}} \widetilde{\boldsymbol{x}}^\top + \eta \boldsymbol{M}^\star \right) (\boldsymbol{x} - \widetilde{\boldsymbol{x}})}_{\textcircled{1}} - \eta \underbrace{\int_0^1 \left( \nabla^2 g(\boldsymbol{x}(\tau)) - \left( \|\widetilde{\boldsymbol{x}}\|_2^2 \boldsymbol{I} + 2\widetilde{\boldsymbol{x}} \widetilde{\boldsymbol{x}}^\top \right) \right) (\boldsymbol{x} - \widetilde{\boldsymbol{x}}) \mathrm{d}\tau}_{\textcircled{2}}$

$- \eta \underbrace{\|\widetilde{\boldsymbol{x}}\|_2^2 \left( \boldsymbol{I}_{\widetilde{\boldsymbol{x}}} - \boldsymbol{I} \right) \widetilde{\boldsymbol{x}}}_{\textcircled{3}} + \eta \underbrace{(\boldsymbol{M}^\circ - \boldsymbol{M}^\star) \boldsymbol{x}}_{\textcircled{4}},$

where $\boldsymbol{x}(\tau) = \widetilde{\boldsymbol{x}} + \tau (\boldsymbol{x} - \widetilde{\boldsymbol{x}})$. For the term $\textcircled{1}$, we require a bound on

$$\left\| (1 - \eta \|\widetilde{\boldsymbol{x}}\|_2^2) \boldsymbol{I} - 2\eta \widetilde{\boldsymbol{x}} \widetilde{\boldsymbol{x}}^\top + \eta \boldsymbol{M}^\star \right\|.$$

If we write $\widetilde{\boldsymbol{x}}$ as $\widetilde{\alpha}_t \boldsymbol{u}^\star + \widetilde{\boldsymbol{x}}_\perp$, we have $\widetilde{\alpha}_t^2 \leq \lambda^\star$ and $\|\widetilde{\boldsymbol{x}}_\perp\|_2 \lesssim \beta_0$. Then, we have

$$\left\| (1 - \eta \|\widetilde{\boldsymbol{x}}\|_2^2) \boldsymbol{I} - 2\eta \widetilde{\boldsymbol{x}} \widetilde{\boldsymbol{x}}^\top + \eta \boldsymbol{M}^\star \right\|$$

$$= \left\| (1 - \eta \|\widetilde{\boldsymbol{x}}\|_2^2) \boldsymbol{I} + \eta (\lambda^\star - 2\widetilde{\alpha}_t^2) \boldsymbol{u}^\star \boldsymbol{u}^{\star\top} - 2\eta (\widetilde{\boldsymbol{x}} \widetilde{\boldsymbol{x}}^\top - \widetilde{\alpha}_t^2 \boldsymbol{u}^\star \boldsymbol{u}^{\star\top}) \right\|$$

$$\leq \left\| (1 - \eta \|\widetilde{\boldsymbol{x}}\|_2^2) \boldsymbol{I} + \eta (\lambda^\star - 2\widetilde{\alpha}_t^2) \boldsymbol{u}^\star \boldsymbol{u}^{\star\top} \right\| + 2\eta \left\| \widetilde{\boldsymbol{x}} \widetilde{\boldsymbol{x}}^\top - \widetilde{\alpha}_t^2 \boldsymbol{u}^\star \boldsymbol{u}^{\star\top} \right\|$$

$$\leq (1 + \eta \lambda^\star) + 2\eta (2\alpha_t \|\widetilde{\boldsymbol{x}}_\perp\|_2 + \|\widetilde{\boldsymbol{x}}_\perp\|_2^2)$$

$$\leq 1 + \eta \lambda^\star + \frac{c_8}{\log^2 n}$$

for some universal constant $c_8 > 0$. This implies the desired bound

$$\|\textcircled{1}\|_2 \leq \left( 1 + \eta \lambda^\star + \frac{c_8}{\log^2 n} \right) \|\boldsymbol{x} - \widetilde{\boldsymbol{x}}\|_2.$$

For all $0 \leq \tau \leq 1$, we have $\|\boldsymbol{x}(\tau) - \widetilde{\boldsymbol{x}}\|_\infty \leq \|\boldsymbol{x} - \widetilde{\boldsymbol{x}}\|_\infty$, and the induction hypothesis (E.2) gives

$$\|\boldsymbol{x} - \widetilde{\boldsymbol{x}}\|_\infty \lesssim \sqrt{\frac{1}{\mu \log^{13} n}} (1 + \eta \lambda^\star)^{T_2} \frac{\beta_0}{n}.$$

Hence, by Lemma G.10, we have

$$\left\| \nabla^2 g(\boldsymbol{x}(\tau)) - \left( \|\widetilde{\boldsymbol{x}}\|_2^2 \boldsymbol{I} + 2\widetilde{\boldsymbol{x}} \widetilde{\boldsymbol{x}}^\top \right) \right\| \lesssim n \|\boldsymbol{x} - \widetilde{\boldsymbol{x}}\|_\infty (\|\boldsymbol{x}\|_\infty + \|\widetilde{\boldsymbol{x}}\|_\infty) + \sqrt{\frac{n \log n}{p}} \|\widetilde{\boldsymbol{x}}\|_\infty^2$$

$$\lesssim \left( \sqrt{\frac{1}{\log^{12} n}} + \sqrt{\frac{\mu^2 \log^3 n}{np}} \right) (1 + \eta \lambda^\star)^{2T_2} \frac{\beta_0^2}{n}$$

$$\lesssim \lambda^\star \sqrt{\frac{1}{\log^{12} n}}$$

if $n^2 p \gtrsim \mu^2 n \log^{15} n$ because $\|\widetilde{\boldsymbol{x}}\|_\infty \lesssim \sqrt{\mu \log n} (1 + \eta \lambda^\star)^{T_2} \frac{\beta_0}{n}$ and $(1 + \eta \lambda^\star)^{T_2} \lesssim \sqrt{\lambda^\star} \frac{\sqrt{n}}{\beta_0}$. This gives

$$\|\textcircled{2}\|_2 \lesssim \lambda^\star \sqrt{\frac{1}{\log^{12} n}} \|\boldsymbol{x} - \widetilde{\boldsymbol{x}}\|_2. \tag{E.6}$$

For the term $\text{\textcircled{3}}$, we use Lemma G.8 to obtain

$$\left\|\text{\textcircled{3}}\right\|_2 \lesssim \lambda^\star \sqrt{\frac{\mu \log n}{np}} \|\widetilde{\boldsymbol{x}}\|_2.$$

Lastly, the term $\text{\textcircled{4}}$ is bounded with

$$\left\|\text{\textcircled{4}}\right\|_2 \lesssim \|\boldsymbol{M}^\circ - \boldsymbol{M}^\star\| \|\boldsymbol{x}\|_2 \lesssim \lambda^\star \mu \sqrt{\frac{\log n}{np}} \|\widetilde{\boldsymbol{x}}\|_2.$$

Combining all, there exists a universal constant $c_9 > 0$ such that

$$\left\|\text{\textcircled{1}}\right\|_2 + \left\|\text{\textcircled{2}}\right\|_2 \le \left(1 + \eta\lambda^\star + \frac{c_9}{\log^2 n}\right) \|\boldsymbol{x} - \widetilde{\boldsymbol{x}}\|_2,$$

$$\eta\left(\left\|\text{\textcircled{3}}\right\|_2 + \left\|\text{\textcircled{4}}\right\|_2\right) \le c_9 \mu \sqrt{\frac{\log n}{np}} \|\widetilde{\boldsymbol{x}}\|_2.$$

Because $\left\|\boldsymbol{x}^{(T_1)}\right\|_2 \lesssim \sqrt{\log n}\beta_0$ by (C.3) and $\left\|\widetilde{\boldsymbol{x}}^{(t)}\right\|_2$ can grow at a rate at most $(1 + \eta\lambda^\star)$, there exists a universal constant $c_{10} > 0$ such that

$$c_8 \mu \sqrt{\frac{\log n}{np}} \left\|\widetilde{\boldsymbol{x}}^{(t)}\right\|_2 \le c_{10} \mu \sqrt{\frac{\log^2 n}{np}} (1 + \eta\lambda^\star)^{t - T_1} \beta_0. \tag{E.7}$$

Hence, for all $T_1 \le s \le t$, we have

$$\left\|\boldsymbol{x}^{(s+1)} - \widetilde{\boldsymbol{x}}^{(s+1)}\right\|_2 \le \left(1 + \eta\lambda^\star + \frac{c_9}{\log^2 n}\right) \left\|\boldsymbol{x}^{(s)} - \widetilde{\boldsymbol{x}}^{(s)}\right\|_2 + c_{10} \mu \sqrt{\frac{\log^2 n}{np}} (1 + \eta\lambda^\star)^{t - T_1} \beta_0.$$

An analysis on the recursive equation

$$x_{s+1} = \left(1 + \eta\lambda^\star + \frac{c_9}{\log^2 n}\right) x_s + c_{10} \mu \sqrt{\frac{\log^2 n}{np}} (1 + \eta\lambda^\star)^{t - T_1} \beta_0, \quad x_{T_1} = c_7 \mu \sqrt{\frac{\log^3 n}{np}} \beta_0$$

proves that

$$\left\|\boldsymbol{x}^{(t+1)} - \widetilde{\boldsymbol{x}}^{(t+1)}\right\|_2 \le 2c_7 \mu \sqrt{\frac{\log^3 n}{np}} \beta_0 (1 + \eta\lambda^\star)^{t+1 - T_1}.$$

**(E.3) at $(t + 1)$**  Similar to the proof of (D.8), we have the decomposition

$$\boldsymbol{x}^{(t+1)} - \boldsymbol{x}^{(t+1,l)} = \underbrace{(1 - \eta\|\widetilde{\boldsymbol{x}}\|_2^2 - 2\eta\widetilde{\boldsymbol{x}}\widetilde{\boldsymbol{x}}^\top)(\boldsymbol{x} - \boldsymbol{x}^{(l)})}_{\text{\textcircled{1}}}$$

$$- \eta \underbrace{\int_0^1 \left(\nabla^2 g(\boldsymbol{x}^{(l)}(\tau)) - \left(\|\widetilde{\boldsymbol{x}}\|_2^2 \boldsymbol{I} + 2\widetilde{\boldsymbol{x}}\widetilde{\boldsymbol{x}}^\top\right)\right)(\boldsymbol{x} - \boldsymbol{x}^{(l)})\mathrm{d}\tau}_{\text{\textcircled{2}}}$$

$$- \eta \underbrace{\left(\frac{1}{p}\mathcal{P}_{\Omega_l}\left(\boldsymbol{x}^{(l)}\boldsymbol{x}^{(l)\top}\right) - \mathcal{P}_l\left(\boldsymbol{x}^{(l)}\boldsymbol{x}^{(l)\top}\right)\right)\boldsymbol{x}^{(l)}}_{\text{\textcircled{3}}}$$

$$+ \eta\boldsymbol{M}^\star(\boldsymbol{x} - \boldsymbol{x}^{(l)}) + \eta \underbrace{(\boldsymbol{M}^\circ - \boldsymbol{M}^\star)(\boldsymbol{x} - \boldsymbol{x}^{(l)})}_{\text{\textcircled{4}}}$$

$$+ \eta \underbrace{\left(\frac{1}{p}\mathcal{P}_\Omega(\boldsymbol{M}^\star) - \mathcal{P}_\Omega^{(l)}(\boldsymbol{M}^\star)\right)\boldsymbol{x}^{(l)}}_{\text{\textcircled{5}}} + \eta \underbrace{\left(\frac{1}{p}\mathcal{P}_\Omega(\boldsymbol{E}) - \boldsymbol{E}^{(l)}\right)\boldsymbol{x}^{(l)}}_{\text{\textcircled{6}}},$$

where $\boldsymbol{x}^{(l)}(\tau) = \boldsymbol{x}^{(l)} + \tau(\boldsymbol{x} - \boldsymbol{x}^{(l)})$. Both of the terms ① and ② can be bounded similar to ① and ② of $\boldsymbol{x}^{(t+1)} - \widetilde{\boldsymbol{x}}^{(t+1)}$ as

$$\left\| ① \right\|_2 \le \left( 1 + \eta\lambda^\star + \frac{c_{11}}{\log^2 n} \right) \left\| \boldsymbol{x} - \boldsymbol{x}^{(l)} \right\|_2,$$

$$\left\| ② \right\|_2 \lesssim \lambda^\star \sqrt{\frac{1}{\log^{12} n}} \left\| \boldsymbol{x} - \boldsymbol{x}^{(l)} \right\|_2$$

for some universal constant $c_{11} > 0$. For the terms ③ and ⑤, we use Lemma G.11 to obtain

$$\left\| ③ \right\|_2 \lesssim \sqrt{\frac{\log n}{p}} \left\| \boldsymbol{x}^{(l)} \right\|_2 \left\| \boldsymbol{x}^{(l)} \right\|_\infty^2 \lesssim \lambda^\star \mu \sqrt{\frac{\log n}{np}} \frac{1}{\sqrt{n}} \| \widetilde{\boldsymbol{x}} \|_2,$$

$$\left\| ⑤ \right\|_2 \lesssim \lambda^\star \mu \sqrt{\frac{\log n}{np}} \frac{1}{\sqrt{n}} \| \widetilde{\boldsymbol{x}} \|_2,$$

and use Lemma G.13 to obtain

$$\left\| ⑥ \right\|_2 \lesssim \lambda^\star \mu \sqrt{\frac{\log^2 n}{np}} \frac{1}{\sqrt{n}} \| \widetilde{\boldsymbol{x}} \|_2.$$

From Lemmas A.1 and A.3, the term ④ is bounded as

$$\left\| ④ \right\|_2 \le \| \boldsymbol{M}^\circ - \boldsymbol{M}^\star \| \left\| \boldsymbol{x} - \boldsymbol{x}^{(l)} \right\|_2 \lesssim \lambda^\star \mu \sqrt{\frac{\log n}{np}} \left\| \boldsymbol{x} - \boldsymbol{x}^{(l)} \right\|_2.$$

Combining all with (E.7), there exists a universal constant $c_{12} > 0$ such that

$$\left\| ① \right\|_2 + \left\| ② \right\|_2 + \left\| ④ \right\|_2 \le \left( 1 + \eta\lambda^\star + \frac{c_{12}}{\log^2 n} \right) \| \boldsymbol{x} - \widetilde{\boldsymbol{x}} \|_2,$$

$$\eta \left( \left\| ③ \right\|_2 + \left\| ⑤ \right\|_2 + \left\| ⑥ \right\|_2 \right) \le c_{12} \mu \sqrt{\frac{\log^3 n}{np}} \frac{\beta_0}{\sqrt{n}} (1 + \eta\lambda^\star)^{t-T_1}.$$

Hence, we have

$$\left\| \boldsymbol{x}^{(s+1)} - \boldsymbol{x}^{(s+1,l)} \right\|_2 \le \left( 1 + \eta\lambda^\star + \frac{c_{12}}{\log^2 n} \right) \left\| \boldsymbol{x}^{(s)} - \boldsymbol{x}^{(s,l)} \right\|_2 + c_{12} \mu \sqrt{\frac{\log^3 n}{np}} \frac{\beta_0}{\sqrt{n}} (1 + \eta\lambda^\star)^{t-T_1},$$

for all $T_1 \le s \le t$. An analysis on the recursive equation

$$x_{s+1} = \left( 1 + \eta\lambda^\star + \frac{c_{12}}{\log^2 n} \right) x_s + c_{12} \mu \sqrt{\frac{\log^3 n}{np}} \frac{\beta_0}{\sqrt{n}} (1 + \eta\lambda^\star)^{t-T_1}, \quad x_{T_1} = c_7 \mu \sqrt{\frac{\log^4 n}{np}} \frac{\beta_0}{\sqrt{n}}$$

proves that

$$\left\| \boldsymbol{x}^{(t+1)} - \boldsymbol{x}^{(t+1,l)} \right\|_2 \le 2 \left( c_{12} \mu \sqrt{\frac{\log^3 n}{np}} (t + 1 - T_1) + c_7 \mu \sqrt{\frac{\log^4 n}{np}} \right) \frac{\beta_0}{\sqrt{n}} (1 + \eta\lambda^\star)^{t+1-T_1}$$

$$\le c_{13} \mu \sqrt{\frac{\log^5 n}{np}} \frac{\beta_0}{\sqrt{n}} (1 + \eta\lambda^\star)^{t+1-T_1}$$

holds for some universal constant $c_{13} > 0$ because $T_2 - T_1 \lesssim \log n$.

**(E.4) at $(t+1)$**   We use the same bound

$$\left| (\boldsymbol{x}^{(t+1,l)} - \widetilde{\boldsymbol{x}}^{(t+1)})_l \right| \le \left( 1 - \eta\|\widetilde{\boldsymbol{x}}\|_2^2 \right) \left| (\boldsymbol{x}^{(l)} - \widetilde{\boldsymbol{x}})_l \right| + \eta(\lambda^\star \|\boldsymbol{u}^\star\|_\infty + \|\widetilde{\boldsymbol{x}}\|_2 \|\widetilde{\boldsymbol{x}}\|_\infty) \left\| \boldsymbol{x}^{(l)} - \widetilde{\boldsymbol{x}} \right\|_2.$$

that was used in the proof of (D.10). From (E.1) and (E.3), we have

$$\left\| \boldsymbol{x}^{(l)} - \widetilde{\boldsymbol{x}} \right\|_2 \leq \left\| \boldsymbol{x}^{(l)} - \boldsymbol{x} \right\|_2 + \| \boldsymbol{x} - \widetilde{\boldsymbol{x}} \|_2 \leq 3c_7 \mu \sqrt{\frac{\log^5 n}{np}} \beta_0 (1 + \eta \lambda^\star)^{t - T_1}.$$

Combined with the fact that $\lambda^\star \| \boldsymbol{u}^\star \|_\infty + \| \widetilde{\boldsymbol{x}} \|_2 \| \widetilde{\boldsymbol{x}} \|_\infty \leq 3\lambda^\star \sqrt{\frac{\mu}{n}}$, there exists a universal constant $c_{14} > 0$ such that

$$\eta (\lambda^\star \| \boldsymbol{u}^\star \|_\infty + \| \widetilde{\boldsymbol{x}} \|_2 \| \widetilde{\boldsymbol{x}} \|_\infty) \left\| \boldsymbol{x}^{(l)} - \widetilde{\boldsymbol{x}} \right\|_2 \leq c_{14} \sqrt{\frac{\mu^3 \log^5 n}{np}} \frac{\beta_0}{\sqrt{n}} (1 + \eta \lambda^\star)^{t - T_1}.$$

Hence, for all $T_1 \leq s \leq t$, we have

$$\left| (\boldsymbol{x}^{(s+1,l)} - \widetilde{\boldsymbol{x}}^{(s+1)})_l \right| \leq \left| (\boldsymbol{x}^{(s,l)} - \widetilde{\boldsymbol{x}}^{(s)})_l \right| + c_{14} \sqrt{\frac{\mu^3 \log^5 n}{np}} \frac{\beta_0}{\sqrt{n}} (1 + \eta \lambda^\star)^{t - T_1},$$

and this implies

$$\left| (\boldsymbol{x}^{(t+1,l)} - \widetilde{\boldsymbol{x}}^{(t+1)})_l \right| \leq c_{14} \sqrt{\frac{\mu^3 \log^5 n}{np}} \frac{\beta_0}{\sqrt{n}} \sum_{s=T_1}^{t} (1 + \eta \lambda^\star)^{s - T_1} + c_7 \sqrt{\frac{\mu^3 \log^8 n}{np}} \frac{\beta_0}{\sqrt{n}}$$

$$\leq 2c_7 \sqrt{\frac{\mu^3 \log^8 n}{np}} \frac{\beta_0}{\sqrt{n}} (1 + \eta \lambda^\star)^{t+1 - T_1}.$$

**(E.2) at $(t+1)$**  The $l$th component of $\boldsymbol{x}^{(t+1)} - \widetilde{\boldsymbol{x}}^{(t+1)}$ is bounded by

$$\left| (\boldsymbol{x}^{(t+1)} - \widetilde{\boldsymbol{x}}^{(t+1)})_l \right| \leq \left\| \boldsymbol{x}^{(t+1)} - \boldsymbol{x}^{(t+1,l)} \right\|_\infty + \left| (\boldsymbol{x}^{(t+1,l)} - \widetilde{\boldsymbol{x}}^{(t+1)})_l \right|$$

$$\leq \left\| \boldsymbol{x}^{(t+1)} - \boldsymbol{x}^{(t+1,l)} \right\|_2 + \left| (\boldsymbol{x}^{(t+1,l)} - \widetilde{\boldsymbol{x}}^{(t+1)})_l \right|$$

$$\leq c_{13} \mu \sqrt{\frac{\log^5 n}{np}} \frac{\beta_0}{\sqrt{n}} (1 + \eta \lambda^\star)^{t+1 - T_1} + 2c_7 \sqrt{\frac{\mu^3 \log^8 n}{np}} \frac{\beta_0}{\sqrt{n}} (1 + \eta \lambda^\star)^{t+1 - T_1}$$

$$\leq 3c_7 \sqrt{\frac{\mu^3 \log^8 n}{np}} \frac{\beta_0}{\sqrt{n}} (1 + \eta \lambda^\star)^{t+1 - T_1}.$$

## F  Fixed Initialization Size

In Section 3, we claimed that the estimation error is improved to $\frac{1}{\sqrt{np}} + \frac{\sigma}{\lambda^\star} \sqrt{\frac{n}{p}}$ if the initialization size is fixed to $n^{-1/4}$ regardless of the sample complexity. We briefly discuss how the proofs should change in such a case. For clear presentation, $\mu$ and $\log n$ factors are ignored in this section.

For every bound of Phase I (Lemmas 5.1 to 5.5), $\frac{1}{\sqrt{np}}$ is changed to $\frac{1}{\sqrt{np}} + \frac{\sigma}{\lambda^\star} \sqrt{\frac{n}{p}}$, while allowing $\sigma$ to be as large as $\frac{\lambda^\star \mu}{n} \sqrt{np}$. More importantly, the definition of Phase I is changed to be the largest $t$ such that $(1 + \eta \lambda^\star)^t \leq \sqrt{n}$, so it is lengthened by $\sqrt{np}$ times than before. In the original proof, the estimation error of $\frac{1}{\sqrt{np}}$ obtained at the end of Phase I was increased to $\frac{1}{\text{poly}(\log n)}$ during the first part of Phase II. However, if $(1 + \eta \lambda^\star)^t$ equals $\sqrt{n}$ at the end of Phase I, we do not have such a part in Phase II, and the estimation error obtained at the end of Phase I is maintained through Phase II.

## G  Technical Lemmas

We introduce some technical lemmas in this section. Most of them are the results of classical concentration inequalities.

**Theorem G.1** (Matrix Bernstein Inequality). *Let $\{\boldsymbol{X}_i\}$ be $n \times n$ independent symmetric random matrices. Assume that each random matrix satisfies $\mathbb{E}\,\boldsymbol{X}_i = \boldsymbol{0}$ and $\|\boldsymbol{X}_i\| \leq L$ almost surely. Then, for all $\tau \geq 0$, we have*

$$\mathbb{P}\left[\left\|\sum_i \boldsymbol{X}_i\right\| \geq \tau\right] \leq n\exp\left(\frac{-\tau^2/2}{V + L\tau/3}\right),$$

*where $V = \left\|\sum_i \mathbb{E}(\boldsymbol{X}_i^2)\right\|$.*

**Corollary G.2** (Matrix Bernstein Inequality). *Let $\{\boldsymbol{X}_i\}$ be $n \times n$ independent symmetric random matrices. Assume that each random matrix satisfies $\mathbb{E}\,\boldsymbol{X}_i = \boldsymbol{0}$ and $\|\boldsymbol{X}_i\| \leq L$ almost surely. Then, with high probability, we have*

$$\left\|\sum_i \boldsymbol{X}_i\right\| \lesssim \sqrt{V\log n} + L\log n,$$

*where $V = \left\|\sum_i \mathbb{E}(\boldsymbol{X}_i^2)\right\|$.*

**Lemma G.3.** *For any fixed matrix $\boldsymbol{M} \in \mathbb{R}^{n \times n}$, we have*

$$\left\|\frac{1}{p}\mathcal{P}_\Omega(\boldsymbol{M}) - \boldsymbol{M}\right\| \lesssim \sqrt{\frac{n\log n}{p}}\|\boldsymbol{M}\|_\infty + \frac{\log n}{p}\|\boldsymbol{M}\|_\infty$$

*with high probability.*

*Proof.* We decompose the matrix into the sum of independent symmetric matrices.

$$\frac{1}{p}\mathcal{P}_\Omega(\boldsymbol{M}) - \boldsymbol{M} = \sum_{i<j}\left(\frac{\delta_{ij}}{p} - 1\right)M_{ij}(\boldsymbol{e}_i\boldsymbol{e}_j^\top + \boldsymbol{e}_j\boldsymbol{e}_i^\top) + \sum_i\left(\frac{\delta_{ii}}{p} - 1\right)M_{ii}\boldsymbol{e}_i\boldsymbol{e}_i^\top$$

We calculate $L$ and $V$ of Corollary G.2. We have $L \leq \frac{1}{p}\|\boldsymbol{M}\|_\infty$ because

$$\left\|\left(\frac{\delta_{ij}}{p} - 1\right)M_{ij}(\boldsymbol{e}_i\boldsymbol{e}_j^\top + \boldsymbol{e}_j\boldsymbol{e}_i^\top)\right\| \leq \frac{1}{p}\|\boldsymbol{M}\|_\infty,$$

$$\left\|\left(\frac{\delta_{ii}}{p} - 1\right)M_{ii}\boldsymbol{e}_i\boldsymbol{e}_i^\top\right\| \leq \frac{1}{p}\|\boldsymbol{M}\|_\infty.$$

We also have the following bound on $V$.

$$V = \frac{1-p}{p}\left\|\sum_{i,j}M_{ij}^{\star 2}\boldsymbol{e}_i\boldsymbol{e}_i^\top\right\| \leq \frac{n}{p}\|\boldsymbol{M}\|_\infty^2$$

Hence, Corollary G.2 implies the desired result. $\qquad\square$

We can prove Lemma A.1 by applying Lemma G.3 to $\boldsymbol{M}^\star$ and using $\|\boldsymbol{M}^\star\|_\infty = \lambda^\star\frac{\mu}{n}$.

We introduce classical Bernstein inequality and the results obtained from it.

**Theorem G.4** (Bernstein Inequality). *Let $\{X_i\}$ be independent random variables. Assume that each random variable satisfies $\mathbb{E}\,X_i = 0$ and $|X_i| \leq L$ almost surely. Then, for all $\tau \geq 0$, we have*

$$\mathbb{P}\left[\left|\sum_i X_i\right| \geq \tau\right] \leq 2\exp\left(\frac{-\tau^2/2}{V + L\tau/3}\right),$$

*where $V = \sum_i \mathbb{E}[X_i^2]$.*

**Corollary G.5** (Bernstein Inequality). *Let $\{X_i\}$ be independent random variables. Assume that each random variable satisfies $\mathbb{E}\,X_i = 0$ and $|X_i| \leq L$ almost surely. Then, with high probability, we have*

$$\left|\sum_i X_i\right| \lesssim \sqrt{V\log n} + L\log n,$$

*where $V = \sum_i \mathbb{E}[X_i^2]$.*

**Lemma G.6.** *Let and $\{X_i\}$ be independent Bernoulli random variables with expectation $p$. Then, for any fixed vector $\mathbf{a}$, we have*

$$\left|\sum_i \left(\frac{X_i}{p} - 1\right) a_i\right| \lesssim \sqrt{\frac{\log n}{p}} \|\mathbf{a}\|_2 + \frac{\log n}{p} \|\mathbf{a}\|_\infty$$

*with high probability.*

*Proof.* We can apply Corollary G.5 with $L = \frac{1}{p}\|\mathbf{a}\|_\infty$ and $V = \frac{1-p}{p}\|\mathbf{a}\|_2^2$. $\qquad\qquad\square$

**Lemma G.7.** *If $n^2 p \gtrsim \mu n \log n$, we have*

$$\left\|\frac{1}{p}\mathcal{P}_\Omega(\mathbf{M}^\star)\right\|_{2,\infty} \lesssim \lambda^\star \sqrt{\frac{\mu}{np}}$$

*with high probability.*

*Proof.* Let us consider $\ell_2$-norm of the $i$th row of $\mathbf{M}^\circ$.

$$\left\|\left(\frac{1}{p}\mathcal{P}_\Omega(\mathbf{M}^\star)\right)_{i*}\right\|_2^2 = \lambda^{\star 2} u_i^{\star 2} \sum_j \frac{1}{p^2} \delta_{ij} u_j^{\star 2}$$

$$\leq \frac{1}{p}\lambda^{\star 2}\|\mathbf{u}^\star\|_\infty^2 \left(\|\mathbf{u}^\star\|_2^2 + \left(\sum_j \frac{1}{p}\delta_{ij}u_j^{\star 2} - \|\mathbf{u}^\star\|_2^2\right)\right)$$

$$\lesssim \frac{\lambda^{\star 2}\mu}{np}\left(1 + \sqrt{\frac{\log n}{np}}\right) \lesssim \frac{\lambda^{\star 2}\mu}{np}$$

The third line follows from Lemma G.6. $\qquad\qquad\square$

*Proof of Lemma A.2.* The spectral norm of a symmetric matrix that has nonzero entries only on the $l$th row/column is bounded by twice of the norm of its $l$th row. Hence,

$$\left\|\frac{1}{p}\mathcal{P}_\Omega(\mathbf{M}^\star) - \mathcal{P}_\Omega^{(l)}(\mathbf{M}^\star)\right\| \leq 2\left\|\left(\frac{1}{p}\mathcal{P}_\Omega(\mathbf{M}^\star) - \mathcal{P}_\Omega^{(l)}(\mathbf{M}^\star)\right)_{l*}\right\|_2 = 2\left\|\left(\frac{1}{p}\mathcal{P}_\Omega(\mathbf{M}^\star) - \mathbf{M}^\star\right)_{l*}\right\|_2$$

$$\lesssim \left\|\frac{1}{p}\mathcal{P}_\Omega(\mathbf{M}^\star)\right\|_{2,\infty} + \|\mathbf{M}^\star\|_{2,\infty} \lesssim \lambda^\star\sqrt{\frac{\mu}{np}},$$

where the last inequality follows from Lemma G.7. $\qquad\qquad\square$

**Lemma G.8.** *Let $\mathbf{y}$ be a vector that is independent from the sampling. Then, if $n^2 p \gtrsim n \log n$, we have*

$$\max_{i\in[n]}\left|\|\mathbf{x}\|_{2,i}^2 - \|\mathbf{y}\|_2^2\right| \lesssim n\|\mathbf{x}-\mathbf{y}\|_\infty(\|\mathbf{x}\|_\infty + \|\mathbf{y}\|_\infty) + \sqrt{\frac{\log n}{p}}\|\mathbf{y}\|_2\|\mathbf{y}\|_\infty + \frac{\log n}{p}\|\mathbf{y}\|_\infty^2$$

*with very high probability.*

*Proof.* Let us fix $i$ and decompose the difference as

$$\|\mathbf{x}\|_{2,i}^2 - \|\mathbf{y}\|_2^2 = \frac{1}{p}\sum_{j=1}^n \delta_{ij}(x_j^2 - y_j^2) + \sum_{j=1}^n \left(\frac{\delta_{ij}}{p} - 1\right)y_j^2.$$

The first term is bounded as

$$\left|\frac{1}{p}\sum_{j=1}^n \delta_{ij}(x_j^2 - y_j^2)\right| \leq \|\mathbf{x}-\mathbf{y}\|_\infty\|\mathbf{x}+\mathbf{y}\|_\infty\frac{1}{p}\sum_{j=1}^n \delta_{ij} \lesssim n\|\mathbf{x}-\mathbf{y}\|_\infty\|\mathbf{x}+\mathbf{y}\|_\infty,$$

and the second term is bounded as

$$\left| \sum_{j=1}^{n} \left( \frac{\delta_{ij}}{p} - 1 \right) y_j^2 \right| \lesssim \sqrt{\frac{\log n}{p}} \|\boldsymbol{y}\|_2 \|\boldsymbol{y}\|_\infty + \frac{\log n}{p} \|\boldsymbol{y}\|_\infty^2$$

by Lemma G.6. □

**Lemma G.9.** *Let $\boldsymbol{y}$ be a vector that is independent from the sampling. Then, if $n^2 p \gtrsim n \log n$, we have*

$$\left\| \frac{1}{p} \mathcal{P}_\Omega \left( \boldsymbol{x}\boldsymbol{x}^\top \right) - \boldsymbol{y}\boldsymbol{y}^\top \right\| \lesssim n \|\boldsymbol{x} - \boldsymbol{y}\|_\infty (\|\boldsymbol{x}\|_\infty + \|\boldsymbol{y}\|_\infty) + \sqrt{\frac{n \log n}{p}} \|\boldsymbol{y}\|_\infty^2$$

*with very high probability.*

*Proof.* We have the following sequence of inequalities

$$\left\| \frac{1}{p} \mathcal{P}_\Omega \left( \boldsymbol{x}\boldsymbol{x}^\top \right) - \boldsymbol{y}\boldsymbol{y}^\top \right\| \le \left\| \frac{1}{p} \mathcal{P}_\Omega \left( \boldsymbol{x}\boldsymbol{x}^\top \right) - \frac{1}{p} \mathcal{P}_\Omega \left( \boldsymbol{y}\boldsymbol{y}^\top \right) \right\| + \left\| \frac{1}{p} \mathcal{P}_\Omega \left( \boldsymbol{y}\boldsymbol{y}^\top \right) - \boldsymbol{y}\boldsymbol{y}^\top \right\|$$

$$\le \|\boldsymbol{x}\boldsymbol{x}^\top - \boldsymbol{y}\boldsymbol{y}^\top\|_\infty \left\| \frac{1}{p} \mathcal{P}_\Omega \left( \boldsymbol{1}\boldsymbol{1}^\top \right) \right\| + \left\| \frac{1}{p} \mathcal{P}_\Omega \left( \boldsymbol{y}\boldsymbol{y}^\top \right) - \boldsymbol{y}\boldsymbol{y}^\top \right\|$$

$$\lesssim \|\boldsymbol{x} - \boldsymbol{y}\|_\infty (\|\boldsymbol{x}\|_\infty + \|\boldsymbol{y}\|_\infty) \left\| \frac{1}{p} \mathcal{P}_\Omega \left( \boldsymbol{1}\boldsymbol{1}^\top \right) \right\| + \left\| \frac{1}{p} \mathcal{P}_\Omega \left( \boldsymbol{y}\boldsymbol{y}^\top \right) - \boldsymbol{y}\boldsymbol{y}^\top \right\|$$

$$\lesssim n \|\boldsymbol{x} - \boldsymbol{y}\|_\infty (\|\boldsymbol{x}\|_\infty + \|\boldsymbol{y}\|_\infty) + \sqrt{\frac{n \log n}{p}} \|\boldsymbol{y}\|_\infty^2,$$

where the second line is derived from a basic inequality $\|\boldsymbol{A}\| \le \||\boldsymbol{A}|\|$ that holds for any matrix $\boldsymbol{A}$, and the last line follows by applying Lemma G.3 to $\boldsymbol{1}\boldsymbol{1}^\top$ and $\boldsymbol{y}\boldsymbol{y}^\top$. □

**Lemma G.10.** *Let $\boldsymbol{y}$ be a vector that is independent from the sampling. Then, if $n^2 p \gtrsim n \log n$, we have*

$$\left\| \nabla^2 g(\boldsymbol{x}) - \left( \|\boldsymbol{y}\|_2^2 \boldsymbol{I} + 2\boldsymbol{y}\boldsymbol{y}^\top \right) \right\| \lesssim n \|\boldsymbol{x} - \boldsymbol{y}\|_\infty (\|\boldsymbol{x}\|_\infty + \|\boldsymbol{y}\|_\infty)$$

$$+ \sqrt{\frac{\log n}{p}} \|\boldsymbol{y}\|_2 \|\boldsymbol{y}\|_\infty + \frac{\log n}{p} \|\boldsymbol{y}\|_\infty^2 + \sqrt{\frac{n \log n}{p}} \|\boldsymbol{y}\|_\infty^2$$

*Proof.* This follows directly from Lemmas G.8 and G.9. □

Let us define an operator $\mathcal{P}_{\Omega_l}$ such that an entry of $\mathcal{P}_{\Omega_l}(\boldsymbol{X})$ is equal to that of $\boldsymbol{X}$ if it is contained both in the $l$th row/column and $\Omega$, and otherwise 0. We also define an operator $\mathcal{P}_l$ that makes the entries outside the $l$th row/column zero. Then, we have

$$\frac{1}{p} \mathcal{P}_\Omega(\boldsymbol{X}) - \mathcal{P}_\Omega^{(l)}(\boldsymbol{X}) = \frac{1}{p} \mathcal{P}_{\Omega_l}(\boldsymbol{X}) - \mathcal{P}_l(\boldsymbol{X}).$$

Also, note that

$$\frac{1}{p} \mathcal{P}_\Omega(\boldsymbol{E}) - \boldsymbol{E}^{(l)} = \frac{1}{p} \mathcal{P}_{\Omega_l}(E).$$

The following lemma was also introduced in [14], but we include the proof for completeness.

**Lemma G.11.** *Suppose that a matrix $\boldsymbol{M}$ and a vector $\boldsymbol{v}$ are independent from sampling of the $l$th row/column. If $n^2 p \gtrsim n \log n$, we have*

$$\left\| \left( \frac{1}{p} \mathcal{P}_{\Omega_l}(\boldsymbol{M}) - \mathcal{P}_l(\boldsymbol{M}) \right) \boldsymbol{v} \right\|_2 \lesssim \|\boldsymbol{M}\|_\infty \left( \sqrt{\frac{\log n}{p}} \|\boldsymbol{v}\|_2 + \frac{\log n}{p} \|\boldsymbol{v}\|_\infty + \sqrt{\frac{n}{p}} \|\boldsymbol{v}\|_\infty \right)$$

*with high probability.*

*Proof.* If we consider the contribution of $l$th term and the other terms separately, we have

$$\left\|\left(\frac{1}{p}\mathcal{P}_{\Omega_l}(\boldsymbol{M}) - \mathcal{P}_l(\boldsymbol{M})\right)\boldsymbol{v}\right\|_2 \leq \left|\sum_{j=1}^n \left(\frac{\delta_{lj}}{p} - 1\right) M_{lj}v_j\right| + |v_l|\sqrt{\sum_{i=1}^n \left(\frac{\delta_{il}}{p} - 1\right)^2 M_{il}^2}$$

$$\leq \|\boldsymbol{M}\|_\infty \left(\left|\sum_{j=1}^n \left(\frac{\delta_{lj}}{p} - 1\right)v_j\right| + \|\boldsymbol{v}\|_\infty \sqrt{\sum_{i=1}^n \left(\frac{\delta_{il}}{p} - 1\right)^2}\right)$$

From Lemma G.6, we have

$$\left|\sum_{j=1}^n \left(\frac{\delta_{lj}}{p} - 1\right)v_j\right| \lesssim \sqrt{\frac{\log n}{p}}\|\boldsymbol{v}\|_2 + \frac{\log n}{p}\|\boldsymbol{v}\|_\infty$$

with high probability. Regarding the second term, notice that

$$\sum_{i=1}^n \left(\frac{\delta_{il}}{p} - 1\right)^2 = n + \left(\frac{1}{p} - 2\right)\sum_{i=1}^n \frac{\delta_{il}}{p}.$$

Lemma G.6 implies that $\sum_{i=1}^n \frac{\delta_{il}}{p} \asymp n$ with high probability if $n^2 p \gtrsim n \log n$. Hence, we have

$$\sum_{i=1}^n \left(\frac{\delta_{il}}{p} - 1\right)^2 \lesssim \frac{n}{p},$$

and this finishes the proof. $\qquad\square$

**Lemma G.12.** *Let $\boldsymbol{M}$ be a matrix and $\boldsymbol{v}$, $\boldsymbol{w}$ be vectors that are independent from sampling of the $l$th row/column. Then, if $n^2 p \gtrsim n \log n$, we have*

$$\left|\boldsymbol{w}^\top \left(\frac{1}{p}\mathcal{P}_{\Omega_l}(\boldsymbol{M}) - \mathcal{P}_l(\boldsymbol{M})\right)\boldsymbol{v}\right|$$

$$\lesssim \|\boldsymbol{M}\|_\infty \left(\sqrt{\frac{\log n}{p}}(\|\boldsymbol{v}\|_2\|\boldsymbol{w}\|_\infty + \|\boldsymbol{w}\|_2\|\boldsymbol{v}\|_\infty) + \frac{\log n}{p}\|\boldsymbol{v}\|_\infty\|\boldsymbol{w}\|_\infty\right)$$

*Proof.* We can consider the $l$th row and column separately by

$$\left|\boldsymbol{w}^\top \left(\frac{1}{p}\mathcal{P}_{\Omega_l}(\boldsymbol{M}) - \mathcal{P}_l(\boldsymbol{M})\right)\boldsymbol{v}\right|$$

$$\leq \left|v_l \sum_i \left(\frac{\delta_{il}}{p} - 1\right) M_{il}w_i\right| + \left|w_l \sum_j \left(\frac{\delta_{lj}}{p} - 1\right) M_{lj}v_j\right| + \left|\left(\frac{\delta_{ll}}{p} - 1\right) M_{ll}v_l w_l\right|$$

$$\leq \|\boldsymbol{M}\|_\infty \left(\|\boldsymbol{v}\|_\infty \left|\sum_i \left(\frac{\delta_{il}}{p} - 1\right) w_i\right| + \|\boldsymbol{w}\|_\infty \left|\sum_j \left(\frac{\delta_{lj}}{p} - 1\right) v_j\right| + \frac{1}{p}\|\boldsymbol{v}\|_\infty\|\boldsymbol{w}\|_\infty\right)$$

If we apply Lemma G.6 to the summations, we get the desired result. $\qquad\square$

**Lemma G.13.** *Let $\boldsymbol{E}$ be a symmetric matrix whose upper and on diagonal entries are drawn from Gaussian distribution $\mathcal{N}(0, \sigma^2)$ independently. Let $\boldsymbol{v}$ be a vector that is independent from sampling of the $l$th row and column. Then, if $n^2 p \gtrsim n \log^2 n$, we have*

$$\left\|\frac{1}{p}\mathcal{P}_{\Omega_l}(\boldsymbol{E})\boldsymbol{v}\right\|_2 \lesssim \sigma \left(\sqrt{\frac{\log n}{p}}\|\boldsymbol{v}\|_2 + \frac{\sqrt{\log^3 n}}{p}\|\boldsymbol{v}\|_\infty + \sqrt{\frac{n}{p}}\|\boldsymbol{v}\|_\infty\right)$$

*Proof.* If we consider the contribution of $l$th term and the other terms separately, we have

$$\left\|\frac{1}{p}\mathcal{P}_{\Omega_l}(\boldsymbol{E})\boldsymbol{v}\right\|_2 \leq \frac{1}{p}\left|\sum_{j=1}^n \delta_{lj}E_{lj}v_j\right| + \frac{1}{p}|v_l|\sqrt{\sum_{i=1}^n \delta_{il}E_{il}^2}$$

For the first term, we will calculate $V$ and $L$ of Corollary G.5. $V$ is calculated as

$$V = \sum_{j=1}^{n} \mathbb{E}\big[(\delta_{lj} E_{lj} v_j)^2\big] = p\sigma^2 \|\boldsymbol{v}\|_2^2.$$

To find $L$, we first note that $\|\boldsymbol{E}_{l*}\|_\infty \lesssim \sigma\sqrt{\log n}$ with high probability, where $\boldsymbol{E}_{l*}$ is the $l$th row of $\boldsymbol{E}$. Thus, for all $j \in [n]$, we have

$$|\delta_{lj} E_{lj} v_j| \lesssim \sigma\sqrt{\log n}\|\boldsymbol{v}\|_\infty.$$

Corollary G.5 implies that the first term is bounded as

$$\frac{1}{p}\left|\sum_{j=1}^{n} \delta_{lj} E_{lj} v_j\right| \lesssim \sigma\left(\sqrt{\frac{\log n}{p}}\|\boldsymbol{v}\|_2 + \frac{\sqrt{\log^3 n}}{p}\|\boldsymbol{v}\|_\infty\right). \tag{G.1}$$

For the second term, it suffices to bound

$$\left|\sum_{i=1}^{n} \delta_{il}(E_{il}^2 - \sigma^2)\right|.$$

As before, we obtain $V$ and $L$ through

$$\sum_{i=1}^{n} \mathbb{E}\big[(\delta_{il}(E_{il}^2 - \sigma^2))^2\big] = p\sum_{i=1}^{n} \mathbb{E}\big[E_{il}^4 - 2\sigma^2 E_{il}^2 + \sigma^4\big] = 2\sigma^4 np,$$

$$\big|\delta_{il}(E_{il}^2 - \sigma^2)\big| \lesssim \sigma^2 \log n.$$

Corollary G.5 implies that

$$\left|\sum_{i=1}^{n} \delta_{il}(E_{il}^2 - \sigma^2)\right| \lesssim \sigma^2\left(\sqrt{np\log n} + \log^2 n\right).$$

Because $\sum_{i=1}^{n} \delta_{il} \asymp np$, we have

$$\sum_{i=1}^{n} \delta_{il} E_{il}^2 \lesssim \sigma^2\left(np + \sqrt{np\log n} + \log^2 n\right) \lesssim \sigma^2 np \tag{G.2}$$

if $n^2 p \gtrsim n\log^2 n$. Combining (G.1) and (G.2), we get the desired bound. $\qquad\square$

**Lemma G.14.** *Let $\boldsymbol{E}$ be a symmetric matrix whose upper and on diagonal entries are drawn from Gaussian distribution $\mathcal{N}(0, \sigma^2)$ independently. Let $\boldsymbol{v}, \boldsymbol{w}$ be vectors that are independent from sampling of the $l$th row and column. Then, if $n^2 p \gtrsim n\log n$, we have*

$$\frac{1}{p}|\boldsymbol{w}^\top \mathcal{P}_{\Omega_l}(\boldsymbol{E})\boldsymbol{v}| \lesssim \sigma\left(\sqrt{\frac{\log n}{p}}(\|\boldsymbol{v}\|_2\|\boldsymbol{w}\|_\infty + \|\boldsymbol{w}\|_2\|\boldsymbol{v}\|_\infty) + \frac{\sqrt{\log^3 n}}{p}\|\boldsymbol{v}\|_\infty\|\boldsymbol{w}\|_\infty\right).$$

*Proof.* We can consider the $l$th row and column separately by

$$\frac{1}{p}|\boldsymbol{w}^\top \mathcal{P}_{\Omega_l}(\boldsymbol{E})\boldsymbol{v}| \le \frac{1}{p}\left|v_l \sum_i \delta_{il} E_{il} w_i\right| + \frac{1}{p}\left|w_l \sum_j \delta_{lj} E_{lj} v_j\right| + \frac{1}{p}|\delta_{ll} E_{ll} v_l w_l|$$

$$\le \frac{\|\boldsymbol{v}\|_\infty}{p}\left|\sum_i \delta_{il} E_{il} w_i\right| + \frac{\|\boldsymbol{w}\|_\infty}{p}\left|\sum_j \delta_{lj} E_{lj} v_j\right| + \frac{1}{p}\|\boldsymbol{v}\|_\infty\|\boldsymbol{w}\|_\infty |E_{ll}|.$$

We bound the two summations similar to (G.1) and for the last term, we note that $|E_{ll}| \lesssim \sigma\sqrt{\log n}$ with high probability. $\qquad\square$

