# OpenReview forum: "Rank-1 Matrix Completion with Gradient Descent and Small Random Initialization"
_NeurIPS.cc/2023/Conference — NeurIPS 2023 poster_

### Official Review · Reviewer_zH8c · 2023-07-01

**Soundness:** 3 good
**Presentation:** 3 good
**Contribution:** 3 good
**Rating:** 6
**Confidence:** 4

**Summary:**

In this work, the authors proved the global convergence of the gradient descent algorithm with a small initialization for the rank-1 matrix completion problem.

**Strengths:**

The results in this work is novel and should be interesting to audiences in optimization and machine learning fields. This work shows that the incoherence regularizer may be unnecessary for the matrix completion problem.

**Weaknesses:**

A more detailed comparison with

Chen, J., Liu, D., & Li, X. (2020). Nonconvex Rectangular Matrix Completion via Gradient Descent Without ℓ₂,∞ Regularization. IEEE Transactions on Information Theory, 66(9), 5806-5841.

should be included. In addition, the importance of analyzing the rank-1 case should be discussed in more detail.

**Questions:**

(1) Line 55: I would suggest the authors to be more specific on what parameters the convergence time has a logarithmic dependence on.

(2) Line 78: it would be better to be consistent in using "GD" as the abbreviation of "gradient descent".

(3) Line 86: it would be better to explicitly mention that u^* is a vector.

(4) Line 102: I think the squared norm of x^0 is expected to be \beta_0^2. But I am not sure if the norm of x^0 is expected to be \beta_0.

(5) Line 128: I think it may be better to include the global convergence result in a formal theorem (instead of a remark).

(6) Line 143: besides the relation between T^* and \beta_0, I wonder if there is a reason why \beta_0 is lower bounded. It seems that the initialization size is not necessarily lower bounded in [17, 23]. It may be better to explain the reason why a lower bound is necessary.

(7) Line 154: the remark on the estimation error could also be formalized as a theorem.

(8) in Line 88, the authors mentioned that the incoherence at be at most poly(log n). But this condition is not included in Theorem 3.1. I wonder if this condition is necessary for the results.

(9) Line 170: please be more specific on the meaning of "incoherent up to a logarithmic factor".

(10) In my opinion, the discussion of proof ideas in Sections 4-5 is a little too long. It would be ideal if the length can be reduced by ~2 pages. With that said, I am okay with the current structure.

(11) Line 319: "an optimal number of samples" is confusing. Please consider using a different word.

(12) Another interesting open problem will be whether the results can be extended to the over-parameterized case, where a small initialization is also required.

(13) Besides the aforementioned problem, it would also be interesting to consider the asymmetric matrix completion problem; see the follow-up work:

Soltanolkotabi, M., Stöger, D., & Xie, C. (2023). Implicit balancing and regularization: Generalization and convergence guarantees for overparameterized asymmetric matrix sensing. arXiv preprint arXiv:2303.14244.

**Limitations:**

See my comments in the previous section.

---

> ### Author Rebuttal · Authors · 2023-08-09
>
> We sincerely thank the reviewer for the feedback. Please see our response to the reviewer’s question.
>
> >**A more detailed comparison with Chen, J., Liu, D., & Li, X. (2020) ... should be included.**
>
> The paper that reviewer mentioned is an extension of the local convergence result [a] to the asymmetric case and it reduces the required sample complexity by some log factors. If we apply the techniques developed in the paper, we may also reduce the sample complexity, but we are not sure about this at this moment because for the rank-1 case, many parts of the analysis in [a] are already simplified.
>
> ---
>
> >**In addition, the importance of analyzing the rank-1 case should be discussed in more detail.**
>
> The question of whether the combination of GD with random initialization can effectively solve a specific problem holds significant importance in the field of optimization. Phase retrieval and matrix sensing are one of the low-rank recovery problems, like matrix completion. For those problems, the global convergence of GD from a randomly initialized point has been proved. However, despite its similarity to matrix sensing, no analogous result had been established for matrix completion. The problem remains open especially after the local convergence result [a] published in 2017.
>
> Although the rank-one case does not have much impact in itself, it will provide a good starting point for someone who challenges the full problem. Our novel analysis on Phase I provides a good understanding on the dynamics of GD that starts from a small random initializer. We expect that our key lemmas such as Lemmas 5.2 and 5.5 will continue to hold for the general rank-r case in a similar way. For Phase II, we have a difficulty in analyzing the singular values of the trajectory, and it is discussed in Section 8 with some simulation results.
>
> ---
>
> >**(1) Line 55: I would suggest the authors to be more specific on what parameters the convergence time has a logarithmic dependence on. (2) Line 78: it would be better to be consistent in using "GD" as the abbreviation of "gradient descent". (3) Line 86: it would be better to explicitly mention that $u^{*}$ is a vector.**
>
> We will change the manuscript accordingly. We appreciate your careful review.
>
> ---
>
> >**(4) Line 102: I think the squared norm of $x^0$ is expected to be $\beta\_0^2$. But I am not sure if the norm of $x^0$ is expected to be $\beta\_0$.**
>
> Yes, the square norm is expected to be $\beta_0^2$, and the expectation of norm is not precisely $\beta_0$, but it is very close to $\beta_0$. We will correct this.
>
> ---
>
> >**(5) Line 128: I think it may be better to include the global convergence result in a formal theorem (instead of a remark).**
>
> We tried to emphasize our own contribution with the main theorem. We will state the global convergence result in a formal theorem in the supplementary due to lack of space.
>
> ---
>
> >**(6) Line 143: besides the relation between $T^{*}$ and $\beta_0$, I wonder if there is a reason why $\beta\_0$ is lower bounded. It seems that the initialization size is not necessarily lower bounded in [17, 23]. It may be better to explain the reason why a lower bound is necessary.**
>
> An exceptionally small initialization size such as $e^{-n}$ could be detrimental to the convergence of GD because it extends convergence time from $\Theta(\log n)$ to $\Theta(n)$. Because we are deriving probabilistic bounds for all iterations, establishing an upper bound on the iteration count is imperative. Similar bounds can also be found in previous work [a]. While imposing a limit on the number of iterations might seem counterintuitive when proving local convergence, Theorem 2 of reference [a] does precisely that by constraining the maximum iteration count to $O(n^5)$. Nonetheless, we can further reduce the lower bound $n^{-10}$ to $n^{-c}$ for any $c > 10$ by tuning some constant factors during the proof.
>
> ---
>
> >**(7) Line 154: the remark on the estimation error could also be formalized as a theorem.**
>
> The potential adjustments to the proof when initialization size is fixed is briefly discussed in Section F of the supplementary. We will include a formal theorem on this result in the supplementary in the final version.
>
> ---
>
> >**(8) in Line 88, the authors mentioned that the incoherence at be at most poly($\log n$). But this condition is not included in Theorem 3.1. I wonder if this condition is necessary for the results.**
>
> Yes, it is also assumed in Theorem 3.1. We will include the condition in Theorem 3.1.
>
> ---
>
> >**(9) Line 170: please be more specific on the meaning of "incoherent up to a logarithmic factor".**
>
> We will define the incoherence of a vector in Section 2.
>
> ---
>
> >**(10) In my opinion, the discussion of proof ideas in Sections 4-5 is a little too long. It would be ideal if the length can be reduced by ~2 pages. With that said, I am okay with the current structure.**
>
> We agree with this, but it would be hard to rewrite the sections in this submission.
>
> ---
>
> >**(11) Line 319: "an optimal number of samples" is confusing. Please consider using a different word.**
>
> We will mention the optimal number of samples as $n \mathrm{poly}(\log n)$.
>
> ---
>
> >**(12) Another interesting open problem will be whether the results can be extended to the over-parameterized case, where a small initialization is also required. (13) Besides the aforementioned problem, it would also be interesting to consider the asymmetric matrix completion problem; see the follow-up work: Soltanolkotabi, M., Stöger, D., & Xie, C. (2023). ...**
>
> We are aware of both directions, and we will mention them in Section 8 if space is allowed. We have tried for both directions, but it was not trivial to extend the current result to both cases.
>
> ---
>
> [a] Cong Ma et al. "Implicit Regularization in Nonconvex Statistical Estimation: Gradient Descent Converges Linearly for Phase Retrieval, Matrix Completion, and Blind Deconvolution."

---

> > ### Comment · Reviewer_zH8c · 2023-08-17
> >
> > I would like to thank the authors for the detailed response! I will increase my rating, but I think the structure of the paper should be improved. The global convergence results are more important and should be included in the main manuscript, potentially using the space by moving the proofs ideas in Sections 4-5 to the appendix.

---

### Official Review · Reviewer_96zD · 2023-07-04

**Soundness:** 4 excellent
**Presentation:** 4 excellent
**Contribution:** 3 good
**Rating:** 8
**Confidence:** 4

**Summary:**

The paper studies global convergence of GD (with a fixed step-size) for the rank-1 matrix completion problem (with symmetric and i.i.d. Bernoulli(p) observation and Gaussian noise on the entries) started from "small" random initialization. The authors prove that such vanilla GD, without any explicit regularization as commonly used in the literature, converges to the ground truth matrix in a polynomial time with near-optimal sample complexity. The result is interesting and the program is well-motivated. The analysis is mostly based on dynamical systems rather than optimization techniques, and a similar approach could be applied for related problems.

The proof idea seems to be novel (although some of the ideas seems to be motivated by recent literatures on GD with small initialization & stepwise for matrix factorization and matrix sensing problem). First, the authors first show that GD for the fully observed case with small initialization converges to the ground truth matrix (Corollary 1 in the appendix). This is possible due to an explicit formula for the GD iterates (a linear combination of the initialization and the ground truth with dynamic coefficients). The analysis is not very difficult, but also not so trivial.

Having established the desired convergence result for the fully observed GD dynamics, the main idea is to couple that with the partially observed GD dynamics by starting at the same initialization and use an interpolated dynamics eq. (13). The main novelty is in showing that these two trajectories remain close, and much closer than the norm of the iterates from the fully observed case. This allows one to show that the partially observed iterates converges to a local region near the ground truth after a polynomial number of iterations, and then one can use existing local convergence result [14].


**Strengths:**


The main text is exceptionally well-written (with minor comments/suggestions) in that it gives the structure of the proof of the main result in a very clear manner (which was a pleasure to read for the most part). The argument in the main text is well-complemented with various simulation results. The general rank-r case was also discussed at the end, and the main difficulty of having the r singular values possibly growing at different rates is well-pointed out. It seems that the appendix gives a rigorous justification of most of the claims in the main text (although some references and pointers are missing). I've only read the appendix until Section C and cannot assess that the remainder (the coupling analysis, which is the most substantial part) is correct, but the sketch in the main text is convincing.

**Weaknesses:**

Minor comments:

L98: "To recover the matrix,"  --> "To recover the matrix $M^{\star}$,"

L110: "controlling the $\ell_{\infty}$-norm in [14]" --> $\ell_{\infty}$-norm of what?

Notation $\lesssim$ is not defined

L166: ".. linear combination of $x^{(0)}$ and $u^{\star}$, .." --> ".. linear combination of $x^{(0)}$ and $u^{\star}$ (see eq. (C.1) in the appendix)"

L214: "becomes" --> "is"

L233 and L234: "between" --> "of"

L259: ".. parallel to $u^{\star}$." --> ".. parallel to $u^{\star}$ (see Lem. A.5 in the appendix)."

L286: ".. in both $\ell_{2}$ and $\ell_{\infty}$ norms, .." --> ".. in both $\ell_{2}$ and $\ell_{\infty}$ norms (see Cor. 1 in the appendix), .."






**Questions:**

eq. (8): Is it intentional to not to cancel out $n^{1/4}$ factor in the upper bound?

L216: Shouldn't $\tilde{x}^{(1)}$ here be $x^{(0)}$?

Lemma 5.5: $u^{(l)}$ is not defined. Same as $u^{\star}$ but 0 at the $\ell$th coordinate?

---

> ### Author Rebuttal · Authors · 2023-08-09
>
> We sincerely thank the reviewer for the feedback. Please see our response to the reviewer’s question.
>
> >**Minor Comments: ...**
>
> We will change the manuscript accordingly. Thanks for your careful review.
>
> ---
>
> >**eq. (8): Is it intentional to not to cancel out $n^{1/4}$ factor in the upper bound?**
>
> Yes, it was intentional to not to cancel out $n^{1/4}$. We tried to emphasize that when optimal samples are provided, i.e. $np = poly(\log n)$, the initialization size should be less than $n^{-1/4}$.
>
> ---
>
> >**L216: Shouldn't $\tilde{\mathbf{x}}^{(1)}$ here be $\mathbf{x}^{(0)}$?**
>
> We tried to compare the norms of $\mathbf{x}^{(1)} - \tilde{\mathbf{x}}^{(1)}$ to those of $\tilde{\mathbf{x}}^{(1)}$, so $\tilde{\mathbf{x}}^{(1)}$ is the right one.
>
> ---
>
> >**Lemma 5.5: $\mathbf{u}^{(l)}$ is not defined. Same as $\mathbf{u}^\star$ but 0 at the $l$th coordinate?**
>
> $\mathbf{u}^{(l)}$ is defined at the end of Lemma 5.5. $\mathbf{u}^{(l)}$  is the first eigenvector of $\mathbf{M}^{(l)}$ .

---

### Official Review · Reviewer_irzy · 2023-07-04

**Soundness:** 3 good
**Presentation:** 2 fair
**Contribution:** 2 fair
**Rating:** 5
**Confidence:** 2

**Summary:**

The authors study a random initialization scheme for gradient descent applied to the problem of rank-one matrices, assuming a known ground truth. Their results concern global convergence properties of the with respect to a particular random model of partially observed matrices. Specifically, starting from a $n\times n$ symmetric positive definite, fully-revealed ground truth matrix $\bf{M}^\star = \lambda^\star \bf{u}^\star (\bf{u}^\star)^T= \bf{x}^\star (\bf{x}^\star)^T$, entries above or on the diagonal are perturbed by noise drawn i.i.d. from a zero-mean Gaussian, and revealed independently with probability $p.$ The main result, Theorem 3.1, requires a coherence assumption: namely, for $\lVert \mathbf{u}^\star \rVert = \sqrt{\frac{\mu}{n}},$ the quantity $\mu$ is polynomially-bounded in $n.$ With this assumption, the main result claims that, if the Gaussian noise is sufficiently small and an initial iterate $\mathbf{x}_0$ is provided whose magnitude is not too large or small according to quantities depending on $n, \mu , \lambda^\star ,$ and $p,$ that gradient descent will converge to the given ground truth with probability tending to $1$ as $n\to \infty .$ In addition to giving proofs, the authors explain the qualitative behavior of convergence as determined by several phases, which appear in both the analysis and a simulation study.

**Strengths:**

It seems that the combination of leave-one-out sequences developed in [14] and the random initialization used in areas like phase retrieval are a novel aspect of this work, although as mentioned in Sec. 6 there is difficulty in extending this combination to the case of arbitrary rank. Theorems and definitions are, for the most part, stated unambiguously, and I was unable to find any errors. The topic is clearly a good fit for NeuRIPS.

**Weaknesses:**

One issue I have with this paper is that rank-1 symmetric matrix completion (as well as rank-1 general matrix completion) is simply a much easier problem than general matrix completion.

Indeed, I would like to point out the reference "Uniqueness of Low-Rank Matrix Completion by Rigidity theory", by Singer and Cucuringu, which is not cited in this work. Section 5 of this paper shows that the existence of an exact completion is guaranteed (with "probability one") based purely on combinatorial conditions of the graph of revealed entries. By contrast, the authors make high-probability statements about matrices whose entries are drawn and hidden according to specific distributional assumptions as the sample size goes to infinity, which seem to be much stronger. In general, the bibliography could be more extensive.

Additionally, equation (8) shows that there are _lower_ bounds in addition to upper bounds on the magnitude of the initialization that is needed. Thus, merely a "small" initialization is not enough to ensure convergence, contrary to the title. Finally, there is a nontrivial coherence assumption. This may be standard in compressed sensing, but it is still a nontrivial assumption.

In summary, the assumptions are overly restrictive, and, unlike the arbitrarily matrix completion problem, rank-1 matrix completion is a fairly simple problem. So, I think the paper's claimed results are not very interesting overall.

**Questions:**

line 3: Do you mean "simplest yet _most_ efficient?"

line 121: It would helpful to clarify that asymptotic notation like $o(1)$ refers to the regime $n\to \infty ,$ as opposed to other parameters tending towards infinity or zero.

Theorem 3.1: It seems that a corollary of this theorem would be that $\pm \bf{x}^\star$ are the only ground-truth solutions to the matrix completion problem. Is that something that is already assumed in the proof of your result? I don't see anywhere an explanation of why the cases with multiple ground truth solutions would be asymptotically rare.

---

> ### Author Rebuttal · Authors · 2023-08-09
>
> We sincerely thank the reviewer for the feedback. Please see our response to the reviewer’s question.
>
> >**One issue I have with this paper is that rank-1 symmetric matrix completion (as well as rank-1 general matrix completion) is simply a much easier problem than general matrix completion.**
>
> The question of whether the combination of GD with random initialization can effectively solve a specific problem holds significant importance in the field of optimization. Phase retrieval and matrix sensing are one of the low-rank recovery problems, like matrix completion. For those problems, the global convergence of GD from a randomly initialized point has been proved. However, despite its similarity to matrix sensing, no analogous result had been established for matrix completion. The problem remains open especially after the local convergence result [a] published in 2017.
>
> Although the rank-one case does not have much impact in itself, it will provide a good starting point for someone who challenges the full problem. Our novel analysis on Phase I provides a good understanding on the dynamics of GD that starts from a small random initializer. We expect that our key lemmas such as Lemmas 5.2 and 5.5 will continue to hold for the general rank-r case in a similar way. For Phase II, we have a difficulty in analyzing the singular values of the trajectory, and it is discussed in Section 8 with some simulation results.
>
> ---
>
> >**Indeed, I would like to point out the reference "Uniqueness of Low-Rank Matrix Completion by Rigidity theory", by Singer and Cucuringu, which is not cited in this work. Section 5 of this paper shows that the existence of an exact completion is guaranteed (with "probability one") based purely on combinatorial conditions of the graph of revealed entries. By contrast, the authors make high-probability statements about matrices whose entries are drawn and hidden according to specific distributional assumptions as the sample size goes to infinity, which seem to be much stronger. In general, the bibliography could be more extensive.**
>
> For more than a decade, the probabilistic model investigated in this paper has stood as the standard model for researchers studying matrix completion, following the pioneering work of [b]. There have been hundreds of papers studying matrix completion under the probabilistic model; therefore, it is difficult to assert that the model is overly restrictive. However, we will try to include the paper that the reviewer suggested in the final manuscript, as it offers a distinct perspective on matrix completion.
>
> ---
>
> >**Additionally, equation (8) shows that there are lower bounds in addition to upper bounds on the magnitude of the initialization that is needed. Thus, merely a "small" initialization is not enough to ensure convergence, contrary to the title.**
>
> An exceptionally small initialization size such as $e^{-n}$ could be detrimental to the convergence of GD because it extends convergence time from $\Theta(\log n)$ to $\Theta(n)$. Because we are deriving probabilistic bounds for all iterations, establishing an upper bound on the iteration count is imperative. Similar bound can also be found in previous work [a]. While imposing a limit on the number of iterations might seem counterintuitive when proving local convergence, Theorem 2 of reference [a] does that precisely by constraining the maximum iteration count to $O(n^5)$. Nonetheless, we can further reduce the lower bound $n^{-10}$ to $n^{-c}$ for any $c > 10$ by tuning some constant factors during the proof. We do not think this is a restrictive assumption.
>
> ---
>
> >**Finally, there is a nontrivial coherence assumption. This may be standard in compressed sensing, but it is still a nontrivial assumption.**
>
> The incoherence condition is another standard assumption that researchers follow when studying matrix completion. The seminal work [b] provides a good explanation on why such an assumption is required. If information of matrix is concentrated on only few entries, we will not be able to recover the matrix unless we observe those entries. The incoherence assumption ensures that information is distributed nearly evenly across all entries of the matrix.
>
> ---
>
> >**line 3: Do you mean "simplest yet most efficient?"**
>
> We will change the sentence to “a simple yet efficient”.
>
> ---
>
> >**line 121: It would helpful to clarify that asymptotic notation like $o(1)$ refers to the regime $n \to \infty$ as opposed to other parameters tending towards infinity or zero.**
>
> We will mention in somewhere that all asymptotic relations in this paper are with respect to $n$.
>
> ---
>
> >**Theorem 3.1: It seems that a corollary of this theorem would be that $\pm \mathbf{x}^\star$ are the only ground-truth solutions to the matrix completion problem. Is that something that is already assumed in the proof of your result? I don't see anywhere an explanation of why the cases with multiple ground truth solutions would be asymptotically rare.**
>
> We are not assuming that $\pm \mathbf{x}^\star$ are the only global minima. What Theorem 3.1 implies is that GD converges to $\pm \mathbf{x}^\star$ with high probability (even if some other global minima exist).
>
> ---
>
> [a] Cong Ma et al. "Implicit Regularization in Nonconvex Statistical Estimation: Gradient Descent Converges Linearly for Phase Retrieval, Matrix Completion, and Blind Deconvolution."
>
> [b] E. J. Candes and B. Recht, “Exact Matrix Completion via Convex Optimization”

---

> > ### Comment · Reviewer_irzy · 2023-08-12
> >
> > ''What Theorem 3.1 implies is that GD converges to $\pm \mathbf{x}^\ast $ with high probability (even if some other global minima exist).''
> >
> > Sorry, but I'm still confused. Suppose $\pm \mathbf{y}^\ast $ is another global minimum---does your theorem not also imply convergence to it w.h.p.?

---

> > > ### Author Response · Authors · 2023-08-13
> > >
> > > We are not assuming anywhere in the proof about the existence (or non-existence) of a point $\mathbf{y}^\star$ such that $f(\mathbf{y}^\star) = 0$ and $\mathbf{y}^\star \neq \pm \mathbf{x}^\star$. Theorem 3.1 asserts that GD will converge to $\pm \mathbf{x}^\star$ even if such a $\mathbf{y}^\star$ exists. Below we explain why such a counterintuitive phenomenon actually happens.
> > >
> > > Let us define $\mathcal{S}$ as the set of incoherent points, which is explicitly written as
> > > $$\mathcal{S} = \left\\{ \mathbf{x} : \Vert \mathbf{x} \Vert_\infty \lesssim \sqrt{\frac{\mathrm{poly} (\log n)}{n}} \Vert \mathbf{x} \Vert_2 \right\\}.$$
> > > Note that $\pm \mathbf{x}^\star \in \mathcal{S}$ by the incoherence assumption. It was proved in [a] that with high probability, there is no global minimum other than $\pm \mathbf{x}^\star$ in the set $\mathcal{S}$. However, we proved through Theorem 3.2 that the trajectory of GD remains in the incoherent region $\mathcal{S}$; the trajectory of fully observed case, $\tilde{\mathbf{x}}^{(t)}$, can be easily shown to be incoherent for all $t$, and both $\ell_2$ and $\ell_\infty$-norms of $\mathbf{x}^{(t)}$ is close to those of $\tilde{\mathbf{x}}^{(t)}$ by Theorem 3.2. Hence, even if such a $\mathbf{y}^\star$ exists, the GD converges only to $\pm \mathbf{x}^\star$, because the trajectory is only allowed to move inside $\mathcal{S}$, but $\mathbf{y}^\star$ must reside outside of $\mathcal{S}$.
> > >
> > > In summary, any global minimum other than $\pm \mathbf{x}^\star$ is **NOT** incoherent as proved in [a], but the whole trajectory of GD is incoherent by Theorem 3.2, so it cannot converge to such a global minimum.
> > >
> > > Final note we want to make is that [a] eliminated all global minimum other than $\pm \mathbf{x}^\star$ by a regularizer that penalizes non-incoherent points, but our result proves that GD converges to $\pm \mathbf{x}^\star$ without any regularizer due to the *implicit regularization* of GD (the trajectory is kept incoherent automatically).
> > >
> > > ---
> > >
> > > [a] R. Ge, J. D. Lee, and T. Ma, “Matrix Completion has No Spurious Local Minimum”

---

> > > > ### Comment · Reviewer_irzy · 2023-08-13
> > > >
> > > > Thanks for the clarification. This would be a helpful remark to include in revision. This level of detail, though perhaps unnecessary for those who are experts in the compressed-sensing approach to matrix completion, is actually very helpful for everyone else.
> > > >
> > > > I am very satisfied with the authors' other answers to my questions. I am considering raising my rating, but will continue to monitor discussions in the coming days.

---

### Official Review · Reviewer_kLXw · 2023-07-08

**Soundness:** 3 good
**Presentation:** 3 good
**Contribution:** 3 good
**Rating:** 6
**Confidence:** 3

**Summary:**

This paper studies the global convergence of vanilla GD for the rank-1 matrix completion problem. It is shown that with small random initialization and after a logarithmic number of steps, GD enters a region around the global minimizers in which linear convergence happens. The paper provides sufficient conditions on the initialization scale to ensure this phenomenon happens and shows a tradeoff between the initialization scale and the number of available samples. Illustrative simulations are provided.

**Strengths:**

The paper studies a topic of interest for the Neurips community, which it presents in a clear manner and for which it provides results of both practical and theoretical importance. The derivations in the main paper are cleanly carried out, and the reasoning behind them is well-presented.

**Weaknesses:**

- The sample complexity seems to be quite large (for example, compared to that in [1]). In this light, can the authors elaborate further on this aspect (my question is despite the further commentary in section 6)?

[1] C. Ma, K. Wang, Y. Chi, and Y. Chen, “Implicit regularization in nonconvex statistical estimation: Gradient descent converges linearly for phase retrieval, matrix completion, and blind deconvolution,” Foundations of Computational Mathematics, vol. 20, no. 3, pp. 451–632, 2020.

**Questions:**

- The nature of the presented proofs lies very much in the detail. While the authors did a good job providing a higher-level view of the proof strategy, and the writing is clear, the text is still difficult to follow at times. A possible improvement would be to include some pictorial description of the phases -- similar to the one in section 4, for the analysis carried out in sections 5, 6. This can significantly aid understanding, in my opinion.

- Figure 2 a: the green line is labelled as $\\| x^{(t)} - x^{\star} \\|$, but in the figure commentary it is written $\\|x^{(t)} \pm x^{\star}\\|$. Which one is the correct one?

- Figure 2 c: Perhaps instead the label "Distance" can be replaced with $\\|x^{(t)} \pm x^{\star}\\|$ for clarity.

**Limitations:**

The limitations are adequately discussed.

---

> ### Author Rebuttal · Authors · 2023-08-09
>
> We sincerely thank the reviewer for the feedback. Please see our response to the reviewer’s question.
>
> >**The sample complexity seems to be quite large (for example, compared to that in [1]). In this light, can the authors elaborate further on this aspect (my question is despite the further commentary in section 6)?**
>
> The global convergence of gradient descent for phase retrieval was proved in [a], and it also required more log factors ($\log^{13} n$) compared to the local convergence result [b] ($\log n$). An improved analysis compared to ours could reduce the required sample complexity to that in [b], but we want to point out that it is difficult as it was for phase retrieval.
>
> The global geometry of loss function is not as benign as the local geometry around the global minimum. Additionally, it is customary to partition the gradient descent trajectory into distinct phases during the analysis of global convergence. However, establishing precise bounds during phase transitions is usually challenging and we have to rely on extra sample complexity.
>
> ---
>
> >**The nature of the presented proofs lies very much in the detail. While the authors did a good job providing a higher-level view of the proof strategy, and the writing is clear, the text is still difficult to follow at times. A possible improvement would be to include some pictorial description of the phases -- similar to the one in section 4, for the analysis carried out in sections 5, 6. This can significantly aid understanding, in my opinion.**
>
> Due to space limit, we could not insert a figure for Sections 5 and 6 in the main text. We will at least include the figure in supplementary in the final version.
>
> ---
>
> >**Figure 2 a: the green line is labelled as $\Vert \mathbf{x}^{(t)} - \mathbf{x}^\star \Vert$, but in the figure commentary it is written $\Vert \mathbf{x}^{(t)} \pm \mathbf{x}^\star \Vert$. Which one is the correct one?**
>
> $\Vert \mathbf{x}^{(t)} \pm \mathbf{x}^\star \Vert$ is the correct one. We appreciate your careful review.
>
> ---
>
> >**Figure 2 c: Perhaps instead the label "Distance" can be replaced with $\Vert \mathbf{x}^{(t)} \pm \mathbf{x}^\star \Vert$ for clarity.**
>
> We will change the label accordingly.
>
> ---
>
> [a] Yuxin Chen et al. "Gradient descent with random initialization: Fast global convergence for nonconvex phase retrieval."
>
> [b] Cong Ma et al. "Implicit Regularization in Nonconvex Statistical Estimation: Gradient Descent Converges Linearly for Phase Retrieval, Matrix Completion, and Blind Deconvolution."

---

> > ### Comment · Reviewer_kLXw · 2023-08-16
> > **Thank you for your response**
> >
> > I thank the reviewers for their responses, which I read along with the other reviews and their respective responses. I maintain my score -- I think this paper makes a solid contribution, supported by well-carried-out proofs and a clear presentation, though having the downside of being restricted to rank one matrices.

---

### Official Review · Reviewer_KM6V · 2023-07-16

**Soundness:** 3 good
**Presentation:** 3 good
**Contribution:** 2 fair
**Rating:** 4
**Confidence:** 4

**Summary:**

This paper shows some convergence properties of gradient descent with small random initialization for rank-1 noisy matrix completion.

**Strengths:**

This paper uses a new approach (gradient descent with small random initialization) to solve the nonconvex formulation for rank-1 noisy matrix completion. It is shown that the GD trajectory will arrive at a local neighborhood (in both $\ell_2$ and $\ell_{\infty$ norm) of the ground truth within a number of iteration.

**Weaknesses:**

1. My major concern of this paper is a lack of theoretical novelty. After looking at the results and quickly go through the proof, I believe the proof idea of this paper is similar to that in [1], which focus on a general low-rank matrix sensing problem (we know that the low-rank matrix sensing problem with certain RIP condition has the same population-level loss function as the noisy matrix completion problem). For example, the analysis idea that the GD dynamic is close to a simplified linear evolution system in the initial phase thanks to the small initialization already appeared in [1]. Compared with [1] which works for general low-rank setting, this paper only works for the rank-1 case, which is more restricted. On the other hand, matrix completion problems are known to be more difficult than the matrix sensing problem in the sense that it requires incoherence and $\ell_{2,\infty}$ error analysis to show that the empirical loss concentrates around the population counterpart. This difficulty was not encountered in [1]. This paper uses leave-one-out analysis that has been widely used in low-rank estimation literature like [2,3] to track the $\ell_{\infty}$ error of the GD trajectory. The analysis does not seems to be challenging, if one is familiar with the above-mentioned literature. If I underestimate the technical novelty of the paper, I hope the authors could clarify and please highlight their technical novelty.

2. I am not satisfied with the convergence guarantees. The estimation error provided by Theorem 3.1 only shows that the output of the proposed algorithm is consistent, namely is $o(1)$. However state-of-the-art results for matrix completion already shows that GD with spectral initialization (which I believe is equivalent to small random initialization in some sense, since the initial phase of the latter algorithm is similar to some form of power method) achieves minimax-optimal estimation error [2]. Could the authors please explain why their analysis leads to looser error bound?

3. In addition, the noise condition in Theorem 3.1 is $\sqrt{np}$ times more stringent than that in [2]. Could the authors please explain why their analysis requires stronger noise conditions?

[1] Li, Yuanzhi, Tengyu Ma, and Hongyang Zhang. "Algorithmic regularization in over-parameterized matrix sensing and neural networks with quadratic activations." Conference On Learning Theory. PMLR, 2018.

[2] Ma, Cong, et al. "Implicit Regularization in Nonconvex Statistical Estimation: Gradient Descent Converges Linearly for Phase Retrieval, Matrix Completion, and Blind Deconvolution." Foundations of Computational Mathematics 20 (2020): 451-632.

[3] Chen, Yuxin, et al. "Gradient descent with random initialization: Fast global convergence for nonconvex phase retrieval." Mathematical Programming 176 (2019): 5-37.

**Questions:**

I have no question at this moment.

**Limitations:**

This paper does not have potential negative social impact.

---

> ### Author Rebuttal · Authors · 2023-08-09
>
> We sincerely thank the reviewer for the feedback. Please see our response to the reviewer’s question.
>
> >**1. My major concern of this paper is a lack of theoretical novelty.  ... If I underestimate the technical novelty of the paper, I hope the authors could clarify and please highlight their technical novelty.**
>
> The results presented in this paper cannot be obtained by just combining the previous works that the reviewer mentioned. The technical novelty of our work is at delicate analysis of both $\ell\_2$ and $\ell\_\infty$ norm in Phase I, which was not studied before.
> In order to show that the norms decrease exponentially at a rate less than 1, the strong convexity of Hessian matrix around $\mathbf{x}^\star$ was mainly used in [2]. In Phase I, we do not have such convexity, and the Hessian matrix implies that both $ \Vert \mathbf{x}^{(t)} - \tilde{\mathbf{x}}^{(t)} \Vert\_2$ and $ \Vert \mathbf{x}^{(t)} - \mathbf{x}^{(t,l)} \Vert\_2$ may grow at a rate $(1 + \eta \lambda^\star)$ in Phase I. However, with some simulation, we observed that both quantities do not increase exponentially, and we actually prove through Lemma 5.1 and 5.4 that the quantities increase polynomially with respect to $t$.
>
> Two techniques were developed to obtain such results. First, we introduced another sequence $\hat{\mathbf{x}}^{(t)}$ that evolves with simpler recursive equation than $\mathbf{x}^{(t)}$, and proved that $\Vert \hat{\mathbf{x}}^{(t)} - \tilde{\mathbf{x}}^{(t)} \Vert\_2 $ increases polynomially as stated in Lemma 5.2. We proved the lemma by expanding both sequences with matrix polynomial, and such an approach was not used in any of the previous work. The quantity $\Vert \mathbf{x}^{(t)} - \hat{\mathbf{x}}^{(t)} \Vert\_2 $ is allowed to grow exponentially with the rate $(1 + \eta \lambda^\star)$ as stated in Lemma 5.3, but since its initial size is proportional to $\beta\_0^3$, it is negligible compared to $\Vert \hat{\mathbf{x}}^{(t)} - \tilde{\mathbf{x}}^{(t)} \Vert\_2 $ in Phase I, thanks to the small initialization.
>
> Second, we proved that $(\mathbf{x}^{(t)} - \mathbf{x}^{(t,l)})$ is almost orthogonal to $\mathbf{u}^\star$ in Phase I (see Lemma 5.5). $ \Vert \mathbf{x}^{(t)} - \mathbf{x}^{(t,l)} \Vert\_2$ can only grow exponentially at the rate $(1 + \eta \lambda^\star )$ when it is parallel to $\mathbf{u}^\star$. However, by showing that they are almost orthogonal to each other in Phase I, the norm grows only linearly with respect to $t$. With this result, we finally succeed to show that the trajectory is close to the fully observed case in the sense of $\ell\_\infty$-norm and the norm is kept to $\frac{1}{\sqrt{np}} \frac{\beta_0}{\sqrt{n}}$ ignoring for log factors, as stated in (12) of Lemma 5.1. To conclude, although the same leave-one-out approach was used, different induction hypotheses from [2] were used in our work to reflect different geometry of Phase I.
>
> Note also that due to our delicate analysis of $\ell\_2$-norm in Phase I, the initialization size of up to $n^{-1/4}$ is allowed in our result, while the upper bound on initialization size reads $n^{-3/4}$ in [4], which generalizes the result of [1].
>
> ---
>
> >**2. I am not satisfied with the convergence guarantees. The estimation error provided by Theorem 3.1 only shows that the output of the proposed algorithm is consistent, namely is o(1). However state-of-the-art results for matrix completion already shows that GD with spectral initialization (which I believe is equivalent to small random initialization in some sense, since the initial phase of the latter algorithm is similar to some form of power method) achieves minimax-optimal estimation error [2]. Could the authors please explain why their analysis leads to looser error bound?**
>
> >**3. In addition, the noise condition in Theorem 3.1 is $\sqrt{np}$ times more stringent than that in [2]. Could the authors please explain why their analysis requires stronger noise conditions?**
>
> Both questions are answered in Section 3: Estimation Error of the main text and Section F of the supplementary material. To summarize, out current approach employs extra samples to enhance the upper bound on initialization size, which is evident from equation (8). (it reads $n^{-1/4} \sqrt[4]{np}$) Nonetheless, by keeping the initialization size to $n^{-1/4}$ regardless of sample complexity, we are able to obtain the minimax-optimal estimation error and the noise condition of [2]. We discussed about this tradeoff in Section 3 of the main text, while the potential adjustments required for proving in the context of a fixed initialization size are briefly outlined in Section F of the supplementary material. Hence, with extra samples, one has the option to either improve the estimation accuracy or to increase the initialization size to reduce the number of iterations.
>
> ---
>
> [1] Li, Yuanzhi, Tengyu Ma, and Hongyang Zhang. "Algorithmic regularization in over-parameterized matrix sensing and neural networks with quadratic activations." Conference On Learning Theory. PMLR, 2018.
>
> [2] Ma, Cong, et al. "Implicit Regularization in Nonconvex Statistical Estimation: Gradient Descent Converges Linearly for Phase Retrieval, Matrix Completion, and Blind Deconvolution." Foundations of Computational Mathematics 20 (2020): 451-632.
>
> [3] Chen, Yuxin, et al. "Gradient descent with random initialization: Fast global convergence for nonconvex phase retrieval." Mathematical Programming 176 (2019): 5-37.
>
> [4] Dominik Stöger and Mahdi Soltanolkotabi “Small random initialization is akin to spectral learning: Optimization and generalization guarantees for overparameterized low-rank matrix reconstruction”

---

### Official Review · Reviewer_ewei · 2023-07-27

**Soundness:** 3 good
**Presentation:** 3 good
**Contribution:** 2 fair
**Rating:** 4
**Confidence:** 5

**Summary:**

This work considers the convergence analyses of gradient descent method for rank-one matrix completion problem. In particular, this work assumes small random initialization, which is relatively relaxed condition compared to existing work. With such assumption, the logarithmic convergence of the gradient descent method has been proved in this work. Also, the impact of the regularization for gradient descent method has been analyzed.

**Strengths:**

The motivation of this work is clear and valid, also this work is well-organized. Meanwhile, this work is technically sound, where the proof and related analyses are provided, also some simulations are provided to further support the major results.

**Weaknesses:**

- I have concern for the novelty or the contribution of this work. Rank-one matrix completion problem is the simplest problem for matrix completion problems. Also there have constraints for the noise assumption of the revealed entries in this paper. For such cases, there are many efficient methods to deal with, like alternating minimization or projected gradient descent method. For gradient descent method, there also have existing works which have proved its convergence. Though this work provided with relaxed conditions, such contribution may be limited for the optimization and machine learning community, let alone the real problems in industry.
- The presentation for the proof of the major results can be further improved. For instance, the proof of Lemma 5.1 is separated into two parts at different places, it may be better to combine them together after finishing other proofs.



**Questions:**

- Can you compare the major results about the rank-one matrix completion problem with the following works?
https://arxiv.org/pdf/2008.04988.pdf
https://proceedings.neurips.cc/paper/2020/file/f86890095c957e9b949d11d15f0d0cd5-Paper.pdf

**Limitations:**

Please see comments above.

---

> ### Author Rebuttal · Authors · 2023-08-09
>
> We sincerely thank the reviewer for the feedback. Please see our response to the reviewer’s question.
>
> >**I have concern for the novelty or the contribution of this work. Rank-one matrix completion problem is the simplest problem for matrix completion problems. Also there have constraints for the noise assumption of the revealed entries in this paper. For such cases, there are many efficient methods to deal with, like alternating minimization or projected gradient descent method. For gradient descent method, there also have existing works which have proved its convergence. Though this work provided with relaxed conditions, such contribution may be limited for the optimization and machine learning community, let alone the real problems in industry.**
>
> The question of whether the combination of GD with random initialization can effectively solve a specific problem holds significant importance in the field of optimization. Phase retrieval and matrix sensing are one of the low-rank recovery problems, like matrix completion. For those problems, the global convergence of GD from a randomly initialized point has been proved. However, despite its similarity to matrix sensing, no analogous result had been established for matrix completion. The problem remains open especially after the local convergence result [a] published in 2017.
>
> Although the rank-one case does not have much impact in itself, it will provide a good starting point for someone who challenges the full problem. Our novel analysis on Phase I provides a good understanding on the dynamics of GD that starts from a small random initializer. We expect that our key lemmas such as Lemmas 5.2 and 5.5 will continue to hold for the general rank-r case in a similar way. For Phase II, we have a difficulty in analyzing the singular values of the trajectory, and it is discussed in Section 8 with some simulation results.
>
> For the noise model, we followed the standard of many previous works on low-rank recovery, and we do not think it is a strong and restrictive assumption. It is common to model noise with the Gaussian distribution in different engineering fields although it may not be appropriate for some specific problems.
>
> ---
>
> >**The presentation for the proof of the major results can be further improved. For instance, the proof of Lemma 5.1 is separated into two parts at different places, it may be better to combine them together after finishing other proofs.**
>
> The lines from 227 to 243 are all related to the proof of (11), and the lines from 246 to 263 are all related to the proof of (12). They are not separated.
>
> ---
>
> >**Can you compare the major results about the rank-one matrix completion problem with the following works? (Rui Liu and Alex Olshevsky, 2020) and (Qianqian Ma and Alex Olshevsky, 2020)**
>
> A special error model was studied in [b]. For some fractional number of rows and columns, the observed matrix is allowed to be corrupted entirely by any order. Based on an algorithm similar to alternating minimization, the corrupted rows and columns are found recursively, and those rows and columns are not used in the estimation of singular vectors. The main theorem gives the required sample complexity as a function of the fraction of corrupted rows and columns, and it is implied from the theorem that at most a fraction of $1 / \log \log(n)$ rows can be corrupted.
>
> Because [b] uses different error model with ours, a direct comparison is difficult. However, our model allows Gaussian noise to be added to every entry of the observed matrix not to some fractional number of entries. Moreover, the variance of Gaussian noise can be as large as $\sqrt{\log n}$ times the maximum entry of the ground truth matrix. Our main theorem shows that by increasing the initialization size, convergence time is reduced if more than optimal number of samples ($n \mathrm{poly} (\log n)$) are provided. In [b], however, only the necessary and sufficient condition on the sample complexity was analyzed, and how extra samples affect the performance or convergence speed of the algorithm was not provided.
>
> [c] studies the alternating minimization algorithm for rank-1 matrix completion problem and convergence of the algorithm was analyzed. However, both [b] and [c] assume that every entry of the ground truth matrix is positive. We suspect that such an assumption is required to show that singular vectors (x, y) at each round is incoherent. Our result does require such an assumption.
>
> Lastly, we would like to mention that the main purpose of this work is not to insist that GD with small random initialization is the best algorithm to use when solving rank-1 matrix completion problem. Our prior interest is at explaning the convergence of GD for nonconvex problems depite the existence of local minima or saddle points, which also gains much attention in machine learning and optimization community in recent years.
>
> ---
>
> [a] Cong Ma et al. "Implicit Regularization in Nonconvex Statistical Estimation: Gradient Descent Converges Linearly for Phase Retrieval, Matrix Completion, and Blind Deconvolution."
>
> [b] Q. Ma and A. Olshevsky, “Adversarial Crowdsourcing Through Robust Rank-One Matrix Completion”
>
> [c] R. Liu and A. Olshevsky, "Asymptotic Convergence Rate of Alternating Minimization for Rank One Matrix Completion"

---

> > ### Author Response · Authors · 2023-08-13
> >
> > There is a typo in the third paragraph of the answer to the last question. The last sentence should be fixed to "Our result does **NOT** require such an assumption". We are sorry for the mistake.

---

### Decision · Program_Chairs · 2023-09-21

**Decision:**

Accept (poster)

**Comment:**

The manuscript does a really outstanding job at explaining the question they pose, that of rank one matrix completion using gradient descent from small random initialisation. The main criticism of the manuscript are concerns about how interesting new results are about the rank one matrix completion question, and that many other manuscripts in the area are not cited; these are fair criticism that I completely agree with. That said, those concern focuses on the results themselves and don't take as much account of the technique of proof a I think is warranted. I agree with the reviewer that is most supportive of acceptance, that the manuscript gives a lot of scope for impacting future study of the question, in particular for higher rank. The value of the manuscript is in the clarity of the proof, see section 4-6, which explain not just how the result are derived, but give good quantitative explanations as to how the algorithm acts in practise. This is explained through a direction finding and then a expansion phase. This is explained so nicely that one imagines it can be extended to the larger rank setting and might also give insight into how to solve related questions. It is this clarity and accuracy of the proof that shine for the manuscript. That said, one can rightly wonder if this approach is ever going to be a sensible method in real applications as the ability to explain the behaviour comes from a very poor starting point and slow step size; surely the one step method of scaled rank projection in [10] will always be preferred to this extremely slow gradient descent approach. That said, I think there is intellectual merit in method of proof which might be applied effectively in other settings.